

# Simple process-led algorithms for simulating habitats (SPLASH v.2.0): calibration-free calculations of water and energy fluxes

David Sandoval[1], Iain Colin Prentice[1,2,3], and Rodolfo L.B. Nóbrega[4]

[1]Georgina Mace Centre for the Living Planet, Department of Life Sciences, Imperial College London, Ascot, UK
[2]Department of Biological Sciences, Macquarie University, North Ryde, NSW, Australia
[3]Department of Earth System Science, Institute for Global Change Studies, Tsinghua University, Beijing, China
[4]School of Geographical Sciences, University of Bristol, Bristol

**Correspondence:** David Sandoval (d.sandoval17@imperial.ac.uk)

**Abstract.**

The current representation of key processes in Land Surface Models (LSM) for estimating water and energy balances still relies heavily on empirical equations that require calibration oriented to site-specific characteristics. When multiple parameters are used, different combinations of parameter values can produce equally acceptable results, leading to a risk of obtaining "right

answers for wrong reasons", compromising the reproducibility of the simulations and limiting the ecological interpretability of the results. To address this problem and reduce the need for free parameters, here we present novel formulations based on first-principles to calculate key components of water and energy balances, extending the already parsimonious SPLASH model v.1.0 (Davis et al. 2017, GMD). We found analytical solutions for many processes, enabling us to increase spatial resolution and include the terrain effects directly in the calculations without unreasonably inflating computational demands.

This calibration-free model estimates quantities such as net radiation, evapotranspiration, condensation, soil water content, surface runoff, subsurface lateral flow and snow-water equivalent. These quantities are derived from readily meteorological data such as near-surface air temperature, precipitation and solar radiation, and soil physical properties. Whenever empirical formulations were required, e.g. pedotransfer functions and albedo-snow cover relationships, we selected and optimized the best-performing equations through a combination of remote sensing and globally distributed terrestrial observational datasets.

Simulations at global scales at different resolutions were run to evaluate spatial patterns, while simulations with point-based observations were run to evaluate seasonal patterns using data from hundreds of stations and comparisons with the VIC-3L model, demonstrating improved performance based on statistical tests and observational comparisons. In summary, our model offers a more robust, reproducible, and ecologically interpretable solution compared to more complex LSMs.



## 1 Introduction

Robust representations of water and energy fluxes provide essential foundations for the analysis of interactions and feedbacks within soil, atmosphere and vegetation continuum in complex Land Surface Models (LSMs) (Wang et al., 2014; Prentice et al., 2015). These fluxes are greatly shaped by complex topography, which determines the amount of solar energy received at the surface; and by gradients of atmospheric pressure, temperature and moisture, soil development and gravitational potential energy, which together control vegetation dynamics and the emergent spatial patterns of ecosystem composition and structure (Tromp-van Meerveld and McDonnell, 2006; Körner, 1999; Sarmiento, 1986). Current models represent the complexity of topographic effects in various simplified ways, for example through discretization of the spatial continuum into hydrological response units (Grayson and Blöschl, 2000; Rodell et al., 2004), or predefined biomes (Liang et al., 1996) or stochastic representation of terrain and land cover at subgrid scales (Lawrence et al., 2019; Liang and Xie, 2001). Or, alternatively, models designed for large-scale applications may simply disregard terrain effects (Davis et al., 2017). This approach has arisen because models typically divide the soil-vegetation-atmosphere column into layers, or small storages, resulting in a large computational demand. So, to run models at higher resolution would increase the required computing power exponentially (Clark et al., 2017).

Although the higher precision of numerical schemes increases the accuracy of the models, the representation of some core hydrological processes still relies on empirical equations that require site-specific calibration (Clark et al., 2017). One outcome of this process is that different combinations of parameter values can produce equally acceptable results, implying a risk of obtaining "right answers for wrong reasons" (Grayson and Blöschl, 2000; Prentice et al., 2015), compromising the reproducibility of simulations, and limiting the ecological interpretability of the results obtained.

The use of optimization algorithms for multiple parameters in ever more complex models may not necessarily improve matters and, indeed, may hide the inadequacy of concepts such as "field capacity" and "permanent wilting point" when representing one of the most important ecological quantities, the soil water availability to plants. Although these constructs make sense conceptually, they can be misleading. For example, field capacity is described as the remaining water in the soil after drainage has ceased (Kramer and Boyer, 1995; Veihmeyer and Hendrickson, 1931). Still, its value is found in laboratory tests using small soil cores, and it is arbitrarily assumed to be equivalent to the water left after applying 33 kPa of suction (or 10 kPa in sandy soils). The permanent wilting point by definition depends on the plant as well as soil properties. Nonetheless, it is assumed by convention to be equivalent to the water left after applying 1.5 MPa of suction (M. B. Kirkham, 2005). Such values are upscaled globally using pedotransfer functions (PFTs), assuming they represent conditions found in nature, but their validity is virtually impossible to test using current LSMs.

The SPLASH model (Davis et al., 2017), is a highly parsimonious, multi-purpose set of algorithms mainly designed for ecohydrological and bioclimatic analysis: see e.g. Harrison et al. (2010); Gallego-Sala and Prentice (2012); Ukkola et al. (2015). Even though the original SPLASH assumes a flat cell, neglecting terrain influence on the fluxes, it includes explicit effects of elevation on biophysical quantities with minimum meteorological inputs. At its core, it conceptualizes the daily cycles of water and energy fluxes, and it solves their respective budgets using analytical integrals at a daily timestep (Cramer and Prentice,



1988; Davis et al., 2017). We propose new formulations to extend the original SPLASH using theory and concepts based on first principles, thus minimizing the need for free parameters while allowing the representation of processes in complex terrain.

To improve the calculations of the energy fluxes we adapted SPLASH v1.0 mathematical framework to use shortwave radiation as input instead of cloudiness as proxy for it. Furthermore, we included terrain (slope and aspect) effects on the analytical integrals of the daily energy fluxes and updated the empirical functions used to estimate net longwave radiation.

Since one of the main applications of SPLASH is to infer the water limitation on photosynthesis (Wang et al., 2014; Stocker et al., 2018), we no longer consider the available plant water capacity as a constant value and added the calculation of subsurface

flows. Here, we enhanced SPLASH with an analytical solution for the Green-Ampt equation to calculate daily infiltration, including corrections for slope effects; and analytical solutions for lateral flow, water viscosity effects on hydraulic conductivity, and Dunne and/or Hortonian runoff generation. To upgrade the "bucket model" used in the estimation of soil water content in SPLASH we have included soil hydrophysical properties estimated by PTFs and proposed a theoretical field capacity found by equilibrating gravity with capillarity force. This new version of SPLASH also includes an analytical solution to estimate soil

moisture at any depth; and a simple snowpack module, which accounts for snowfall occurrence, snow mass balance and effects on albedo. Processes that still require empirical formulations in the model (i.e., snowfall occurrence, snow-albedo feedback, and the effect of soil physical properties on the water retention curve) were optimized using "big data" from remote sensing and in situ measurements.

Some simplifications were adopted in order to allow analytical solutions, based on the prevalence of shallow soils and

impervious bedrock in mountain regions around the world:

1.  The drop of the saturated hydraulic conductivity with depth is neglected.

2.  Soil moisture redistribution through the soil profile (down to 2 m) takes no longer than one day.

3.  Water fluxes in the soil column are in a steady state.

4.  The shape of the moisture profile follows Hilberts et al. (2005) and Fan et al. (2007).

5.  The snow temperature is 0°C, so implicitly the energy required to raise the snow temperature is neglected.

The proposed analytical solutions greatly reduce the computational demand compared to numerical schemes, enabling the model to perform calculations using global high-resolution datasets at daily or monthly timesteps, and to provide emergent spatial patterns of key model outputs such as net radiation, snowpack size, lateral flow, surface runoff, condensation, evapotranspiration and soil water content.

The inputs of the model are precipitation, solar radiation and air temperature. To derive terrain information (slope, aspect and upslope contributing area) the algorithm requires a digital elevation model (DEM) when the grid functionality is used, but, if used with site-specific data (i.e. station data), these variables should be computed beforehand. To estimate some soil hydrophysical properties, the algorithm also requires soil texture, organic matter content and thickness.





## 2 Methods: Model description

### 2.1 Energy fluxes

#### 2.1.1 Surface solar radiation

Starting from the original formulation for extraterrestrial solar radiation flux $I_0$ ($W\ m^{-2}$) from SPLASH (Davis et al., 2017), which is defined as:

$$I_0 = I_{SC}\, d_r\, cos\,\theta_z \tag{1}$$

Where, $I_{SC}$ is the solar constant ($W\,m^{-2}$), $d_r$ ($unitless$) is the distance factor, and $cos\theta_z$ is inclination factor. The effects of the slope inclination and orientation on the surface solar radiation were included by using a more complex formulation of $cos\,\theta_z$, parameterized after Allen et al. (2006) as follows:

$$
\begin{aligned}
cos(\theta_z) = {}& sin(\delta)\,sin(\phi)\,cos(s) \\
& - sin(\delta)\,cos(\phi)\,sin(s)\,cos(\gamma) \\
& + cos(\delta)\,cos(\phi)\,cos(s)\,cos(h) \\
& + cos(\delta)\,sin(\phi)\,sin(s)\,cos(\gamma)\,cos(h) \\
& + cos(\delta)\,sin(\gamma)\,sin(s)\,sin(h)
\end{aligned}
\tag{2}
$$

Where, $\delta(\text{rad})$ is the declination angle between earth's equator and the sun at solar noon, and describes the seasonal changes at different latitude $\phi(\text{rad})$, the hour angle $h(\text{rad})$ describes the sun's position above the horizon, $s(\text{rad})$ is the slope inclination and $\gamma(\text{rad})$ is the slope orientation, or aspect, being $\gamma = 0$ for slopes oriented due south with its values increasing clockwise.

The hour angle when the solar radiation flux reaches the horizon or sunset hour[1] $h_s$ was found by replacing Eq. (2) in Eq. (1), setting $I_o = 0$, and solving for $h$, thus:

$$h_s = arccos\left(-\frac{sin(\delta)\,sin(\phi)\,cos(s)\,-\,sin(\delta)\,cos(\phi)\,sin(s)\,cos(\gamma)+cos(\delta)\,sin(\gamma)\,sin(s)\,sin(h_s)}{cos(\delta)\,cos(\phi)\,cos(s)\,+cos(\delta)\,sin(\phi)\,sin(s)\,cos(\gamma)}\right) \tag{3}$$

Furthermore, to simplify the notation, Eq. (3) can be rewritten as:

$$h_s = arccos\left(-\frac{r_u}{r_v}\right) \tag{4}$$

Where, $r_u = sin(\delta)sin(\phi)cos(s) - sin(\delta)cos(\phi)sin(s)cos(\gamma) + cos(\delta)sin(\gamma)sin(s)sin(h_s)$ and $r_v = cos(\delta)cos(\phi)cos(s) + cos(\delta)\,sin(\phi)\,sin(s)\,cos(\gamma)$, respectively. To account for the occurrences of polar days (i.e., no sunset) or polar nights (i.e., no sunrise), $h_s$ is set to $\pi$ when $r_u/r_v \geq 1$, and to zero when $r_u/r_v \leq -1$ respectively. Here, to approximate the value of $sin(h_s)$, the analytical solution proposed by Allen et al. (2006) is used as follows:

$$sin(h_s) = \frac{ac + b\sqrt{b^2 + c^2 - a^2}}{b^2 + c2} \tag{5}$$

---

[1]If we evaluate Eq. (3) for flat surfaces ($s = 0$), it becomes $h_s = arccos\left(-\frac{sin(\delta)\,sin(\phi)}{cos(\delta)\,cos(\phi)}\right)$, which is the original SPLASH equation described by Davis et al. (2017)





Where,

$$a = sin(\delta)\, cos(\phi)\, sin(s)\, cos(\gamma) - sin(\delta)\, sin(\phi)\, cos(s) \tag{6a}$$

$$b = cos(\delta)\, cos(\phi)\, cos(s) + cos(\delta)\, sin(\phi)\, sin(s)\, cos(\gamma) \tag{6b}$$

$$c = cos(\delta)\, sin(\gamma)\, sin(s) \tag{6c}$$

Therefore, the daily accumulated incoming radiation ($MJ\,m^{-2}\,d^{-1}$) is calculated as twice the integral of the Eq. (1), with $cos(\theta_z(h))$ ranging from solar noon ($h = 0$) to sunset ($h = h_s$), times the atmosphere's transmittance $\tau$ ($unitless$).

$$H = 2\int_0^{h_s} \tau I_0 = 2\int_0^{h_s} \tau I_{SC}\, d_r\, cos\theta_z dh = \frac{86400}{\pi}\,\tau\, I_{SC}\, d_r\, (r_u h_s + r_v\, sin(h_s)) \tag{7}$$

To exploit datasets of daily average incoming shortwave radiation $SW$ ($W\,m^{-2}$) and deprecate the empirical parameters in the previous model version which uses the classic Ångstrom–Prescott formula and cloudiness data, we set, $H = SW(W\,m^{-2}) * 86400(s\,d^{-1})$ in Eq. (7). Then multiplying both sides of Eq. (7) by $(1 - \beta_{SW})$, we solve for $\tau\, I_{SC}\, d_r\, (1 - \beta_{SW})$ to match the original formulation of the variable $r_w$ ($Wm^{-2}$) as follows:

$$r_w = \tau\, I_{SC}\, d_r\, (1 - \beta_{SW}) = \frac{SW\,\pi(1 - \beta_{SW})}{r_u h_s + r_v\, sin(h_s)} \tag{8}$$

Where $\beta_{SW}$ is the albedo and the other variables previously defined.

### 2.1.2  Net surface radiation

The net radiation flux at the surface, $I_N$ ($W\,m^{-2}$), is defined as the difference between the net shortwave radiation flux, $I_{SW}$ ($W\,m^{-2}$) and the net long-wave radiation flux, $I_{LW}$ ($W\,m^{-2}$),

$$I_N = I_{SW} - I_{LW} \tag{9}$$

Where, $I_{SW}$ is computed simply as the fraction of the incoming shortwave radiation flux not reflected by the albedo, $\beta_{SW}$ ($unitless$):

$$I_{SW} = SW(1 - \beta_{SW}) \tag{10}$$

$I_{LW}$ is computed in a similar fashion as the original SPLASH, by merging empirical formulations for clear and cloudy skies, both fitted using Eddy covariance data from the whole FLUXNET database, thus, replacing the old empirical formulations from (Monteith and Unsworth, 1990) and (Linacre, 1968) used in the first version.

$$I_{LW} = (k_4 + (1.0 - k_3)\, S_f)\, (k_1 + k_2\, T_{air}); \tag{11}$$

Where $k_{1-4}$ are empirical coefficients, $T_{air}$ ($°C$) is the daily mean near-surface air temperature and $S_f$ ($unitless$) is the sunshine fraction, derived from a general form of the Ångstrom–Prescott equation, with parameters fitted from global databases



according to Suehrcke et al. (2013):

$$s_f = \left( \frac{\tau - \tau_o \, k_5}{\tau_o \, (1 - k_5)} \right)^{(1/k_6)} \tag{12}$$

Where $k_{5-6}$ are empirical coefficients, $\tau$ is the atmosphere's transmittance, calculated as the ratio between $TOA$ solar radiation and surface $SW$ data, and $\tau_o$ is the clear sky atmospheric transmittance, computed following (Allen, 1996). Values for the coefficients are provided in the Table (1).

**Table 1.** Constants and Standard Values

| Variable | Value | Units | Description | Reference |
|---|---|---|---|---|
| $I_{SC}$ | 1360.8 | $W\,m^{-2}$ | solar constant Eq. (1) | (Kopp and Lean, 2011) |
| $\beta_o$ | 0.17 | – | shortwave background albedo Eq. (25) | (Federer, 1968) |
| $\beta_o^{snw}$ | 0.85 | - | new fallen snow albedo Eq. (26) | (Wang and Zeng, 2010; Barry, 1996) |
| $k_1$ | 91.86 | $(°C)$ | empirical constant Eq. (11) | (This study) |
| $k_2$ | 1.95 | – | empirical constant Eq. (11) | (This study) |
| $k_3$ | 0.20 | – | empirical constant Eq. (11) | (Linacre, 1968) |
| $k_4$ | 0.088 | – | empirical constant Eq. (11) | (This study) |
| $k_5$ | 0.1898 | – | empirical constant Eq. (12) | (Suehrcke et al., 2013) |
| $k_6$ | 0.7410 | – | empirical constant Eq. (12) | (Suehrcke et al., 2013) |
| $k_7$ | -0.5827 | – | empirical constant Eq. (19) | (This study) |
| $k_8$ | 1.319 | – | empirical constant Eq. (19) | (This study) |
| $k_9$ | $4.18 \times 10^{-4}$ | – | empirical constant Eq. (19) | (This study) |
| $k_{10}$ | $1.140 \times 10^{-2}$ | – | empirical constant Eq. (19) | (This study) |
| $k_{11}$ | 0.443 | – | empirical constant Eq. (26) | (This study) |
| $k_{12}$ | 0.895 | – | empirical constant Eq. (26) | (This study) |
| $L_f$ | 334000 | $J\,kg^{-1}$ | Latent heat of fusion Eq. (32) | (Monteith and Unsworth, 1990) |
| $g$ | 9.81 | $m\,s^{-2}$ | gravitational acceleration Eq. (19) | (Monteith and Unsworth, 1990) |
| $SWE_n^c$ | 140 | $mm$ | Minimum snow water equivalent for full snow cover Eq. (27) | (This study) |

The daily accumulated net radiation, $H_N$ $(MJ\,m^{-2}d^{-1})$, is calculated as net positive $H_N^+$ (daytime approximately), and net negative $H_N^-$ (night-time approximately), the threshold between $H_N^+$ and $H_N^-$ is the hour angle when $I_{SW}$ equals $I_{LW}$, $(h_n)$

(Fig.1). It is found by setting $I_N = 0$ in Eq. (9) as follows:

$$h_n = arccos \left( \frac{I_{LW} - r_w \, r_u}{r_w \, r_v} \right) \tag{13}$$



For cases where the net radiation is always positive $((I_{LW} - r_w r_u)/(r_w r_v) \geq -1)$ $hn$ is limited to $\pi$, while for the opposite cases, where the net radiation is always negative $((I_{LW} - r_w r_u)/(r_w r_v) \geq 1)$, $hn$ is limited to zero.

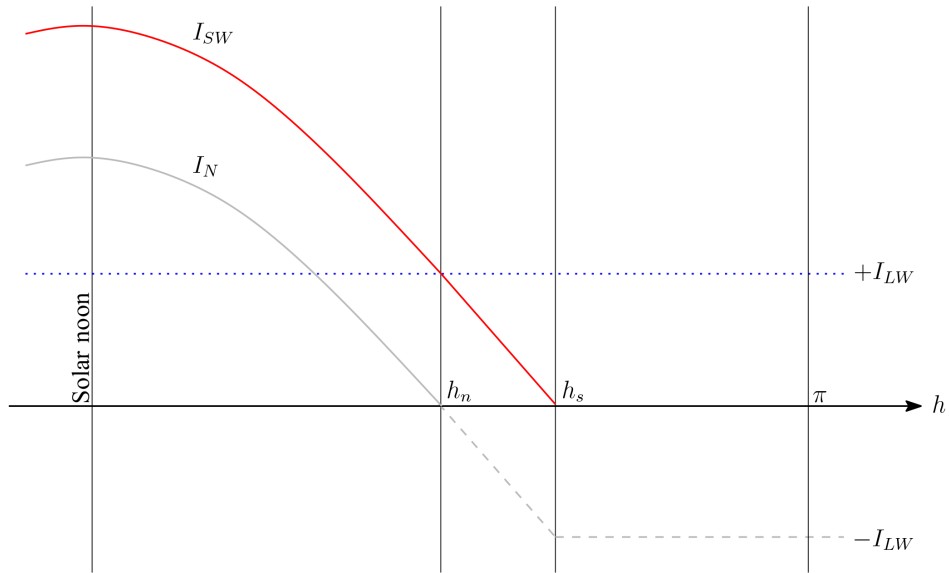

**Figure 1.** Conceptualization of the net radiation flux between solar noon (i.e., $h = 0$) and solar midnight (i.e., $h = \pi$) after Davis et al. (2017).

Therefore, as described by Davis et al. (2017), $H_N^+$ is defined as twice the integral of $I_N$ from solar noon to the flux cross-over hour angle $h_n$ (Eq.14), while $H_N^-$ is calculated as twice the integral of $I_N$ between $h_n$ and solar midnight ($h = \pi$) (Eq.15),

$$H_N^+ = 2 \int_{h=0}^{h_n} I_N = \frac{86400}{\pi} \left( (r_w r_u - I_{LW}) h_n + r_w r_v sin(h_n) \right) \tag{14}$$

$$H_N^- = 2 \left( \int_{h_n}^{h_s} I_N - \int_{h_s}^{\pi} I_{LW} \right) = \frac{86400}{\pi} \left[ r_w r_v (sin(h_s) - sin(h_n)) + r_w r_u (h_s - h_n) - I_{LW} (\pi - h_n) \right] \tag{15}$$

## 2.2 Water fluxes and storages

### 2.2.1 Snowfall

The freezing temperature of water, 0.0 °C is the usual threshold to categorize rainfall as snowfall in several models (Pomeroy and Brun, 2001), however, other atmospheric variables like cloudiness, atmospheric pressure and relative humidity are defining the snowfall formation (Jennings et al., 2018), therefore changing this temperature threshold in a narrow range across locations (Kienzle, 2008). Here, to get the rainfall/snowfall proportion it is usual to find, among different methods in the literature, linear approximations based on air temperature (Harder and Pomeroy, 2014; Marks et al., 1999; Rene Orth and Seneviratne, 2015),





or simply the 100% of the precipitation falling below 0.0 °C assigned as snowfall (Bergström, 1995; Dirmeyer et al., 2006) which might lead to miscalculations in some regions like the Alps, where up to 80% of its annual precipitation might be in the form of snowfall (Barry, 2008).

Therefore, in the current version of the model a sigmoid curve is used to describe the rain-snow proportion ($f_{rain}$) following Kienzle (2008), who fit empirical equations using 64 years of measurements of rainfall/snowfall proportions from 113 Canadian stations as follows:

$$f_{rain} = 5 \left( \frac{T_{air} - T_{tm}}{1.4 T_{rm}} \right)^3 + 6.76 \left( \frac{T_{air} - T_{tm}}{1.4 T_{rm}} \right)^2 + 3.19 \left( \frac{T_{air} - T_{tm}}{1.4 T_{rm}} \right) + 0.5 \tag{16}$$

Where, $T_{tm}$ ($C$) is the monthly temperature threshold for the 50% of rain-snow occurrence, and $T_{rm}$ ($°C$) is the monthly range

of temperatures for snowfall occurrence, both calculated according to:

$$T_{tm} = T_t + T_t \sin \left( \frac{m_i + 2}{1.91} \right) \tag{17}$$

$$T_{rm} = T_r \left( 0.55 + \sin(m_i + 4) \right) * 0.6 \tag{18}$$

Where, $m_i$ is a monthly index (from 1 to 12), $T_r$ ($°C$) is the annual range of temperatures for snowfall occurrence, found in 13°C as first approximation by Kienzle (2008). And $T_t$ ($°C$) is the annual threshold for snowfall formation, defined for each

170    year as the annual maximum air temperature when the probability of snowfall occurrence $p(snow)$ is equals or exceeds 0.5.

$p(snow)$ was estimated using a binary logistic regression, following the method and datasets provided by Jennings et al. (2018), but reducing the number of explanatory variables to air temperature $T_{air}$ ($°C$), elevation $z$ ($m.a.s.l$) and latitude $\phi$ (°) as follows (Appendix A4.3):

$$p(snow) = \frac{1}{1 + e^{(k_7 + k_8 \, T_{air} + k_9 \, z + k_{10} \, \phi)}} \tag{19}$$

Where, $k_7, k_8, k_9$ and $k_{10}$ are coefficients (Table 1). Then, the snowfall is calculated by,

$$Sf = P_n \left( 1 - f_{rain} \right) \tag{20}$$

### 2.2.2   Snowmelt

Snowmelt $Sm$ ($mm \, d^{-1}$) was calculated using a simple relationship between available energy and the size of the snowpack $SWE$ ($mm$) (snow water equivalent) as follows:

$$Sm = \min \left( SWE, \frac{H_N^+}{\rho_w L_f} * 1000 \right) \tag{21}$$

Where $SWE$ is the size of the snowpack expressed as snow water equivalent ($mm$), $H_N^+$ ($MJ \, m^{-2} d^{-1}$) is the daytime accumulated net radiation, $\rho_w$ ($kg \, m^{-3}$) is the water density at 0°C, $L_f$ ($J \, kg^{-1}$) is the latent heat of fusion, and 1000 is the factor to convert $m^3$ to $litres$. Following Barry (2008), we assumed direct-sublimation as negligible, however, if there is





residual energy after the melting occur, we directed that energy to evaporate $Sm$, flux hereafter denoted simply as "sublimation" ($E^{swe}$), which reduces the amount of $Sm$ reaching the soil or producing runoff.

$$E^{swe} = \min\left(S_m, \frac{H_A^+}{E_{con}} * 1000\right) \tag{22}$$

Where $H_A^+$ ($MJ\,m^{-2}d^{-1}$) is the daytime available energy, (daytime accumulated net radiation – energy used in melting), $E_{con}$ ($m^3 J^{-1}$) is the energy to water equivalent conversion factor (Davis et al., 2017) and 1000 is the factor to convert $m^3$ to $litres$. Thus, the water from snowmelt reaching the soil is:

$$Sm_e = Sm - E^{swe} \tag{23}$$

### 2.2.3 Snowpack

The size of the snowpack, expressed as snow water equivalent $SWE$ ($mm$) is computed as a simple balance using the previous-day $SWE$, inputs and outputs as follows:

$$SWE_n = SWE_{n-1} + Sf - Sm \tag{24}$$

The effect of the snow on the albedo was formulated as a simple weighted-average using the snow cover fraction, following Wang and Zeng (2010); Roesch and Roeckner (2006); Niu and Yang (2007),

$$\beta_{sw} = \beta_o * (1.0 - f_{snw}) + (f_{snw} * \beta_{snw}) \tag{25}$$

Where $\beta_o$ the is background albedo (Federer, 1968), $\beta_{snw}$ is the snow albedo, calculated according to the age of the snow, following the widely used formulation from the Corps of Engineers (1956)

$$\beta_{snw} = (\beta_o^{snw} - k_{11}) + k_{11}\,e^{-k_{12}\,n_d} \tag{26}$$

Here $\beta_o^{snw}$ is the albedo of the new fallen snow, $n_d$ is the number of days since a snowfall event greater than 3mm (Chen et al., 2014), and $k_{11,12}$ empirical constants. The snow cover fraction ($f_{snw}$) from Eq.(25) was estimated using a simple hyperbolic function following Dickinson et al. (1986) and Barry (1996).

$$f_{snw} = \frac{SWE_n}{SWE_c + SWE_n} \tag{27}$$

Where $SWE_c$ is the snow water equivalent where $f_{snw}$ starts to saturate.

The optimized parameters of eq. 26 and eq. 27 using remote sensing and ground observations are listed in the Table 1.

### 2.2.4 Infiltration

The infiltration flux rate $i$ ($mm\,h^{-1}$), was conceptualized as a two stages process, which can happen independently or one after another according to the magnitudes of the incoming flux $r$ ($mm\,h^{-1}$) (rain/snowmelt), and the infiltration capacity of the soil (Fig. 2).





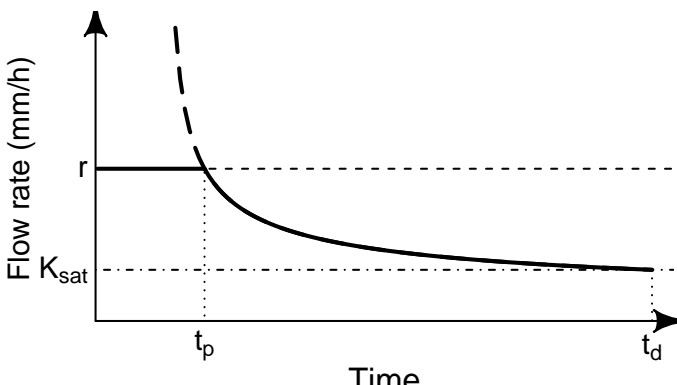

**Figure 2.** Conceptualization of the infiltration process with a constant rainfall $r$ (modified from Tindall et al. (1999)). Here, $K_{sat}$ is the saturated hydraulic conductivity, $t_p$ and $t_d$ stand for ponding and duration times respectively.

The first stage, usual when the soil is dry, describes an infiltration rate lower than the infiltration capacity, thus, it is limited by the rainfall/snowmelt rate, and lasts until the water flux starts to pond at $t_p$ (Vereecken et al., 2019; Assouline, 2013). A second stage describes the system once the water starts to pond on the surface, here the infiltration rate is limited by the infiltration capacity, which in turn is decreasing inversely to the water content of the soil, reaching its minimum value (equivalent to $K_{sat}$) at saturation following the Green-Ampt formulation, as described by Assouline (2013); Tindall et al. (1999).

$$i_{t+1} = \frac{dI}{dt} = K_{sat}\left(\frac{\psi_f(\theta_{sat} - \theta_t)}{I(t)} + 1\right) \tag{28}$$

Where $K_{sat}$ ($mm\,h^{-1}$) is the saturated hydraulic conductivity, $\theta_{t,sat}$ ($m^3 m^{-3}$) are the volumetric soil water content at the time $t$ and at saturation $sat$ respectively, $I(t)$ is the cumulative infiltration at the time $t$, and $\psi_f$ ($mm$) is the capillary head at the wetting front, which according to Tindall et al. (1999) is calculated as:

$$\psi_f = \frac{2 + 3\lambda}{1 + 3\lambda}\frac{\psi_b}{2} \tag{29}$$

Being $\lambda$ the pore-size distribution index ($unitless$), and $\psi_b$ ($mm$) the air-entry pressure, both shaping parameters of the soil-water retention curve proposed by Brooks and Corey (1964) (referred as BC model hereafter).

Therefore, the ponding time $t_p$ can be found by setting Eq.(28) equals to $r$, which yields:

$$t_p = \frac{K_{sat}\psi_f(\theta_{sat} - \theta_t)}{r(r - K_{sat})} \tag{30}$$





Moreover, to account for the slope (s) effects on $t_p$, the factor $\frac{1}{cos^2(s)}$, is used to reduce $t_p$, following the analysis of Morbidelli et al. (2018), thus, the cumulative infiltration is defined as follows:

$$I = rt_p + \int_{t_p}^{t_d} i\, dt = r\frac{t_p}{cos^2(s)} + K_{sat}\left(t_d - \frac{t_p}{cos^2(s)}\right) - \psi_f\, \Delta\theta\, ln\left(1 - \frac{r\frac{t_p}{cos^2(s)}}{\psi_f\, \Delta\theta}\right); r > K_{sat} \tag{31a}$$

$$I = rt_d; r \le K_{sat} \tag{31b}$$

Where $t_d$ is the duration of the precipitation event, and $\Delta\theta$ is the difference between $\theta_{sat}$ and the previous $\theta$ at the near soil

surface, which is calculated using an analytical solution of the Brooks and Corey (1964) model with the previous-day moisture and the depth of the profile (See section 2.7).

The set of equations presented above still requires rainfall intensity and the event's duration to perform the calculations, however, the "minimum inter-event time", which is used to define a precipitation event, is not consistent in the literature, and varies according to the author, location and application, ranging from 15 min to 24 hours (Dunkerley, 2008; Molina-Sanchis

et al., 2016), making it difficult to define a criteria for global applications. Therefore, as a simplification, an average daily rainfall duration was proposed instead, similar conceptually to the design storm, which is used in calculations for infrastructure design (Smith and Parlange, 1978). In this way, to find the average daily rainfall duration, the daily number of hours with precipitation and the daily precipitation amount were extracted from the Global Satellite Mapping of Precipitation (GsMap v6.0) dataset during the 2000-2014 period (Mega et al., 2014; Yamamoto and Shige, 2015) using the Google Earth Engine

platform (GEE), and the most frequent value was chosen ($6hrs$).

The parameters $\lambda$ and $\psi_b$, which shape the BC model, and $K_{sat}$ were estimated using pedotransfer functions detailed in Saxton and Rawls (2006) which use soil texture and soil organic matter (SOM) as inputs.

To account for the effects of temperature and atmospheric pressure on the water viscosity and hence on $K_{sat}$ (Fig. 3), we used the formula described by Hillel (1998):

$$K_{sat} = k_i\frac{\rho g}{\eta} \tag{32}$$

Where, $k_i\ (m^2)$ is the soil's intrinsic permeability, $\rho\ (kg\,m^{-3})$ is the water density, $g\ (m\,s^{-2})$ gravitational acceleration and $\eta$ ($Pa\,s$) is the dynamic viscosity. Thus, we simply used $K_{sat}$ from the pedotransfer functions, assuming $\frac{\rho}{\eta}$ at standard conditions to find $k_i$, which was later replaced in Eq. 32 using actual environmental conditions.



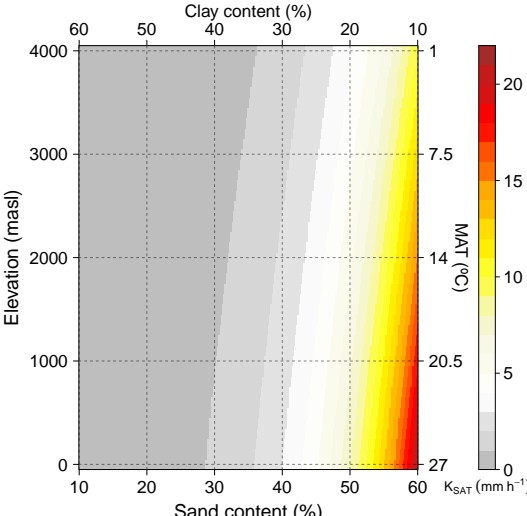

**Figure 3.** Effects of elevation (atmospheric pressure and temperature) on the saturated hydraulic conductivity, using a hypothetical soil with 10% SOM, 30% Silt, and varying Sand and Clay

### 2.2.5 Surface Runoff

The runoff formulation considers the different generation mechanisms: the saturation excess overland runoff $RO_D$ $(mm\,d^{-1})$ (Dunne runoff), which is produced after the soil became saturated and is frequent in humid climates or riparian areas (Vereecken et al., 2019), and, the infiltration excess overland runoff $RO_H$ $(mm\,d^{-1})$ (Hortonian runoff), which is produced when the precipitation rate exceeds the infiltration capacity, more frequent in semi-arid climates (Grayson and Blöschl, 2000; Vereecken et al., 2019).

$$RO_D = max(0, W_n - W_{sat}) \tag{33a}$$

$$RO_H = r - I \tag{33b}$$

Where, $r$ $(mm\,d^{-1})$ is water input (rainfall + snowmelt), $I$ $(mm\,d^{-1})$ is the infiltration, $W_{n,sat}$ $(mm)$ are the actual and soil water content at saturation respectively.

Therefore, the daily total $RO$ is simply defined as:

$$RO = RO_D + RO_H \tag{34}$$

### 2.2.6 Lateral flow

The lateral flow in one cell was defined at steady state as:

$$q_{in} = q_{out} \tag{35}$$




Where $q_{in}$ $(mm\,d^{-1})$ is the water draining into the cell from the upslope contributing area and $q_{out}$ $(mm\,d^{-1})$ is the water
draining out from the cell.

The lateral outgoing flow $q_{out}$ $(mm\,d^{-1})$ was conceptualized using some of the TOPMODEL's ideas (Beven and Kirby, 1979) on the profile transmissivity (soil hydraulic conductivity $K$ integrated over the soil column), and the hydraulic gradient defined by local topography $tan(s)$ as follows:

$$q_{out} = \frac{w}{A_i} \int_0^{z_s} K(\theta, z)\, dz\, \tan(s) \tag{36}$$

Where $w$ is the width of the profile's cross-section and $A_i$ is the area of the cell, used here to convert the volumetric flow through the cross-section to equivalent water column units over the cell.

In order to solve the transmittance, the soil moisture distribution through the profile was conceptualized following Hilberts et al. (2005) and Fan et al. (2007) (Fig. 4) with hydrostatic equilibrium at the water table (Remson and Randolph, 1962). Here, the soil moisture redistribution after infiltration was assumed to last shorter than one day within the first 2 m. of soil depth, 275  implying a permanent shape of the moisture profile.

Similarly to Fan et al. (2007), due to the lack of information on how $K_{sat}$ is decreasing with depth, the model assumes $K_{sat}$ constant through the first 2 meters of depth extending the original 1.5 m. proposed by Fan et al. (2007).

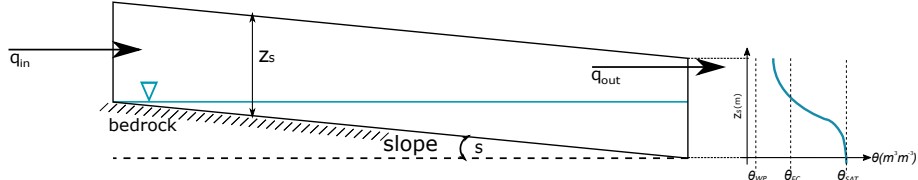

**Figure 4.** Conceptualization of the soil moisture profile in a shallow soil column, after Hilberts et al. (2005) and Fan et al. (2007).

Therefore, we defined the transmissivity of the profile as the sum of the transmissivities in the unsaturated and saturated parts of the profile as follows:

$$280 \quad T = \int_0^z K(\theta, z)\, dz = T_{uns} + T_{sat} = \int_0^{z_{wtd}} K(\theta, z)\, dz + \int_{z_{wtd}}^z K(\theta, z)\, dz \tag{37}$$

Thus, to approximate the distribution of $\theta(z)$ through the unsaturated part of the soil column, the BC model was used with the total soil-water potential (matric+gravitational) after Hino et al. (1988); Beldring et al. (1999).

$$\theta(z) = \theta_r + (\theta_{sat} - \theta_r)\left(\frac{\psi_m + \psi_g(z)}{\psi_b}\right)^{-\lambda} \tag{38}$$

Where, $\theta_r$ $(m^3 m^{-3})$ is the residual soil water content, $\psi_m$ $(mm)$ is the soil-matric potential and $\psi_g(z)$ $(mm)$ the gravitational
potential.





The hydraulic conductivity, $K(\theta)$ $(mm\,h^{-1})$, was defined according to Brooks and Corey (1964) as follows:

$$K(\theta) = K_{sat}\left(\frac{\theta}{\theta_{sat}}\right)^{\left(3+\frac{2}{\lambda}\right)} \tag{39}$$

Therefore, replacing Eq.(26) in Eq. (27), and solving the integral analytically for the unsaturated part, the transmissivity is defined as:

$$T_{uns} = \int_0^{z_{wtd}} K(\theta, z)\, dz = \frac{K_{sat}\psi_b}{3\lambda+1}\left[\left(\frac{\psi_b}{\psi_m}\right)^{3\lambda+1} - \left(\frac{\psi_b}{\psi_m + z_{wtd}}\right)^{3\lambda+1}\right] \tag{40}$$

Where $z_{wtd}$ $(m)$ is the depth to the water table, found when $\psi_m + \psi_g = \psi_b$.

While the transmissivity in the saturated part of the profile $T_{sat}$ is calculated as

$$T_{sat} = \int_{z_{wtd}}^{z} K(\theta, z)\, dz = K_{sat}(z - z_{wtd}) \tag{41}$$

The lateral incoming flux $q_{in}$ was formulated using a simple linear reservoir model (Buytaert et al., 2004; Yang et al., 2018; Vogel and Kroll, 1996), with a decaying volumetric flux as a function of time.

$$Q_f = Q_o\, K_b{}^t \tag{42}$$

Where $Q_f$ is the final flux after the time $t$ and $Q_o$ $(m^3\,d^{-1})$ is the initial volumetric flow soon after the precipitation event. If we set the cease of drainage at field capacity we get:

$$\lim_{t\to\infty} Q_o K_b{}^t = Q_{fc} \tag{43}$$

Where $Q_{fc}$ $(m^3\,d^{-1})$ is the volumetric flow after the time $t$ $(d)$, and $K_b$ is the recession constant $(unitless)$, which was found using the drainable porosity, see Appendix (A). Therefore, the total volume result of a daily $(1d)$ recharge $(R > 0)$ over the upslope area (Fig. 5), theoretically, can be approximated by:

$$A_u\, R\,(1d) = \int_{t_0}^{t} Q_o K_b{}^t dt \tag{44}$$

Where $A_u$ is the upslope area $(m^2)$, and $R$ is the recharge $(mm\,d^{-1})$, defined as infiltration minus evapotranspiration $R = I - E^a$.



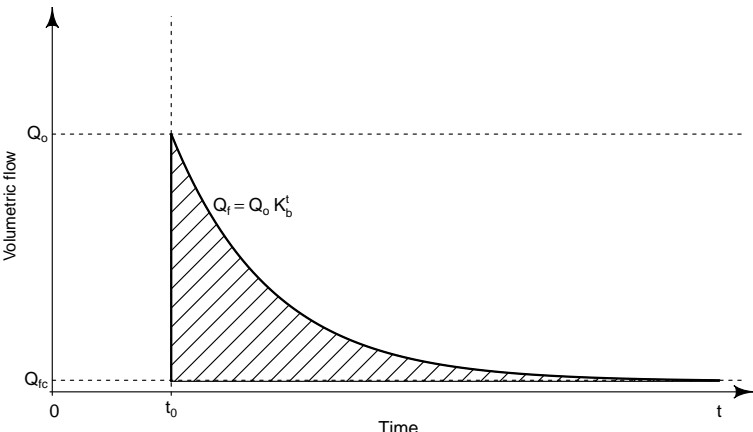

**Figure 5.** Conceptualization of the recharge from the upslope after one precipitation at $t_0$, using a simple linear reservoir model. The shaded area represents the Eq.(44).

Therefore, if we find $t$ from both, Eq.(43) and Eq.(44), we can set:

$$\frac{1}{\ln(K_b)} \ln\left(\frac{Q_{fc}}{Q_o}\right) = \frac{1}{\ln(K_b)} \ln\left(\frac{A_u R \ln(K_b)}{Q_o} + 1\right) \tag{45}$$

Which, solving for $Q_o$ yields:

$$Q_o = Q_{fc} - A_u R \ln(K_b) \tag{46}$$

Where $Q_{fc}$ can be found by setting $\theta$ to field capacity in Eqs.(38 and 40). Thus,

$$q_{in} = \begin{cases} \frac{Q_{fc} - A_u R \ln(K_b)}{Ai}; & \text{if } R > 0 \wedge \theta > \theta_{fc} \\ \frac{Q_{in\,n-1} K_b^{\Delta t}}{A_i}; & \text{if } R \le 0 \wedge \theta > \theta_{fc} \\ 0; & \text{if } \theta \le \theta_{fc} \end{cases} \tag{47}$$

### 2.2.7 Evapotranspiration

The actual evapotranspiration, $E_n^a \ (mm\,d^{-1})$ is computed following the original formulation Davis et al. (2017) with modifications to account for the reduction in available energy, which is diverted to snow melting and sublimation, if any. It starts

by defining the actual instantaneous evapotranspiration $E^a \ (mm\,h^{-1})$ as the minimum between supply $S_W \ (mm\,h^{-1})$ and demand $D_p \ (mm\,h^{-1})$ rates (Federer, 1982)

$$E^a = min(S_W, D_p) \tag{48}$$



Then, the daily integration of this flux is computed using an analogous form to the daily energy fluxes calculation (Fig. 6), as follows,

$$E_n^a = 2 \int_{h=0}^{h_n} E^a = 2 \left( \int_{h=0}^{h_i} S_W + \int_{h_i}^{h_n} D_p \right) \tag{49}$$

Where $h(\mathrm{rad})$ is the hour angle and $h_i$ is the cross-over angle when the supply is equal to the demand.

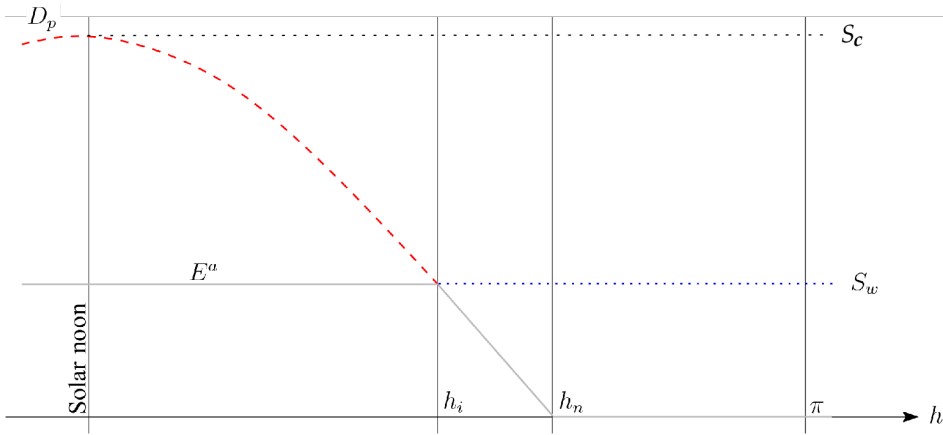

**Figure 6.** Conceptualization of the actual instantaneous evapotranspiration flux between solar noon (i.e., $h = 0$) and solar midnight (i.e., $h = \pi$), modified from Davis et al. (2017). The evaporative demand $D_p$ (red dashed line) is maximum at solar noon, equivalent to the maximum supply rate $Sc$. The actual supply rate $S_W$ is constant throughout the day, an depends on soil moisture limitations.

The evaporative demand $D_p$ is defined following the Priestley and Taylor (1972) formulation for potential evapotranspiration:

$$D_p = 3.6 \times 10^6 \, E_{con} \, I_N \tag{50}$$

Where $I_N (W \, m^{-2})$ is the instantaneous net radiation, and $E_{con} (m^3 \, J^{-1})$ is the energy-to-water conversion factor, defined following the Priestley-Taylor theory (hereafter PT), with adjustments proposed by Yang and Roderick (2019), which, reproduce the feedback between the surface temperature and $E^a$, hence their effect on $I_N$, thus, replacing the need of a Priestley-Taylor $\alpha_{PT}$ coefficient:

$$E_{con} = \frac{s}{L_v \, \rho_w (s + 0.24\gamma)} \tag{51}$$

Where $L_v (J \, kg^{-1})$ is the latent heat of vaporization, $\rho_w (kg \, m^{-3})$ is the water density, $s(Pa \, K^{-1})$ is the slope of the temperature-pressure curve, and $\gamma(Pa \, K^{-1})$ the psychrometric constant, and 0.24 is the constant defined by Yang and Roderick (2019). Equations for temperature and pressure dependencies to calculate $\rho_w$ and $\gamma$ were used, while only temperature-dependant equations were used for $s$ and $L_v$ (Davis et al., 2017).





The stress factor controlling the evaporative supply rate $S_W$ was conceptualized as a piece-wise linear function, where we assumed the stress follows the depletion of the water content in the plant available water region (Fig. 7).

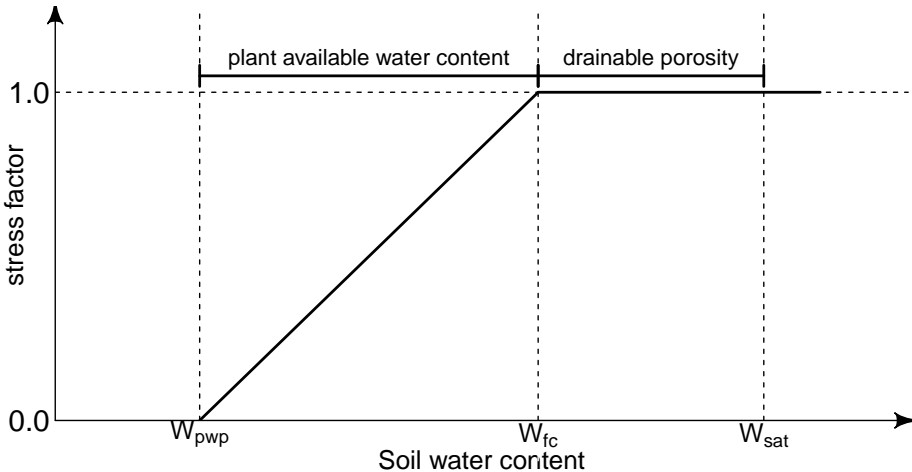

**Figure 7.** Conceptualization of the stress factor controlling the evaporative supply rate $S_W$

Therefore, $S_W$ is defined as:

$$S_W = \begin{cases} S_c \frac{W_{n-1}-W_{pwp}}{W_{fc}-W_{pwp}}; & \text{if } W_{pwp} \leq W_n < W_{fc} \\ S_c & \text{if } W_n \geq W_{fc} \end{cases} \tag{52}$$

Where, $S_c\ (mm\ h^{-1})$ is the maximum evaporative supply rate, and $W_{n-1}$ the previous day soil water content.

Here, Davis et al. (2017) adopts $S_c$ as a constant, following Federer (1982), however, the same Federer (1982) points out that this value should change according to morphological traits of the vegetation (i.e. root density and depth).

Therefore, since $E^a = min(S_W, D_p)$, and, under well watered conditions $S_W = S_c$, hence, $S_c$ with a value higher than the maximum $D_p$ (at solar noon) won't affect the resultant $E^a$.

Thus, to estimate $S_c$, so it applies for non-vegetated areas as well, the supply rate is approximated as the maximum rate of evaporation as follows,

$$Sc = D_{pMAX} = r_x \left( (r_w(r_u + r_v)) - I_{LW} \right) \tag{53}$$

Where, to simplify the notation, $r_x\ (mm\ m^2\ W^{-1}\ h^{-1})$ is equal to $3.6 \times 10^6\ E_{con}$.

The upper limit of the Eq.(52) is the water content at field capacity $W_{fc}\ (mm)$, which was defined as the amount of water held after the drainage ceased (Kramer and Boyer, 1995). This was calculated by setting the total water potential to equilibrium, following Remson and Randolph (1962):

$$\psi_m + \psi_g = 0 \tag{54}$$



Where, the matric potential $\psi_m$ was calculated following Saxton and Rawls (2006):

$$\psi_m = A(\theta)^{-B} \tag{55}$$

with,

$$A = e^{\ln(33) + B \ln(\theta_{33})} \tag{56a}$$

$$\theta = W_n/(1000z) \tag{56b}$$

$$B = \frac{1}{\lambda} = \frac{\ln(1500) - \ln(33)}{\ln(\theta_{33} - \ln(\theta_{1500})} \tag{56c}$$

Where, $\theta_{33}$ is the volumetric water content at 33 $kPa$, (usually assumed to be field capacity), $\theta_{1500}$ is the volumetric water content at 1500 $kPa$ and $z$ $(m)$ is the depth of the soil profile. Then, using the minimum between 2 m. and the depth to the bedrock as a reference plane, the gravitational potential is defined as:

$$\psi_g = \rho_w \, g \, W_n \tag{57}$$

Therefore, $W_{fc}$ can be found by solving the replacing Eqs. (55 to 57) in Eq. (54) and soving for $W_n$ (See Appendix A for intermediate steps):

$$W_{fc} = 1000z \left( \frac{1000 \, A}{\rho_w \, g \, z} \right)^{\frac{1}{1+B}} \tag{58}$$

The lower limit of the Eq. (52) is assumed to be the water content at permanent wilting point[2] $W_{pwp}$ $(mm)$, which was computed as:

$$W_{pwp} = \theta_{1500} * 1000z \tag{59}$$

In order to adopt the best option to compute soil hydro-physical properties ($\theta_{33}$, $\theta_{1500}$, $\theta_{sat}$ and $K_{sat}$) and hence the thresholds proposed, a set of the most widely used PTFs in LSMs were evaluated (Van Looy et al., 2017) were tested with a global dataset of soil physical properties which was compiled from different sources. The models, ranging in complexity, from the most simple mathematical formulations are described in Table 2.

---

[2]Even though the wilting point is an approximation, defined as the water extracted applying 1500kPa of pressure, it also depends on plant physiology, so its value, as a lower boundary, might be not entirely true, especially for dry ecosystems.





**Table 2.** Evaluated pedotransfer functions and their use in Land Surface Models

| PTF | LSM/Product | PTF formulation approach* | N samples used for training | Inputs |
|---|---|---|---|---|
| Cosby et al. (1984) | NASA CATCHMENT LSM, CLM 4.5, JULES, Noah-MP/VIC | MLR | 1448 | Sand (%), clay (%) |
| Balland et al. (2008) | – | NLR | 13088 | Sand (%), clay (%), SOM (%), bd ($g\,cm^{-3}$) |
| Saxton and Rawls (2006) | ESA CCI SM v03.2 | MLR | 5320 | Sand (%), clay (%), SOM (%) |
| Tóth et al. (2015) | Soil Hydraulic Database of Europe (ESDAC) | RT/LR | 2356-5530 | Sand (%), clay (%), SOM (%), bd ($g\,cm^{-3}$) |
| Rosetta 3 (Zhang and Schaap, 2017) | Noah-MP | ANN | 2134 | Sand (%), clay (%), SOM (%), bd ($g\,cm^{-3}$) |

*MLR, Multiple linear regression; ANN, Artificial neural networks; RT, regression trees; LR, Linear regression; NLR, non-linear regression.

Thus, finally Eq.(49) is solved analytically as:

$$E_n^a = \frac{24}{\pi}\left(S_W h_i + r_x r_v r_w(\sin h_n - \sin h_i) + (r_x r_u r_w - r_x I_{LW})(h_n - h_i)\right) \tag{60}$$

Where, the intersection hour angle $h_i$ is found by setting Eq.(50) equal to Eq. (52) and and solving for h:

$$h_i = \arccos\left(\frac{S_W}{r_x r_v r_w} + \frac{I_{LW}}{r_v r_w} - \frac{r_u}{r_v}\right) \tag{61}$$

**2.2.8 Condensation**

The daily dew formed by condensation $C_n(mm\,d^{-1})$ is assumed to represent 10% of the water equivalent (Eq.52) of the negative net radiation $H_n^-$ (Eq.15) (Jones, 2013). The remnant energy is assumed to be lost as convective heat. Thus,

$$C_n = 100\,E_{con}\,H_n^- \tag{62}$$

**2.2.9 Soil water content**

Once calculated inputs and outputs, the total soil water content $W_n$ ($mm$) can now be calculated using a simple balance expression, with the previous day soil water content $W_{n-1}$ as follows:

$$W_n = W_{n-1} + I + q_{in} - q_{out} - RO_H - E_n^a \tag{63}$$



Furthermore, to calculate the water content $SWC\,(mm)$ accumulated to any depth $(z')$ for further comparison with the observations, if we assume the same moisture profile from Eq. (26), it can be defined as:

$$SWC = \int_0^{z'} \theta(z)dz = \theta_r z + \left.\frac{(\psi_m + \psi_z)(\theta_r - \theta_{sat})\left(\frac{\psi_b}{\psi_m + \psi_z}\right)^{\lambda}}{\lambda - 1}\right|_0^{z'} \tag{64}$$

## 2.3 Initial conditions

The SPLASH algorithm assumed steady-state condition as the initial state for the simulations, which is reached by looping $ntimes$ the first year of data until the water balance is preserved:

$$\sum \left(f_{rain}P_n + C_n + Sm_n + q_n^{in}\right) = \sum \left(E_n^a + q_n^{out} + RO\right) \tag{65}$$

## 3 Methods: Calibration and validation data

### 3.1 Eddy covariance towers

Data from the whole FLUXNET database (Pastorello et al., 2020), comprising 212 stations distributed around the world (Fig. 8) was used to update the empirical functions to compute net longwave radiation, superseding the equations formulated by (Monteith and Unsworth, 1990) and (Linacre, 1968) used in the first version of SPLASH.

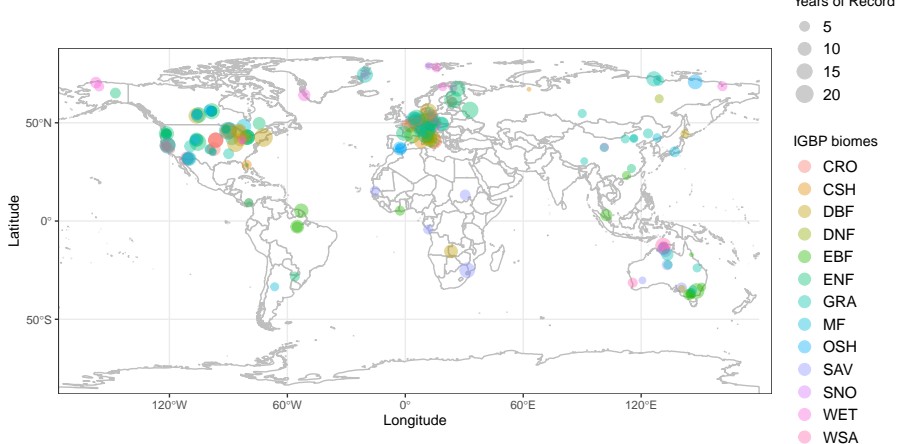

**Figure 8.** FLUXNET stations used for the $I_{LW}$ parameter calibration.

All the data was aggregated to daily means, while the originally reported latent heat flux was transformed to its equivalent in water flux density $(mm\,d^{-1})$ by using the heat of vaporization corrected for field conditions.



To test the validity of the theorized daily cycles and crossover angles, positive values of net radiation were subsetted and aggregated daily, which in theory should be equivalent to Eq.(14). A simple threshold for measured albedo of 0.3 was used to identify snow presence, then, latent heat measurements when snow was present were excluded from the estimations of daily

evapotranspiration and condensation, trying to prevent the latent heat used in melting/refreeze/sublimation introduce error in the evaluations.

## 3.2    Meteorological/hydrometric stations

To calibrate the parameters used in the snowcover/albedo functions, 315 stations reporting snow water equivalent or snow-depth from the SNOwpack TELemetry (SNOTEL) network (Serreze et al., 1999), managed by the U.S. Natural Resources

Conservation (NCAR) service were used together with remote sensing data (Fig. 9), most of these stations also report solar radiation, precipitation, air temperature and volumetric soil moisture at different depths.

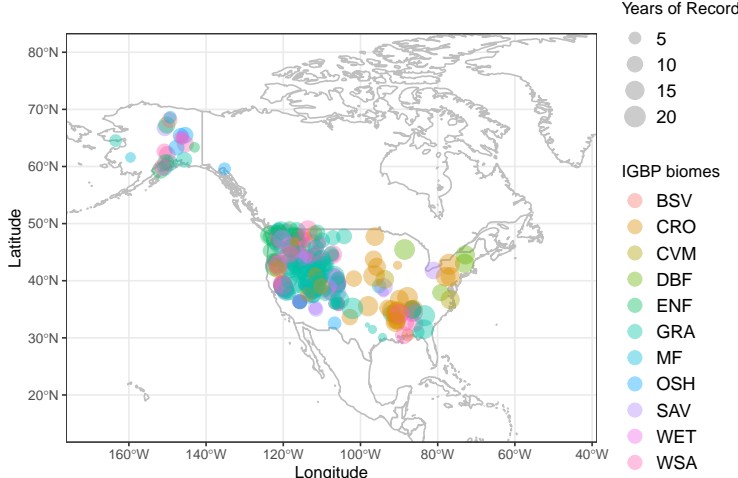

**Figure 9.** SNOTEL stations used for albedo/snow-cover analysis.

To evaluate the model, we subset sites on mountains regions with joint measurements of snow, soil physical properties (i.e. texture, bulk density and SOM), and soil moisture deeper than $30cm$, resulting in 127 sites (Table A4). Data from the DAYMET database (Thornton et al., 2018) was used whenever the solar radiation was not reported (Fig. 10).



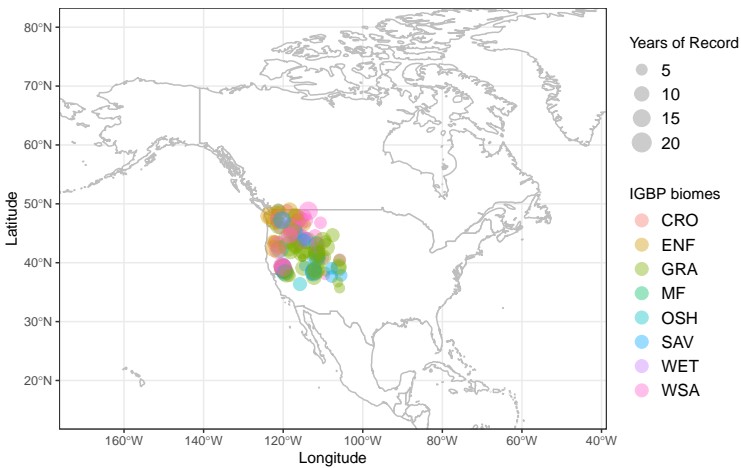

**Figure 10.** SNOTEL soil moisture and $SWE$ validation sites.

To fit the binary logistic regression used to estimate the probability of snowfall occurrence $p(snow)$, the dataset described
by Jennings et al. (2018) was used. Which comprises 11924 stations distributed over the northern hemisphere, adding a total
of 17810805 binary observations of snowfall (Fig. 11)

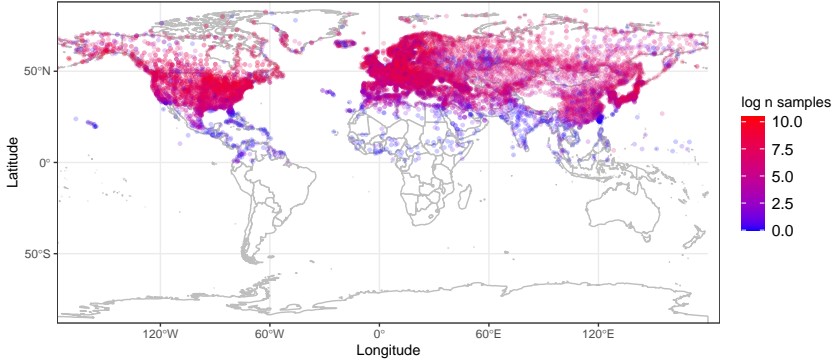

**Figure 11.** Stations providing snowfall observations from the Jennings et al. (2018) dataset

To test the model capabilities predicting streamflow, with its improvements accounting for slope, small watersheds with
areas between 5 and 2000 km$^2$ located in the mountain regions of Canada and the USA were selected from the global GSIM
database (Do et al., 2018) due to the quality of the available forcing data over these regions. The GSIM database provides
curated streamflow data from multiple sources and geographic watershed boundaries (Fig. 12). Only watersheds with natural
cover higher than 90% and streamflow data covering at least 10 years since 1980 were subset (Table A5), resulting in 15963
station-years. Here, the separation surface runoff/ baseflow was done following the method described by Ladson et al. (2013).





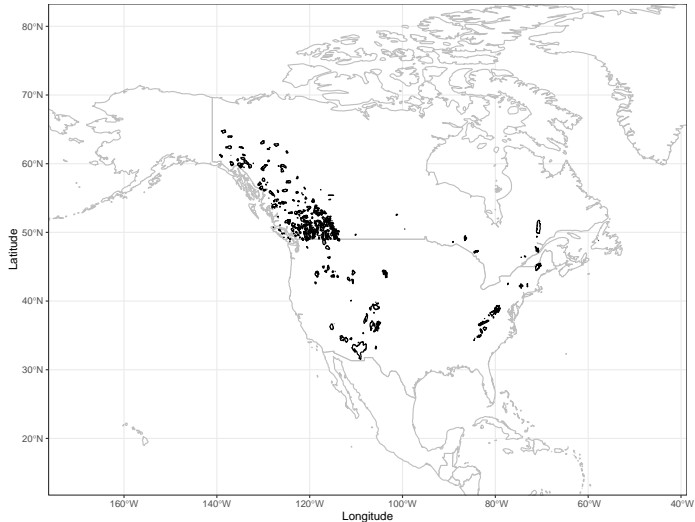

**Figure 12.** GSIM watersheds for streamflow validation.

To evaluate the spatial patterns produced by the model with long term data, hydro-meteorological and soil moisture mea-

surements from the Rietholzbach Research Catchment were used. These datasets are publicly available, and described by
Seneviratne et al. (2012); Hirschi et al. (2017). While, to test the model in regions were the rainfall/runoff response is mainly
dominated by subsurface flow (Crespo et al., 2011; Correa et al., 2020), hydro-meteorological data from the tropical Andes,
compiled by Ochoa-Tocachi et al. (2018) was used.

## 3.3   Soil physical properties and Terrain

To calibrate the pedotransfer functions which compute field capacity ($\theta_{33}$) and wilting point ($\theta_{1500}$) a dataset containing data
on water retention, texture, organic matter content ($SOM$), and bulk density was compiled from the U.S Natural Resources
Conservation services through the 'soilDB' R-Package (Skovlin and Roecker, 2018), and the 'Wosis' databases (Batjes et al.,
2020). Both databases have global coverage and resulted in a total of 68567 usable samples[3] out of 324380 (Fig. 13).

---

[3]At least one of the response variables ($\theta_{fc}$ or $\theta_{wp}$) and all the predictors





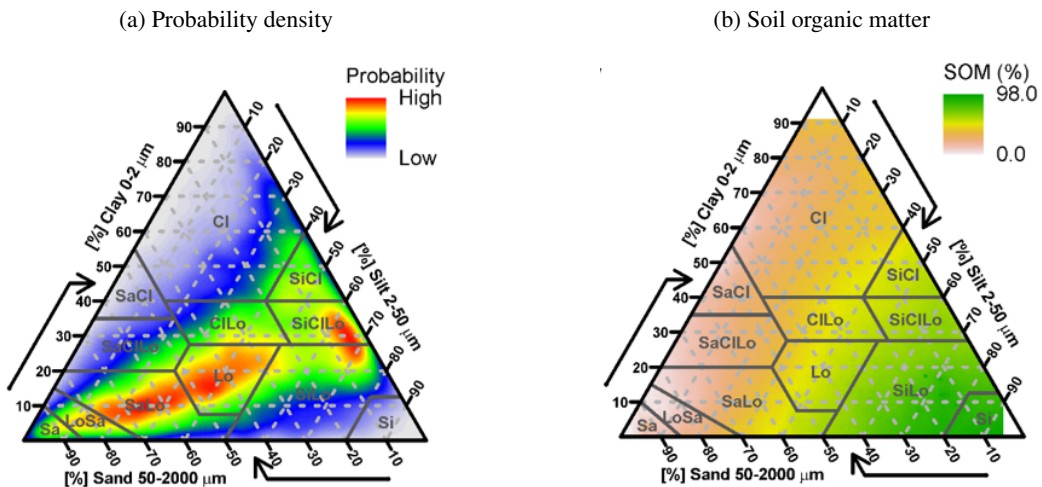

**Figure 13.** $\theta_{fc}$ and $\theta_{wp}$ measurements used to calibrate the pedotransfer functions. (a) Probability density of the soil samples according their textural classes. (b) Average soil organic matter content of the samples per textural class.

While, to optimize the functions to estimate soil moisture at saturation ($\theta_{sat}$) and saturated hydraulic conductivity ($K_{sat}$) data was gathered from the HYBRAS (Ottoni et al., 2018), the SWIG (Rahmati et al., 2018), the UNSODA (Leij et al., 1996) and the Florida University (IFAS, 2007) datasets, for a total of 9346 usable samples out of 15160 (Fig. 14).

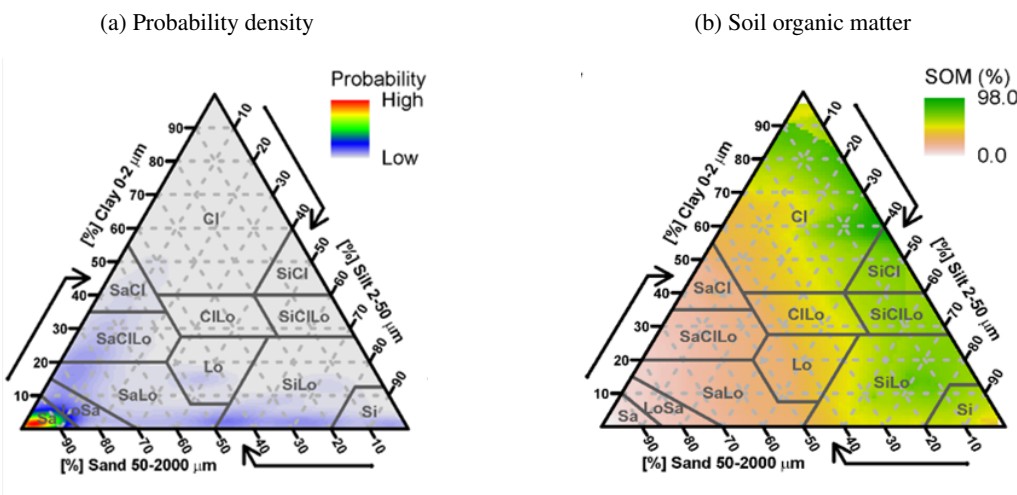

**Figure 14.** $Ksat$ and $\theta_{sat}$ measurements used to calibrate the pedotransfer functions. (a) Probability density of the samples according their textural classes. (b) Average soil organic matter content of the samples per textural class.



To test the model in the $EC$ sites, soil physical properties abovementioned were retrieved from the SoilGrids.org dataset (Hengl et al., 2017), while the 'soilDB' R-Package (Skovlin and Roecker, 2018) was used to retrieve soil data at SNOTEL sites.

Slope, slope orientation (aspect) and upslope area were computed using the TauDEM software (Tarboton, 2016) at the Imperial College high performance computing facility (HPC), from the global SRTM digital elevation model resampled to 250m (Jarvis et al., 2008).

## 3.4   Remote sensing

To calibrate the functions which calculate the snow cover fraction, and the snow cover effect on the albedo, data from the
MODIS MOD10A1 500m-daily product (Hall et al., 2016) was compared against 15 years (2001-2015) of daily data from the 315 SNOTEL stations described previously in this section (Fig. 9).

Information on the biome classification, used to interpret the data, was gathered from the simple typology defined by the International Geosphere-Biosphere Programme (IGBP), available as a MODIS product (MOD12Q1) (Friedl et al., 2019).

To assess the spatial patterns of evapotranspiration in selected small watersheds, the SEBAL algorithm (Bastiaanssen et al.,
1998a, b), implemented in Google Earth Engine (GEE) by Laipelt et al. (2021), was used with Landsat 5 atmospherically corrected surface reflectances from 1994-2007 and Landsat 8 from 2014.

To propose a reasonable assumption for the duration of a precipitation event, the global hourly precipitation GsMAP dataset was used. This dataset has 0.1° of resolution and was built using retrievals from NASA's satellite constellation, including infrared, microwave and radar sensors ( 20 sensors), merged and corrected with NOAA's ground stations (Mega et al., 2014;
Yamamoto and Shige, 2015). Since the gauge count is also available within the dataset, only pixels with 3 or more gauges were used to extract and analyse the hourly data using GEE (Fig. 15).

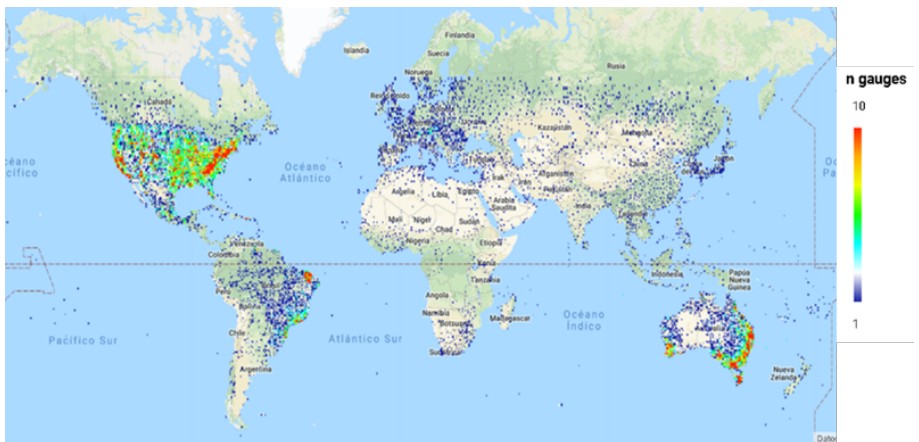

**Figure 15.** Average number of gauges/pixel used to adjust the RS observations. Visualization from GEE
 (© Google Earth Engine).



## 3.5 Spatially distributed forcing

For the global simulations at 5km of resolution, monthly precipitation data from Beck et al. (2019) was resampled and subset to 2010-2016, air temperature was obtained from the Terraclim dataset (Abatzoglou et al., 2018), together with the solar radiation
produced by Ryu et al. (2018), which uses MODIS atmospheric and albedo retrievals as some of the inputs.

For regional simulations (e.g north America), 1km-resolution, temperature and precipitation from CHELSA (Karger et al., 2017) were used, while the solar radiation from Ryu et al. (2018) was downscaled using the theoretical effects of terrain described in Section (2.1.1). Elevation datasets at 1km and 5km resolution used in the respective runs were obtained from Amatulli et al. (2018). While the soil data was resampled from the global 250m SoilGrids dataset (Hengl et al., 2017) (sand,
clay, organic matter, coarse fraction and bulk density). The soil depth/thickness was averaged between SoilGrids and the Pelletier et al. (2016) datasets.

## 4 Methods: Simulation protocol and Performance evaluation

The SPLASH model was run at the individual sites located on mountain ecosystems ( Figs. **??** and 10) with their entire daily time-series of meteorological measurements. Due to the different variables measured by the different networks of monitoring,
the performance evaluation, with the pooled data, was done separately by networks. The statistics used for the evaluation were the determination coefficient ($R^2$), the root mean squared error (RMSE), bias and the slope of the regression observations simulations. To evaluate the seasonal patterns of fluxes/storages, all the results were aggregated as daily means and grouped by climate zone using the Köppen–Geiger climate classification system (Beck et al., 2018). Only direct measurements were used for the performance evaluations, while some indirect observations were estimated, using the variation with previous-day ob-
servations, to visualize seasonal patterns of some fluxes (e.g. $Sf$ and $Sm_e$). To complement the analysis and interpretation of the results, simulations with the 3-layer variable infiltration capacity model (VIC-3L) (Liang et al., 1996; Liang and Xie, 2001) were run using the same inputs and included alongside the results from SPLASH. The vegetation and soil parameters required by the VIC-3L simulations were extracted at the site locations from Schaperow and Li (2020). Extra forcing data required by the VIC-3L, like wind speed and vapour pressure, not measured at the SNOTEL sites, was extracted from the daily, high-
resolution GRIDMET (Abatzoglou, 2013) and DAYMET (Thornton et al., 2020) datasets respectively. Since some quantities computed by SPLASH are not standard outputs of the VIC-3L model (e.g. $H_N^+$ and $C_n$), some calculations were applied to obtain comparable outputs (Appendix A3.1). To compare seasonal patterns of soil moisture a relative moisture content was calculated with the observations and results from SPLASH, in the same way as the VIC output:

$$\Theta = \frac{W_n - W_{pwp}}{W_{sat} - W_{pwp}} \tag{66}$$

Where $W_n, W_{pwp}$ and $W_{sat}$ are the water content $(mm)$, actual, at wilting point and at saturation, respectively.

Since the model lacks a routing algorithm, to test the runoff/lateral flow simulations against streamflow observations, data from micro-catchments was used. While, to visualize major spatial patterns of the fluxes/storages, and test the computational





performance of the model, global simulations were run using monthly spatially distributed datasets at 1km and 5 km of resolution at the Imperial College high-performance computing facility (HPC).

## 4.1 Fitting/Optimization of empirical functions

Parameters from the selected equations (pedotransfer functions) were optimized using the Nash-Sutcliffe (NSE) coefficient, which relates the variance of the residuals with the variance of the data, as the objective function (Nash and Sutcliffe, 1970; Gupta and Kling, 2011). Here, an NSE closer to 1.0 expresses ideal estimates. The probabilistic model for snowfall occurrence was fitted using a binomial family Generalized Linear Model (GLM), while the albedo-related functions were fitted using nonlinear least-squares. To assess the accuracy of the snowfall probability estimation, the Receiver Operating Characteristic (ROC) curve was used, which plots true positive rate[4] (specificity) against false positive rate[5] (sensitivity), using a probability of 0.5 as a threshold. Here the area under the curve (AUC) closer to 1.0 expresses a better overall prediction (Fawcett, 2006).

## 5 Results

## 5.1 Fitting/optimization results

### 5.1.1 Net longwave radiation functions

Quadratic equations were the best fit for both, incoming and outgoing longwave radiation during clear-sky conditions, improving noticeably the predictions for temperatures below $0°C$, particularly useful for regions at high elevations (Figs. 16a and 16c). The net longwave equation, resulting from algebraically subtracting $LW_{IN} - LW_{OUT}$, showed a very small quadratic coefficient, which was neglected to adopt a simpler linear equation (Fig. 16c).

---

[4] ratio $Positives\ correctly\ classified/Total\ positives$

[5] ratio $Negatives\ incorrectly\ classified/Total\ negatives$





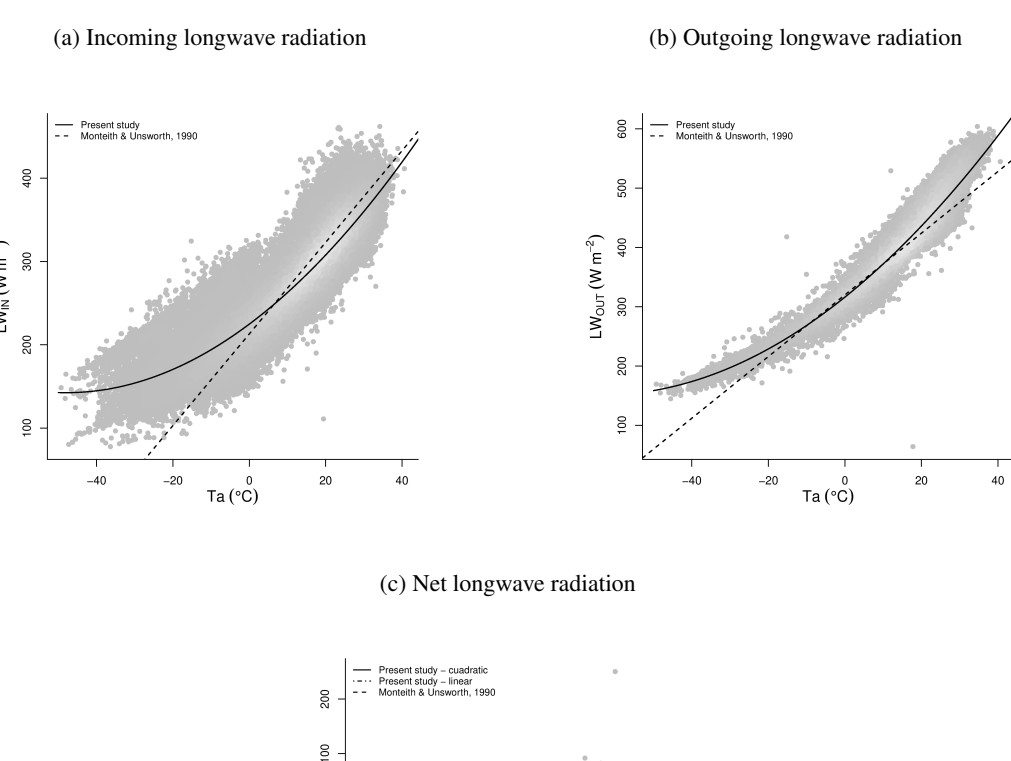

**Figure 16.** Clear-sky longwave radiation as a function of air temperature. (a) Clear-sky incoming longwave radiation. (b) Clear-sky outgoing longwave radiation. (c) Clear-sky Net longwave radiation.

### 5.1.2 Snowfall probability

The performance of the snowfall occurrence calculation resulted in an AUC of 0.97 (17), which considering a maximum value of 1.0, suggests this method is highly accurate.



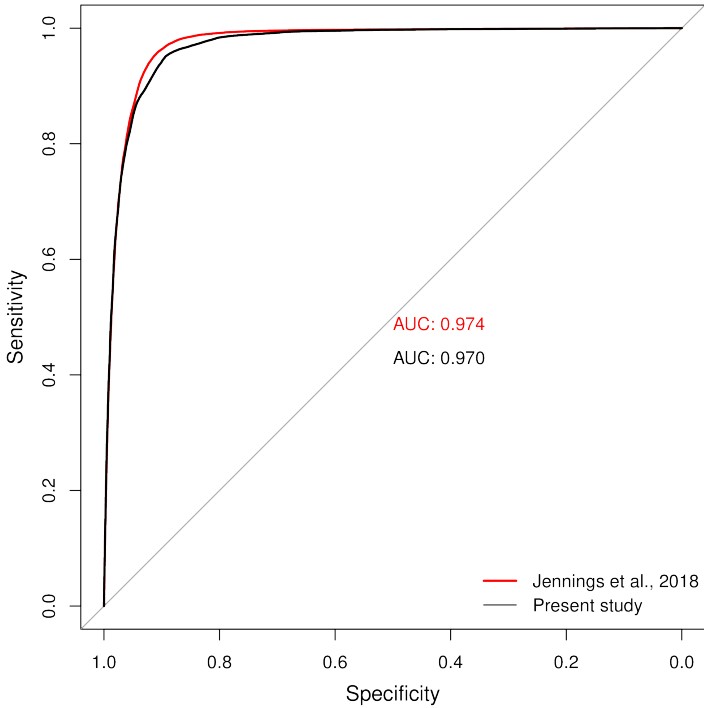

**Figure 17.** Evaluation of the predicted snowfall probability using a ROC curve.

### 5.1.3 Snow cover fraction and Snow-albedo correction

The simple hyperbolic function used to describe the response of the snow cover fraction to the size of the snowpack suggests
that the inflection starts when the $SWE$ reaches 140mm, and most of the variation in snow cover (up to 80%) happens in the
first 1000mm of $SWE$. Moreover, the standard deviations of $SWE$ aggregated by biomes show that in the sampled period
values higher than 1000 mm are uncommon (Fig. 18a).

The snow ageing function suggests that a reduction of about 50% of the albedo can happen in the first 10 days without new
snow falling and the lowest albedo can reach 0.4 (Fig. 33b).



(a) Snow cover fraction

(b) Snow ageing

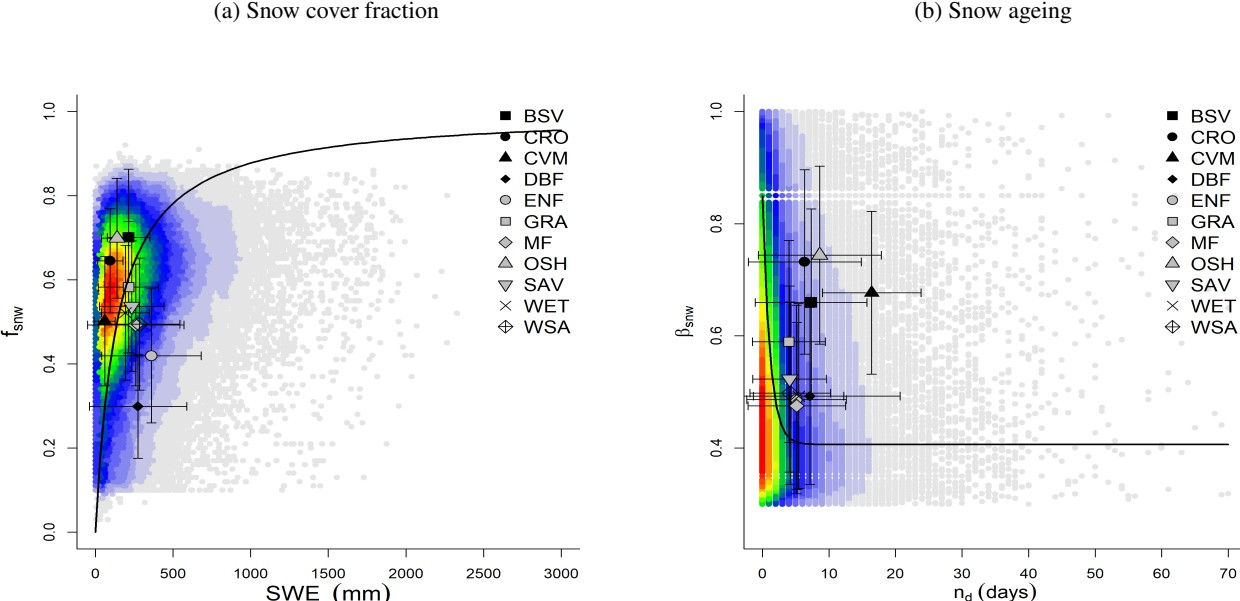

**Figure 18.** Ground-based observations of snow against satellite retrievals. (a) Daily snow cover fraction from MODIS ($f_{snw}$) vs. daily SNOTEL SWE. The black line shows the optimized Eq.(27). (b) MODIS albedo ($\beta_{snw}$) when the snow cover exceeds 70% vs. Days without fresh snowfall ($n_d$) from the daily SNOTEL SWE. The black line shows the optimized Eq.(26).

### 5.1.4 Pedotransfer functions

From the models tested (Table 2), the non-linear equations from Balland et al. (2008), which were fit to the largest dataset outperformed the other models. Further optimization of these equations (Eq. A25a,A25b,A25c) using the full dataset employed for this evaluation yielded a slight improvement of around 10% for field capacity ($\theta_{33}$) and saturation ($\theta_{sat}$) (Fig. 21). The new parameters are detailed in Table A1.

$$515 \quad \theta_{1500} = \theta_{33}\left(c_{wp} + (d_{wp} - c_{wp})\, CLAY^{0.5}\right) \tag{67a}$$

$$\theta_{33} = \theta_{sat}\left(c_{fc} + (d_{fc} - c_{fc})\, CLAY^{0.5}\right) e^{\frac{a_{fc}SAND - b_{fc}SOM}{\theta_{sat}}} \tag{67b}$$

$$\theta_{sat} = 1 - \frac{\rho_b}{\rho_p} \tag{67c}$$

$$K_{sat} = 10^{a_{ks} + b_{ks}\, log_{10}(\rho_p - \rho_b) + c_{ks}SAND} \tag{67d}$$

Were $\theta_{1500}$ is wilting point (water held at 1500kPa), $\theta_{33}$ is field capacity (water held at 33kPa), $\theta_{sat}$ is saturation, $K_{sat}$ the
520 saturated hydraulic conductivity, and, SAND, CLAY and SOM refer to sand, clay and organic matter contents (%). $a, b, c$ are constants with the subscripts referring to wilting point, field capacity or hydraulic saturated conductivity respectively, $\rho_b$ is the





bulk density, and $\rho_p$ is the particle density, calculated as follows (Balland et al., 2008):

$$\rho_p = \frac{1}{\frac{SOM}{1.3} + \frac{1-SOM}{2.65}} \tag{68}$$

**Table 3.** Updated parameters for the Balland et al. (2008) PTFs

|  | $a_x$ | $b_x$ | $c_x$ | $d_x$ |
|---|---|---|---|---|
| $\theta_{1500}$ | – | – | 0.2018 | 0.7809 |
| $\theta_{33}$ | -0.0547 | -0.0010 | 0.4760 | 0.9402 |
| $K_{sat}$ | -2.6539 | 3.0924 | 4.2146 | – |

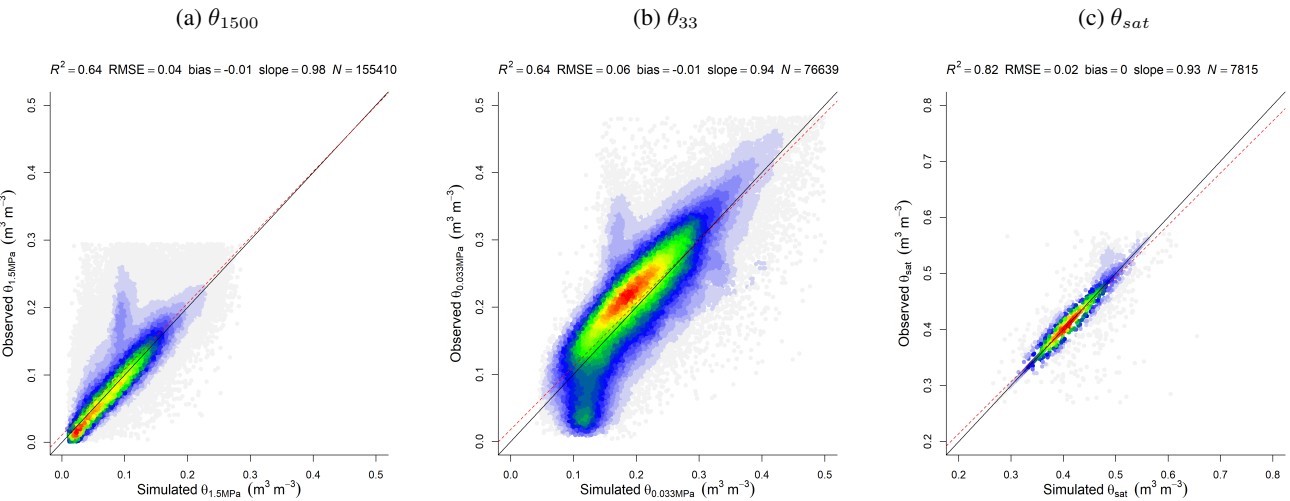

**Figure 19.** Correlation of observed and simulated values of soil hydro-physical properties. (a) Permanent wilting point ($\theta_{1500}$). (b) Field capacity ($\theta_{33}$). (c) Soil porosity or saturation point ($\theta_{sat}$).

$K_{sat}$ estimated by the Saxton and Rawls (2006) PFT was the best performing model, however, it leads to unrealistic values when the drainable porosity ($\theta_{sat} - \theta_{33}$) is relatively high. Thus a simple saturating curve was adopted here, which yields similar estimations to Saxton and Rawls (2006) at the lower end of the drainable porosity but it flattens at a fitted $K_{sat}$ maximum of 623 $mm\,h^{-1}$.





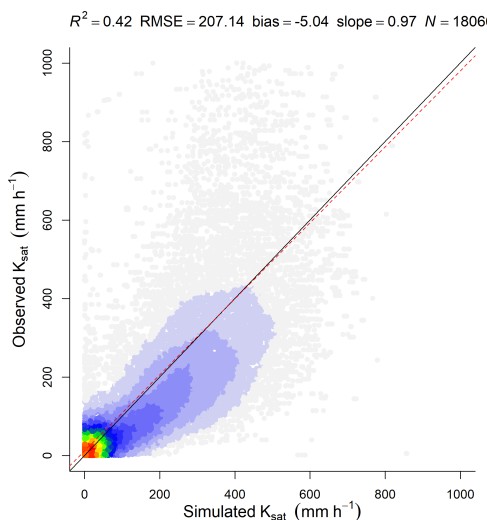

**Figure 20.** Correlation of observed and simulated values of saturated hydraulic conductivity $K_{sat}$.

## 5.2 Fluxes

### 5.2.1 Net longwave radiation

The evaluation of $I_{LW}$ shows the values clustering around -100 and 0 $W\ m^2$, nonetheless the simple linear model was able to explain 70% of the variance with a very low bias ($-0.11$) (Fig. 21).

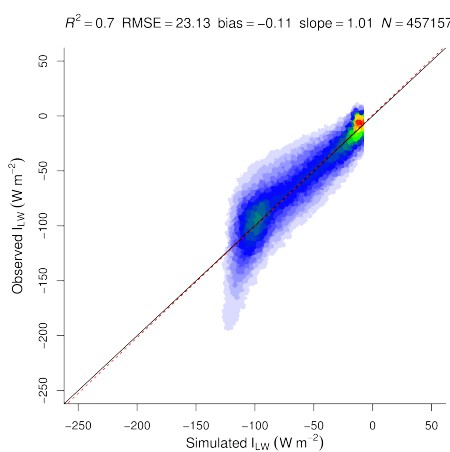

**Figure 21.** Correlation of observed and simulated values of $I_{LW}$, with data from all the sites pooled.





### 5.2.2 Daytime Net Radiation

The Daytime net radiation, or positive net radiation was compared against 87 $EC$ sites distributed over mountain regions covering several biomes. The comparison, using all the data pooled at daily resolution shows that the model is able to explain

more than 70% of the observations' variance with a very small bias (Fig. 22). The evaluation also shows that the highest overestimation happens in ecosystems with sparse vegetation (BSV biome).

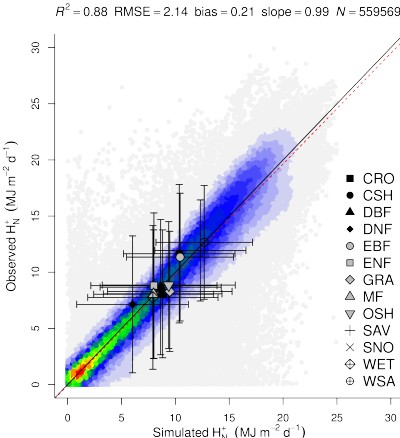

**Figure 22.** Correlation of observed and simulated values of $H_N^+$ with data from all the sites pooled.

The seasonal patterns of $H_N^+$ averaged by climate zone show a classic bell-shaped curve, peaking during the summer months, the greater deviations from the mean appear in climate zones with no dry season (Cfb, Dfc). SPLASH simulations are reproducing the observations more closely than the VIC results in most of the climate zones, noticeably outperforming VIC in

climate zones with dry summer (Csa, Csb). SPLASH overestimation happen primarily in cold deserts during summer months (BWk), while underestimation are noticeable in temperate zones with no dry season (Cfa) and in the hot steppe (BSh), both during summer months as well. (Fig 23).





**Figure 23.** Mean seasonal cycle of $H_N^+$ per climate zone. The gray areas show one SD from the observed mean.

### 5.2.3 Evapotranspiration

The performance evaluation of $E_n^a$ using the data of all the EC stations located in the mountains shows the model reproducing

44% of the variance of daily observations with a small bias and the slope of the regression of observations simulations equal
to 0.9. The standard deviation of the data aggregated by biome, in the observation axis, shows that at daily timestep $E_n^a$ is
highly variable, and there is no clear difference between woody and herbaceous ecosystems. The evaluation shows a greater





underestimation for evergreen broadleaf forests (EBF), while the greater overestimation happens in deciduous broadleaf forests (DBF). Furthermore the lowest $E_n^a$ is shown by ecosystems with sparse vegetation (BSV) and the highest by EBF (Fig. 24).

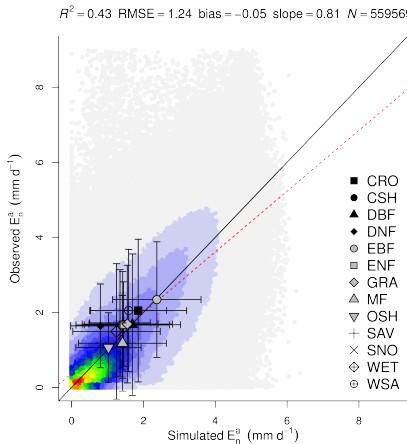

**Figure 24.** Correlation of observed and simulated values of $E_n^a$ with data from all the sites pooled.

The seasonal patterns of $E_n^a$ broadly follow the pattern of $H_N^+$ in zones with no water limitations (Cf* and Cf* types), the polar tundra exhibits similar patterns but with a higher difference between summer and winter months. While arid zones (BS* and BW* types) show a very different pattern compared to $H_N^+$. SPLASH simulations correctly reproduced most of the seasonal patterns, and for certain climate zones (BSk, Csb, Dsc) outperformed VIC simulations. Although SPLASH captures the overall seasonal patterns, it overestimates $E_n^a$ in Dsc sites, while, similarly to the VIC results, it underestimates $E_n^a$ in the 555   polar tundra (ET) and in the Cfb sites in the southern hemisphere (Fig. 25).







**Figure 25.** Mean seasonal cycle of $E_n^a$ per climate zone. The gray areas show one SD from the observed mean.

At a global scale, SPLASH produces major spatial patterns that roughly follow the distribution of the Köppen-Geiger climate zones. In northern latitudes like Alaska it overestimates $E_n^a$ during summer months, nonetheless, it outperforms VIC-3L which is producing high values during winter (Fig. 26).



**Figure 26.** Spatial patterns of mean annual $E^a$ for the period 2010-2016 at 5km resolution along with site-simulation examples from the mountains. Inset plots show seasonal cycles where observations are in black, SPLASH v2.0 simulations are in red and VIC 3L simulations are in blue. The grey areas show one SD from the observed mean.



The simulations, with the long-term data from Rietholzbach, exhibit a good overall agreement with the lysimeter-based
observations (Fig. 27e). However, a minor but systematic overestimation happens during the spring months. Here, the spatial
patterns produced by SPLASH show higher magnitudes on south-facing slopes, and, at the valley bottom close to the outlet,
coinciding with the area where a small forest is present (Seneviratne et al., 2012) (Fig. 43c). These patterns contrast with the
spatially distributed LE, calculated from Landsat 5 retrievals, which shows a more uniform LE, except for a few forest patches,
where LE spikes (Fig. 27d). Both datasets barely agree over the small forest at the valley bottom.



(a) Slope

(b) Aspect

(c) SPLASH annual Evapotranspiration

(d) SEBAL/Landsat 5 mean LE

(e) Monthly evapotranspiration time series

**Figure 27.** Spatial and temporal patterns of evapotranspiration in a small wet temperate watershed, Rietholzbach - Switzerland. (a) Slope in degrees. (b) Slope orientation, N stands for north, S for south and so on. (c) Mean annual simulated evapotranspiration 1994-2007. (d) Mean instantaneous LE from L5's clear-sky pixels during 1994-2007. (e) Time series of monthly evapotranspiration, simulated and lysimeter-based observations.

In the tropical watershed, the spatial patterns of $E_n^a$ produced by SPLASH (Fig. 28c) show a better agreement with the RS LE than in the temperate watershed, as shown by some emergent coldspots in the northern part of the watershed (Fig. 28e).





Moreover, in both datasets, slopes facing the equator show higher magnitudes compared to flat areas, however, in the SPLASH results, this difference is stronger than in the RS LE estimation.





(a) Slope

(b) Aspect

(c) Mean SPLASH $E_n^a$

(d) SD of SPLASH $E_n^a$

(e) Mean SEBAL/Landsat8 LE

(f) SD SEBAL/Landsat8 LE

**Figure 28.** Spatial patterns of daily evapotranspiration in a wet tropical watershed, Jatunhuayco - Ecuador. (a) Slope in degrees. (b) Slope orientation, N stands for north, S for south and so on. (c) Mean daily evapotranspiration during 2014. (d) Standard deviation of the daily evapotranspiration during 2014). (e) Mean instantaneous LE from L8's clear-sky pixels during 2014. (f) Standard deviation of the instantaneous LE from L8's clear-sky pixels during 2014.



### 5.2.4 Condensation

SPLASH model showed the poorest performance simulating $C_n$, it was able to capture around 10% of the variance, with a bias of -0.621. The slope of the regression observations-simulations points to overestimations at high simulated values. Most of the observations seem to cluster in the 0-10 $mm\,yr^{-1}$, the greater underestimation happens in broadleaf evergreen forests while $C_n$ is overestimated in barren/sparse vegetated ecosystems 29).

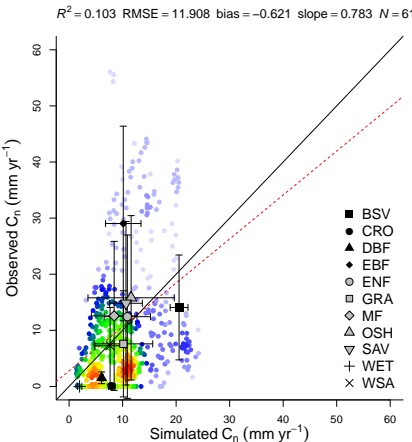

**Figure 29.** Correlation of observed and simulated values of $C_n$ with data from all the sites pooled.

Just arid climate zones (B* types) showed seasonal patterns of $C_n$, and potentially important magnitudes, both dimensions
captured by SPLASH. However, the SPLASH model is still underestimating $C_n$ in the hot steppe (BSh) and overestimating $C_n$ in the cold steppe and desert (e.g. BSk and BWk). In some climate zones with no apparent seasonal pattern and with random peaks through the year (e.g. Cfc and Dsc) SPLASH model shows a smother prediction, underestimating all the peaks. Compared to the VIC simulations, the SPLASH model seems to reproduce the general patterns more reasonably (Fig. 30).



**Figure 30.** Mean seasonal cycle of $C_n$ per climate zone. The gray areas show one SD from the observed mean.

### 5.2.5 Snowfall

The seasonal patterns of $Sf$ show differences in the magnitudes between climate zones with and without dry season only in temperate types (Cfb, Csb), while wet continental (Df*) types did not show noticeable differences with their dry counterparts (Ds*). Arid climates (B*), on the other hand, showed the lowest magnitudes. The patterns simulated by SPLASH match almost perfectly with the observations for all the climate zones, in agreement with the VIC simulations as well. Some underestimation is noticeable at the end of the winter in the wet temperate zone (Cfb) (Fig. 31).







**Figure 31.** Mean seasonal cycle of $Sf$ per climate zone. The gray areas show one SD from the observed mean.

### 5.2.6 Snowmelt

Seasonal patterns of $Sm$ appear more clearly in the continental climate zones (D* types) with an expected peak at the beginning of spring, the other climate zones apparently don't show a general pattern. SPLASH model was able to simulate the start of the melting process in all the climate zones, however, it captures the seasonal pattern only in wet continental climates (Df*), while overestimates $Sm$ in their dry counterparts (Ds*). Overall the seasonal patterns from SPLASH seem to agree with the



simulations better than the results of the VIC model, which shows a temporal lag in the start and peak of the melting period (Fig. 32).

**Figure 32.** Mean seasonal cycle of $Sm$ per climate zone. The gray areas show one SD from the observed mean.





### 5.2.7 Surface Runoff and lateral flow

The SPLASH simulations of total streamflow (TSF) ($RO + q_{out}$) in the watersheds were able to explain 71% of the variation while the estimations of surface runoff accounted for 69% of the variation. The bias in both analyses shows a systematic underestimation of this flux, especially at the lower end (Fig. 33a).

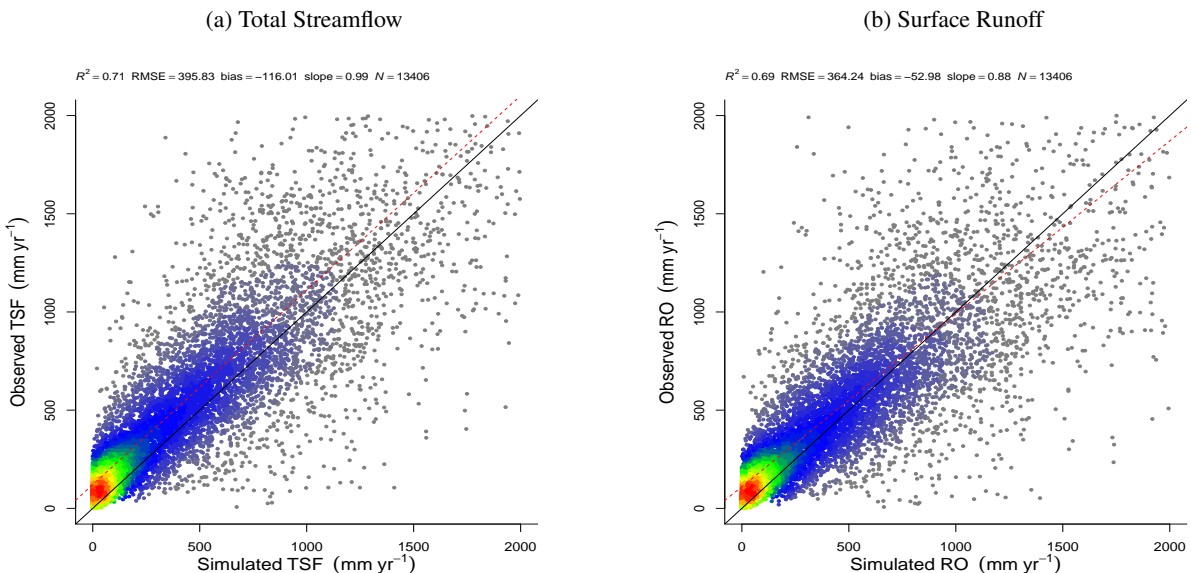

(a) Total Streamflow

(b) Surface Runoff

$R^2 = 0.71$ RMSE = 395.83 bias = −116.01 slope = 0.99 $N = 13406$

$R^2 = 0.69$ RMSE = 364.24 bias = −52.98 slope = 0.88 $N = 13406$

**Figure 33.** Correlation of observed and simulated values of Discharge with data from all the watersheds pooled. (a) Total Streamflow. (b) Surface Runoff

Tracking down the source of the systematic underestimation by analysing in detail the time series of observed and simulated TSF from Rietholzbach, the underestimation appeared greater during winter months when the precipitation is below the average. On the other hand, when the precipitation is peaking, overestimation occurs in some years seemingly without showing any systematic pattern (Fig. 34f).

The simulated time series of baseflow index BFI $\left(\frac{q_{out}}{q_{out}+RO}\right)$ roughly follows the temporal dynamics of the observations. Overestimations appear in the winter months due to the underestimations of surface RO during these months (Fig. 34e). Furthermore, the spatial patterns resulting from the simulation show the runoff is higher in areas surrounding the stream, which emerges from the flux accumulation at the valley bottom (Fig. 34b).

Moreover, the lateral flow is mostly produced in the north-facing slopes and in some areas next to the main stream, upslope from the main outlet. The BFI is close to 1 in most of the watershed, except in areas close to the stream, suggesting that most of the simulated hydrological response is subsurface flow (Fig. 34a).

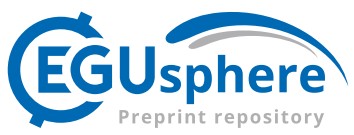

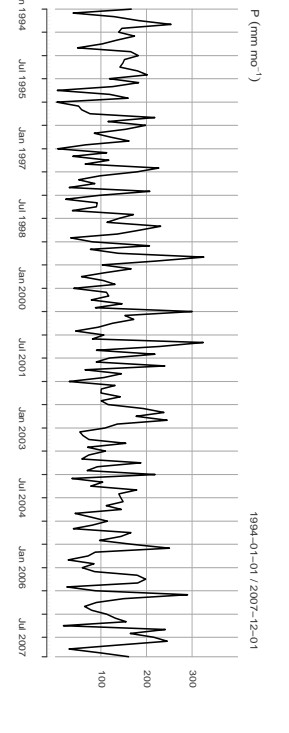

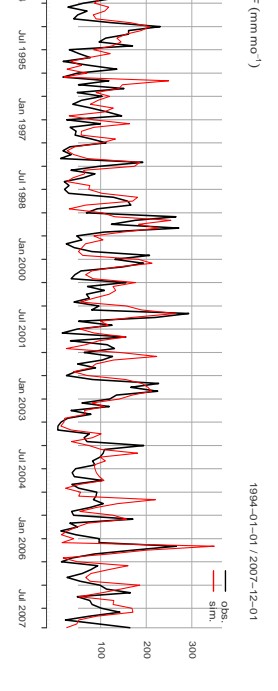

**Figure 34.** Spatial and temporal patterns of runoff in a small wet temperate watershed, Rietholzbach - Switzerland. (a) Spatial patterns of mean baseflow index in the first 2m during 1994-2007. (b) Spatial patterns of mean annual runoff during 1994-2007. (c) Spatial patterns of mean annual baseflow during 1994-2007. (d) Time series of precipitation 1994-2007. (e) Time series of SWC in the first 1m of depth during 1994-2007. (f) Time series of total streamflow during 1994-2007





The monthly means of BFI from the long-term data from Rietholzbach suggest that the underestimation of surface $RO$ is indeed systematic and the major discrepancies appear in November and December (Fig. 35).

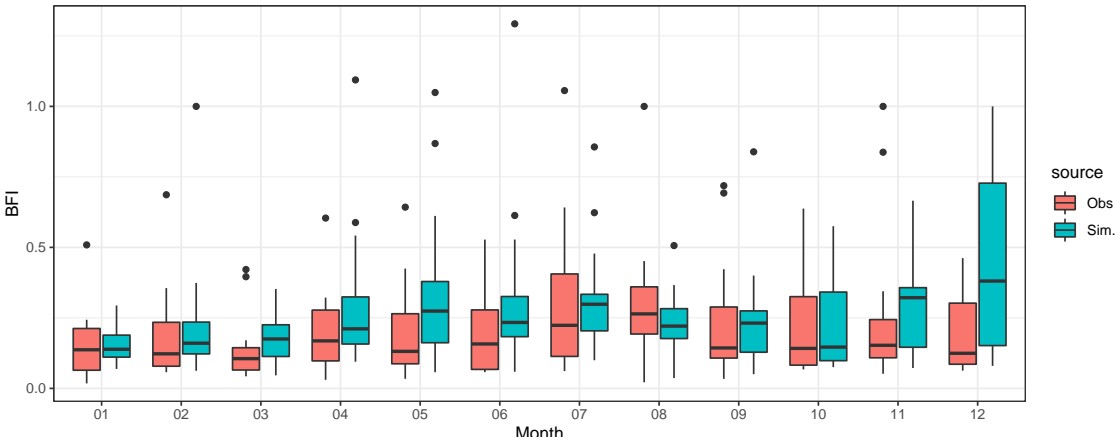

**Figure 35.** Mean monthly BFI 1994-2007 in a small wet temperate watershed, Rietholzbach - Switzerland for the period 1994-2007

At daily timestep, the results of the simulation in the tropical watershed, show that the performance of SPLASH simulating
fast storm response is poor, the model fails in capturing the expected runoff peaks during relatively big storms at daily timescale (Fig. 36e). Nonetheless, the spatial patterns generated by SPLASH in this small watershed, reproduce mostly saturation excess runoff in areas close to the streams (Fig. 47a), while the simulated lateral flow appear stable, spatially and over time (Figs. (47c) and (47d)).



(a) Mean daily runoff

(b) SD daily runoff

(c) Mean daily lateral flow

(d) SD daily lateral flow

(e) Time series of daily precipitation and total streamflow

**Figure 36.** Spatial and temporal patterns of daily fluxes in a tropical small watershed, Jatunhuayco - Ecuador (a) Mean daily runoff during 2014. (b) Standard deviation of the daily runoff in 2014). (c) Mean daily lateral flow during 2014. (d) Standard deviation of the daily lateral flow during 2014 (e) Time series of daily precipitation and total streamflow during 2014



## 5.3 Storages

### 5.3.1 Snow water equivalent

The simple formulations proposed to simulate $SWE$ were able to explain 88% of the observed variation 127 SNOTEL sites located in mountain regions, the mean value of the observations aggregated by biome, show the $SWE$ storage in evergreen needle-leaf forests (ENF) is the largest among the biomes, while the lowest value was from open/deciduous canopies (OSH,DBF). The bias suggests that SPLASH is underestimating $SWE$ at low values, however, the huge variation of the observations in all the biomes suggest the differences are not significant (Fig. 37).

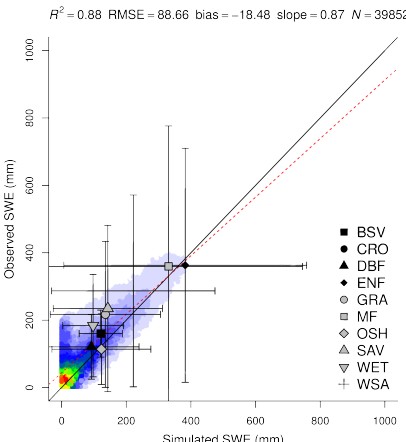

**Figure 37.** Correlation of observed and simulated values of daily $SWE$, values of all sites pooled

The seasonal patterns of $SWE$ depict the well-known bell-shaped curve during winter with great deviations from the seasonal mean. Here, the sites in the temperate climate zones without dry season showed the greatest deviation (Cfb). SPLASH model was able to reproduce the seasonal patterns in all the climate zones. It underestimates, to different degrees, the averages within the range of the observations, which contrasts with the overestimation of the VIC simulations. Nonetheless, SPLASH

captures almost perfectly the length of the snow-covered period, failing only in the Cfb zone, where it overestimates $SWE$ during the melting period, here VIC outperforms SPLASH (Fig. 38).







**Figure 38.** Mean seasonal cycle of $SWE$ per climate zone. The gray areas show one SD from the observed mean.

The spatially distributed high-resolution simulation, over North America, showed strong patterns defined by the topography, with higher $SWE$ over the mountain regions, and variations according to the slope exposure. A well-defined lower boundary for the snow-covered area emerged in the eastern US, around $40°$ of latitude, coinciding with the transition from climates Cfa to Dfa, while on mountain summits the simulated $SWE$ fades at around $35°$ (Fig. 39).





**Figure 39.** Spatial patterns of mean monthly maximum $SWE$ for the period 2010-2016 over North-America at 1km resolution, along with site-simulation examples from the mountains. The gray areas show one SD from the observed mean.





### 5.3.2 Soil water content

The performance evaluation of daily $SWC$ was done with data from stations measuring soil moisture deeper than 30 cm located in the mountain regions, resulting in 16 EC (Fig. 40a) and 127 SNOTEL (Fig. 40b) stations. Comparison among biomes was not possible due to the different depths of measurement in each station, this shows for example, because of mostly superficial measurements, wetlands at the lower end of the observations axis (Fig. 40b). Nevertheless, the SPLASH model was able to explain 86% of the variation in the EC dataset which is around 7 times smaller than the SNOTEL dataset where SPLASH explained 54% of the variation. The evaluation also shows that $SWC$ was overestimated by the SPLASH model in Grassland ecosystems (GRA) and in open shrublands (OSH) (Fig. 40).

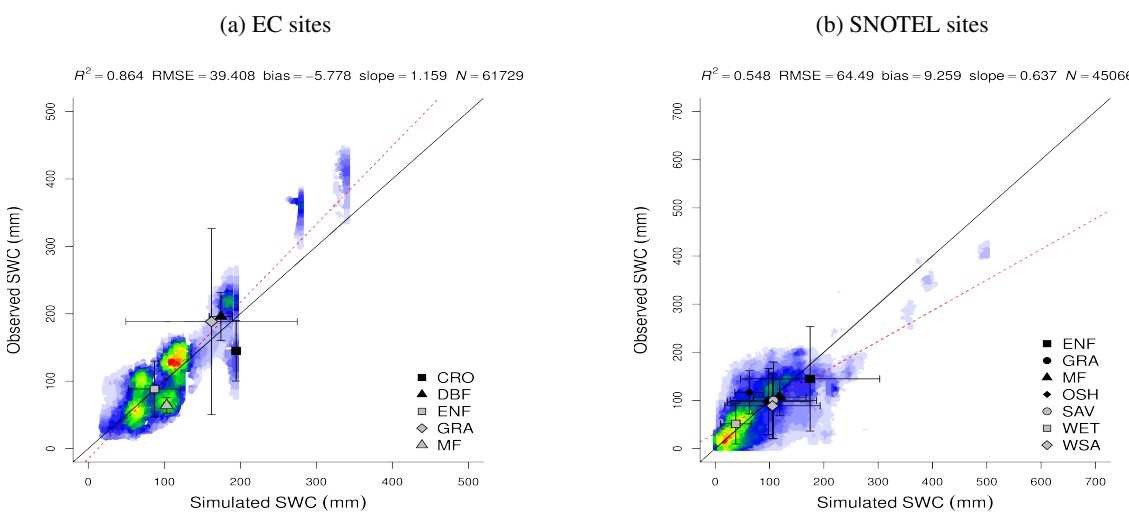

**Figure 40.** Correlation of observed and simulated values of daily $SWC$, values of all sites pooled.

The seasonal patterns of $\Theta$, extracted from the data and simulations at SNOTEL sites, show a sinusoidal shape in all the climate zones. The highest point appears during the first half of the spring, and the lowest point by the end of the summer. Most of the climate zones show huge deviations from the mean, especially the zones without dry season (e.g. Cfb, Dfc), except for the cold desert (BWk) where this variation is smaller.

SPLASH model shows a reasonable agreement of the seasonal pattern for climate zones with warm and dry summers (i.e. Csb, Dsb) and for the cold steppe (BSk). At these climate zones, SPLASH reproduces the seasonality in a better way than VIC. In some continental climates (Dfb, Dfc, Dsc), SPLASH produce a similar pattern to the observed mean but downshifted inside the expected deviation. At these climate zones, the VIC model approaches the observed mean only during spring-summer months. In the Warm-summer temperate climate with no dry season (Cfb), SPLASH is not able to reproduce the seasonal pattern, the amplitude of the sinusoidal shape is too small to match the observations. Here, VIC agrees with the observations briefly during winter. In the cold desert (BWk) SPLASH does not produce a seasonal pattern, the average through





the year remains close to the minimum. Here VIC overestimates $\Theta$ during spring and produces negative values during the spring and summer months (Fig. 41).



**Figure 41.** Mean seasonal cycle of $\Theta$ per climate zone. The gray areas show one SD from the observed mean.

The spatially distributed simulation of relative soil moisture saturation shows that the patterns of the annual average roughly follow the global climate zones distribution. At this resolution, the model was incapable of reproducing the streams and most of the mountain regions emerge as areas half to low saturated (Fig. 42).





**Figure 42.** Spatial patterns of mean annual Θ for the period 2010-2016 at 5km resolution along with site-simulation examples from the mountains. The grey areas show one SD from the observed mean.



The spatial patterns of the total soil water content in the temperate watershed show an emergent accumulation at the valley bottom, shaping the stream. Here, the s.d. is very small, suggesting the valley bottom remains wet most of the year (Fig. 43d).

The different patterns over north and south-facing slopes, show the former with more water content over the latter, but more variations over time. The emergent variations in the north-facing slope are shaped by the inclination of the slope, while on the south-facing slope, this pattern seems to follow the aspect (Fig. 43c).

The time series of soil water content over the first meter of depth were poorly reproduced by SPLASH, in some years the simulation matches the observations (e.g. 2006), however, in most of the years the droughts are underestimated and the peaks are overestimated (Fig. 43e).



(a) Slope

(b) Aspect

(c) SPLASH mean $W_n$

(d) SPLASH SD $W_n$

(e) Monthly swc 1m time series

**Figure 43.** Spatial and temporal patterns of soil water content in a small wet temperate watershed, Rietholzbach - Switzerland. (a) Slope in degrees. (b) Slope orientation, N stands for north, S for south and so on. (c) Mean annual simulated soil water content in the whole column during 1994-2007. (d) The standard deviation of the daily soil water content in the whole column during 1994-2007. (e) Time series of monthly soil water content, simulated and observed over the first 1 m. of depth.

In the tropical watershed, the spatial patterns of the daily average $W_n$ show the emergent stream at the valley bottom, according to the simulations the daily water content here is more variable than in the rest of the watershed (Fig. 44d), contrary





to the temperate watershed where this pattern was opposite. Here the east-west aspects are defining the spatial patterns, the eastern flank show less water content than their Western counterparts (Fig. 44c).

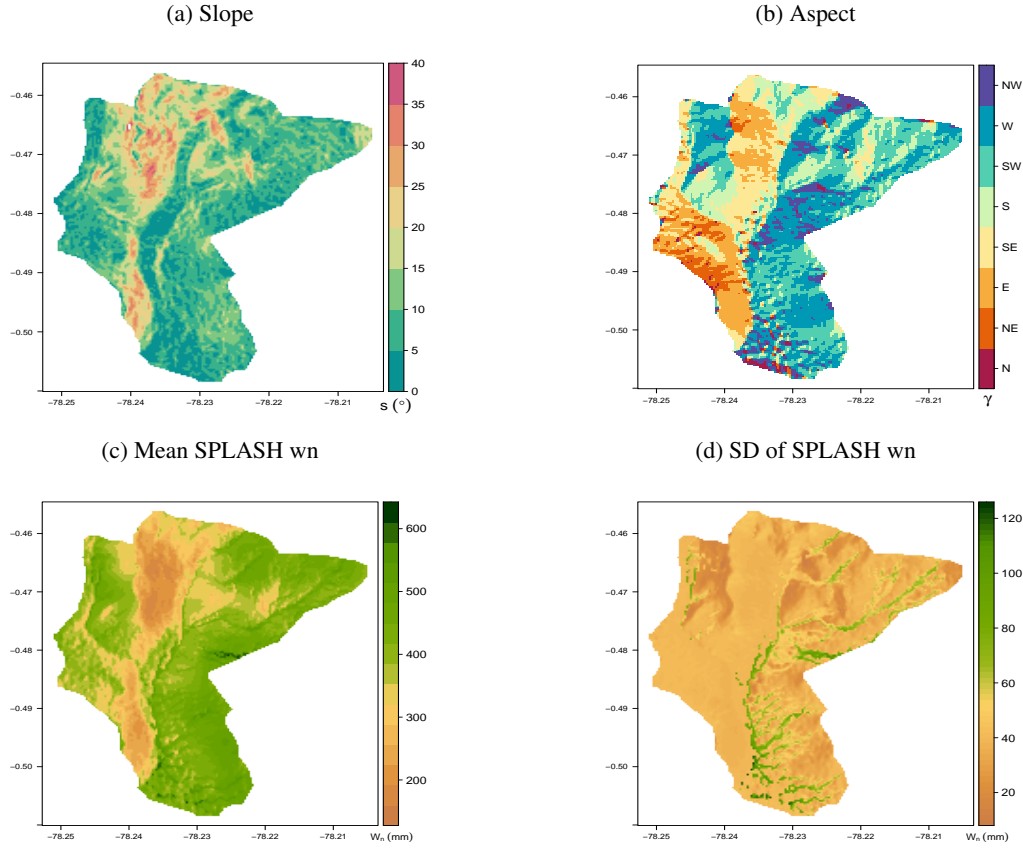

**Figure 44.** Spatial patterns of daily soil water content in a wet tropical watershed, Jatunhuayco - Ecuador. (a) Slope in degrees. (b) Slope orientation, N stands for north, S for south and so on. (c) Mean daily soil water content during 2014. (d) Standard deviation of the daily soil water content during 2014).

## 6 Discussion

The updated SPLASH model exhibited a reasonable agreement with the observations in all the fluxes analysed here without any local calibration or prescribed land-cover information. The data requirements to run the model (precipitation, solar radiation,

air temperature, elevation and soil texture) are modest; the open-source code, compiled in a ready-to-use package, facilitates replication.

The analytical approach used to solve the energy and water fluxes allowed the model to run with high-resolution data on global scales without using statistical–dynamical flux parameterizations, or hydrological unit responses. Emergent patterns



produced by the model follow the natural accumulation of soil moisture downslope. Most of the fluxes and storages analysed
here agree with — and sometimes outperform— the more complex VIC-3L model, which was chosen for comparison due to
its wide use in ecohydrological applications, and its well-known good performance.

SPLASH assumes background albedo, a parameter particularly crucial due to its synergy with water fluxes, to be constant for
all biomes. This contrasts with the VIC model, which uses monthly albedo per vegetation class (Gao et al., 2009). Nonetheless,
SPLASH showed overall good agreement of $H_n^+$ with the observations, suggesting that the snow effect on the albedo is much
stronger than the effects of the phenology in snow-covered regions (Xiao et al., 2017). This global albedo assumption, however,
is more likely to be the cause of the discrepancy of $H_n^+$ in ecosystems with sparse vegetation cover (e.g. BSV and OSH), where
the extent of exposed soil and its moisture status modify the albedo (Campbell and Norman, 1998; Barry, 2008).

Although the ground heat flux is ignored by the model, an improvement in the calculation of $H_n^+$ is noticed relative to the
previous version (Davis et al., 2017), where the overestimation in the arid desert (BWh) is fixed with the new parameterization
of the longwave radiation, and the overestimation in the polar tundra (ET) is corrected by the new included feedback snow-
albedo.

Simulated actual evapotranspiration ($E_n^a$) is underestimated to various degrees in all biomes. As model performance for
$H_n^+$ is high, this suggests that the $E_n^a$ discrepancies are related to the empirical parameterization of the water supply/uptake
($S_W$). Theoretically, this should be driven by the soil-to-leaf water potentials gradient ($\Delta\psi$) (Prentice et al., 2014), thus,
reflecting different plant strategies to deal with drought. However, when this idea was tested during the development stage, the
performance of the simulations decreased (Fig. 45c), probably due to the calculation method used for the leaf-water potential,
and its assumptions (both taken from the literature): canopy well coupled to the atmosphere at pre-dawn, thus $T_s = Ta$; and
the relative water content of the leaf close to saturation (Appendix A4).

Although in this version of SPLASH, I propose a physically-based calculation for the upper threshold of $S_W$, its response
to the water deficit is conceptualized as a linear function, which has been reported as the most simple and reasonable empirical
description (e.g. Federer (1982)). Nonetheless, several authors report more complex formulations depicting, convex (Campbell
and Norman, 1998), concave (Metselaar and de Jong van Lier, 2007) or trapezoidal (Feddes and Raats, 2004) shapes. Some
of these formulations were tested during the development stage of the model, with no significant improvement over the simple
linear formulation (Fig. 45).

From this experimentation, the assumption made on the maximum supply rate $S_c$ as the maximum rate of evaporation
yielded the best approximations. This assumption makes more sense in bare-ground areas, however, in vegetated areas, this
value should be reflecting the plant controls on transpiration, which ideally, needs to be addressed with ideas based on Eco-
evolutionary optimality theory (Harrison et al., 2021).





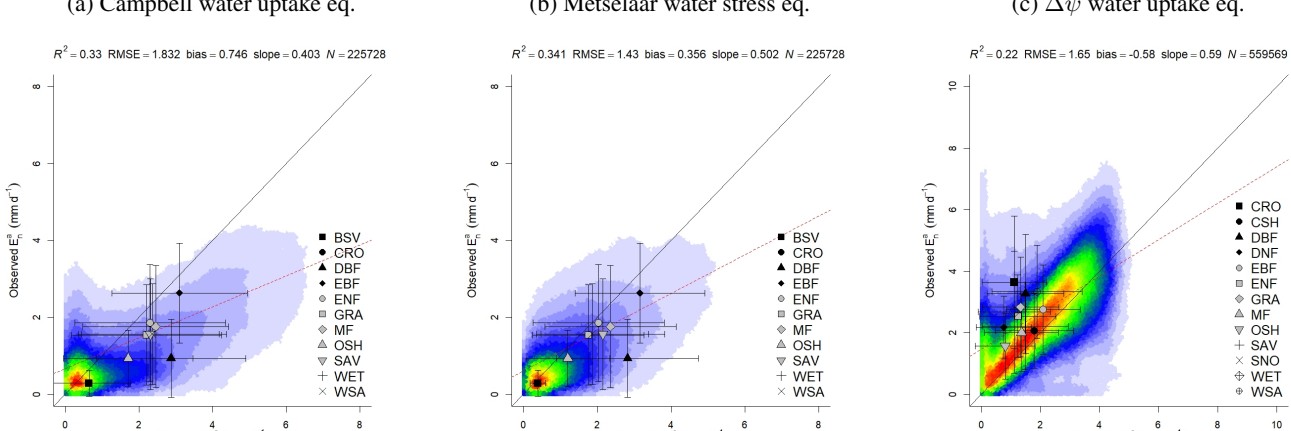

**Figure 45.** Simulation experiments against EC observations using different soil water stress/water uptake functions. (a) Using Campbell and Norman (1998) water uptake function in mountain sites. (b) Using Metselaar and de Jong van Lier (2007) water stress function and a constant $Sc = 1.05 \, (mm \, h^{-1})$ (Federer, 1982) in mountain sites. (c) Using the gradient of water potentials soil-leaf $\Delta\psi$ to drive the water uptake all FLUXNET database.

Since SPLASH seems to reproduce the evapotranspiration better over non-water limited areas, in such areas, slopes facing
the equator (south-facing slopes in the northern hemisphere) should, in theory, show higher values than their opposite-facing counterparts, which receive less radiation (Körner, 2021; Chapin et al., 2011).

This spatial pattern is indeed produced by SPLASH in the Rietholzbach experimental catchment. However, here the latent heat calculated from the Landsat5 retrievals does not show any strong differences between north and south-facing slopes. It is still unclear if the spatial patterns from Landsat are correct, the SEBAL algorithm used the calculate LE is limited to
710 clear-sky pixels only, thus a large amount of data was excluded from the calculation, furthermore, this algorithm computes an instantaneous $\frac{LE}{Rn-G}$ (at the satellite overpass), then, it assumes this proportion is constant through the day, so the daily $E^a$ can be calculated from the daily accumulated $Rn$ measured on the ground (Bastiaanssen et al., 1998a). Therefore, a more accurate estimation from SEBAL would involve terrain-corrected independent calculations of $Rn$ at Landsat spatial resolution, which were unavailable at the time of this comparison. Land use in Rietholzbach also plays a key role in shaping the spatial patterns
of LE as well.

SPLASH in theory reflects the environment the plants experience. The spin-up routine produces an initial state of equilibrium, thus in areas with natural vegetation the spatial patterns produced by SPLASH should reflect this vegetation cover to some degree. This is shown in the agreement (to some extent) of the spatial patterns of $E_n^a$ produced by SPLASH compared with the Landsat8 LE in the tropical watershed, which always had natural vegetation. This microclimatic gradient created by
720 the slope and aspect is particularly important to explain outlier populations existing beyond their major distribution zone, which can colonize their surroundings during rapid climatic changes (Chapin et al., 2011).

Although the results presented here are encouraging, more rigorous comparisons are needed to evaluate how well the fluxes produced by SPLASH reflect patterns of naturally occurring vegetation.





The less-than-optimal performance of SPLASH simulating condensation is mainly due to the lack of other environmental variables needed to calculate the dew point and surface temperature, such as air humidity, wind speed, and aerodynamic resistance. Nonetheless, the simple assumption made to estimate this flux (10% of $H_N^-$) reproduces the seasonal $Cn$ better than VIC. The major discrepancies of VIC's $Cn$ happen during the spring-summer months suggesting that some of the heat lost as $Cn$ (latent) is actually lost from the surface by convection, cooling the leaves. The yearly magnitudes of dew formed by $Cn$ suggest that its impact on the water balance is minimum in most climate zones, except for hot arid climate zones (BSh), where $Cn$ has ecological importance, in agreement with the observations reported by Guo et al. (2016) and Yu et al. (2020).

The size of the snowpack ($SWE$) simulated by SPLASH agrees more than 80% with the observations, however, its seasonal patterns show a systematic underestimation in most of the climate zones. Since the seasonal patterns of the snowfall ($Sf$) produced by SPLASH match the observations in all the climate zones, and, the snowmelt ($Sm$) is reasonably well predicted in the steppe (BSk) and the wet continental climates (Df*). In these climate zones, the discrepancies seem to be due to re-distribution of snow by the wind, which is greatly dependent on the structure of the vegetation (Barry, 2008; Pomeroy and Brun, 2001) and it is not considered by the model. In dry continental and temperate climates (Ds*, Csb), on the other hand, SPLASH systematically underestimates $SWE$, here the cause is more likely to be neglecting the "cold content" of the snow, which in turn causes an overestimation of $Sm$. This effect is stronger at high elevations where the temperature is lower, here VIC delivers better estimations than SPLASH (Fig. 45). Moreover, although the discrepancies in the simulated magnitudes at these sites, the duration of the snow-covered period is reasonably predicted, considering that the multi-annual variation can be up to one month (Körner, 2021).



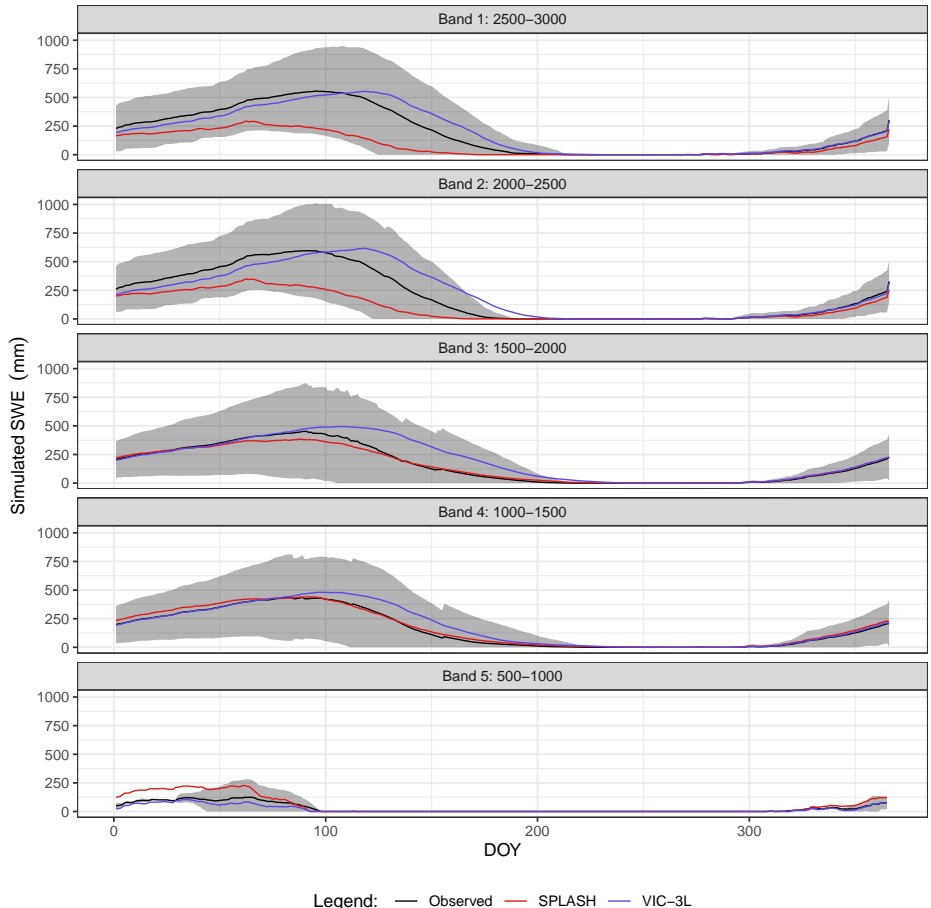

**Figure 46.** Mean seasonal cycle of $SWE$ over dry temperate climates (Csb) per elevation band. Results aggregated every 500 masl. The lines show the means, while the gray shaded area the mean $\pm$ sd.

Although in theory, different sets of parameters in the snow-albedo functions, depending on the complexity of the canopy, should yield more accurate predictions (e.g. Romanov (2003) ), the performance of SPLASH simulating $SWE$ show that one set of parameters is able to deliver reasonable approximations for all types of biomes, such approximation was possible using roughly $10^6$ of RS/ground datapoints spawned over 15 years in 315 stations during the optimization process. The use of one set of parameters is nothing new and it has been used for a long time, however, in previous studies, the number of observations and available tools were limiting the spatial representativity of these calibrations (Clark et al., 2017). Nonetheless, the parameters found in SPLASH for the snow-albedo decay function, are consistent with the values proposed in the Noah model by Livneh et al. (2010), although in SPLASH there are no different sets of parameters for accumulation and melt seasons.

The spatial patterns of the surface runoff generated by SPLASH in the small watersheds show that most of this flux is generated in the saturated zone in the valley bottom, in agreement with what is expected in montane regions (Grayson and Blöschl, 2000; Weizu and Freer, 1995).



Moreover, the spatial patterns of BFI produced by SPLASH in Rietholzbach agree with previous studies which show that most of the streamflow is produced by subsurface flow, specially interflow (Von Freyberg et al., 2014; Gurtz et al., 2003).

However, the poor performance of SPLASH simulating the daily surface runoff in the Andean catchment, suggest that the assumption made for the event duration doesn't hold for this area, or the saturated zone close to the stream, which controls this flux in Andean catchments (Correa et al., 2020) is underestimated.

Although the simulations of soil moisture are overall reasonable and the results mostly matched the temporal dynamics of the observations, two recurrent errors were observed among the simulations, or a combination of these errors, the resultant soil

moisture appears up or downshifted randomly (1st error), or the amplitude of the variations is different from what is observed (2nd error).

The first type of error seems to be related to the estimation of the bucket size, which in turn is defined by the pedotransfer functions and soil data. Although the pedotransfer functions were optimized with a global dataset, which covers a wide spectrum of textural classes with a wide range of SOM combinations, the empirical nature of these equations is a well-documented

source of error (Pachepsky et al., 2015; Van Looy et al., 2017; Paschalis et al., 2022). This error might explain why the evapotranspiration and total streamflow agree with the observations in Rietholzbach, but the performance of the soil moisture simulation was very poor.

The second type of error was less recurrent with the SNOTEL dataset, which includes actual measurements of soil properties, suggesting that the data obtained from SoilGrids for the EC sites might be a source of this error. The coarser volumetric fraction

(stoniness), particularly, can reduce the size of the bucket dramatically here. However, a rigorous evaluation and sensitivity analysis is needed to define this source of error in a broader modelling context.

The spatial patterns of soil moisture at a global scale show how the assumption of the model of a maximum depth breaks the hydrological connectivity in large watersheds. Here, the model forces the water to flow down instead of laterally if the bedrock is not in the first two meters of depth. On the other hand, in small watersheds, the hydrological connectivity emerges from the

775 conceptualization of the model accumulating the moisture at the valley bottom and shaping the streams.

Furthermore, the lateral flow simulated by SPLASH is strongly related with the soil water content, consistent with what has been reported in Rietholzbach (Teuling et al., 2010) and in Andean watersheds (Crespo et al., 2011).

The exponential decay of the saturated conductivity with depth, widely used in TOPMODEL-type models, was deliberately excluded from the model. The saturated hydraulic conductivity is highly dependent on the organic matter content and was

780 calculated using a weighted average (by depth) of SOM as an input. Ssince SOM generally decreases exponentially with depth (Kramer and Gleixner, 2008; Hobley and Wilson, 2016; Bai et al., 2016) this estimated $K_{sat}$ should be reproducing the decay to some degree. However, the available data used for the optimization of the pedotransfer functions calculating $K_{sat}$ was biased towards sandy and loamy-sand soils, affecting the performance in the rest of the textural classes.

The lateral flow equation proposed here is based on the profile transmissivity, originally proposed by TOPMODEL and

785 the assumption of steady-state flow. However, it's been reported that the steady-state flow assumption holds better in wet catchments with high hydraulic gradients. In dry catchments or during dry periods, areas of the catchment may lose their hydrological connectivity (Woods et al., 1997; Tague and Band, 2001).





Analyzing the response of the lateral flow to soil moisture generated by SPLASH the minimum $q_{out}$ reflects the stable gravity-driven drainage (which in shallow soils defines the baseflow), while the maximum $q_{out}$ appears as pulses product of

790 individual precipitation events mimicking the behaviour of the interflow. This response is consistent in both, wet and arid sites in different magnitudes.

In VIC3-L, however, $q_{out}$ reaches values in the order of hundreds (truncated in the figure) while soil moisture is not even saturated. In the comparable region of the analysis, the upper envelope of the scattered points shows a linear threshold and a sharp transition to a plateau. The flux here seems to display relatively high values when the soil moisture is close to the wilting

point, which is theoretically impossible.

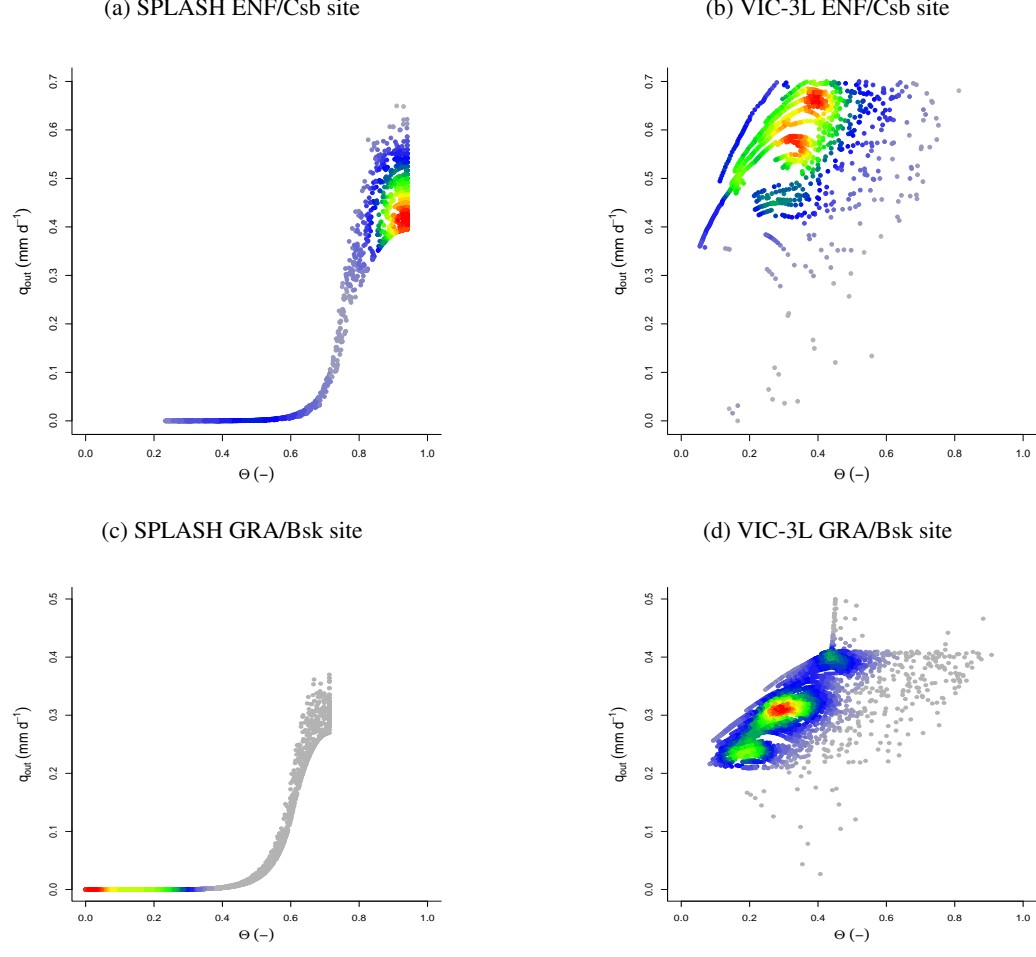

**Figure 47.** Emergent response of the lateral flow to soil moisture from an ENF in a temperate climate, with a dry and warm summer (Csb) (Site site SNTL:529), simulated from: (a) SPLASH. (b) VIC-3L. The same relationship from a GRA in an arid cold steppe (Bsk) (Site site SNTL:871), simulated from: (c) SPLASH. (d) VIC-3L.





*Code and data availability.* The equations and methods presented in this chapter were coded in C++ to improve the speed of the computation. Aiming for replicability, the codes were wrapped in an open-source R package called 'rsplash' available on Zenodo at https://zenodo.org/records/10047627

and on github: https://github.com/dsval/rsplash. The algorithms can run either at site-scale or spatially distributed on a grid. The package is coded to automatically exploit parallel computing capabilities when required. A companion R package called "splashTools" was created (https://github.com/dsval/splashTools) including all the wrappers and original code for downloading and pre-processing forcing and soil data from the U.S Natural Resources Conservation Services, SoilGrids, FLUXNET and other cited sources. All the data used in this research is open access and is available through their respective sources cited in the text.

## Appendix A: Derivations and extended mathematical analysis

### A1 Recession constant

Analysing the flux from one cell, according to the linear reservoir model (Eq. 42) and the BC model (Eq.39), the maximum
(initial) lateral flux will happen when the soil is saturated. In the same way it will be close to zero at field capacity. Thus, if we set the volume of the drainable porosity, equals to the total volume drained by Eq. (42):

$$(W_{sat} - W_{fc})\, Ai = \int\limits_{0}^{t} Q_{sat} K_b{}^t dt \tag{A1}$$

Then, solving both, Eq.(42) and Eq.(A1) for $t$, we can set:

$$\frac{1}{\ln(K_b)} \ln\left(\frac{Q_{fc}}{Q_{sat}}\right) = \frac{1}{\ln(K_b)} \ln\left(\frac{Ai\,(W_{sat} - W_{fc})\ln(K_b)}{Q_{sat}} + 1\right) \tag{A2}$$

Therefore, solving Eq. (A2) for $K_b$, we get:

$$K_b = e^{\frac{Q_{fc} - Q_{sat}}{Ai\,(W_{sat} - W_{fc})}} \tag{A3}$$





## A2 Actual field capacity

Replacing Eq. (55), Eq. (56b) and Eq.(57) in Eq.(54):

$$A(\theta)^{-B} - \rho_w \, g \, \theta \, (1000z) = 0 \tag{A4}$$

Then, converting units to SI and simplifying, we can substitute:

$$c = \frac{1000}{\rho_w \, g} \tag{A5}$$

Where 1000 is a factor to correct the units. This can be rearranged to,

$$c \, \frac{A}{z} = \frac{\theta}{\theta^{-B}} \tag{A6}$$

Thus, solving Eq.(A3) for $\theta$:

$$\theta = \left( \frac{c \, A}{z} \right)^{\frac{1}{1+B}} \tag{A7}$$

### A3 Comparable quantities from the VIC-3L model

### A3.1 $H_N^+$ from $\overline{I_N}$

VIC-3L provides daily $\overline{I_N}(W \, m^{-2})$ as output, so, the comparable quantity representing the total input of energy can be found as:

$$H_N^+ = 86400 \, \overline{I_N} + H_N^- \tag{A8}$$

Among the outputs from the VIC-3L model are $I_{SW}max$ and $I_{LW}$, so, if at solar noon $(h = 0)$ $I_{SW} = I_{SW}max$, and, at $h = h_n$, $I_{SW} = I_{LW}$, then:

$$I_{SW}max \cos h_n = I_{LW} \qquad \Rightarrow h_n = \arccos \frac{I_{LW}}{I_{SW}max} \tag{A9}$$

$I_{SW} = I_{LW}$, and $I_{SW} = 0$ at $h = h_s$ implies as well the line:

$$0 = I_{SW}max - \left( \frac{I_{SW}max - I_{LW}}{h_s} \right) h_n \tag{A10}$$

Which solving for $h_s$ yields:

$$h_s = h_n \left( 1 + \frac{I_{LW}}{I_{SW}max} \right) \tag{A11}$$

Then, the area representing $H_N^-$ is:

$$H_N^- = \frac{86400}{\pi} (\pi - h_s) I_{LW} + \frac{(h_s - h_n) I_{LW}}{2} \tag{A12}$$



### A3.2 $C_n$ from latent heat components

VIC-3L provides the daily average of the net latent heat as output, along its components used to melt/refreeze soil and snow, so $C_n$ was computed simply as the remnant negative latent heat after the components were subtracted from the net latent heat, in the way

$$C_n = (OUT_{LATENT} + OUT_{FUSION} + OUT_{MELT\ ENERGY} + OUT_{RFRZ\ ENERGY}) * (86.4/2260) - OUT_{EVAP} \quad \text{(A13)}$$

Here, the factor $(86.4/2260)$ transforms from $W\,m^{-2}$ to $mm\,d^{-1}$ assuming constant water density and vaporization heat of $2260kJ/kg$.

### A4 Water uptake/water stress functions

### A4.1 Water uptake from $\Delta\psi$

Transpiration can be defined following Campbell and Norman (1998) as:

$$E = S_W = \left(\frac{\psi_s - \psi_L}{R_p}\right) \quad \text{(A14)}$$

Where, $\psi_s$ is the soil water potential, $\psi_L$ the leaf water potential and $R_p$ the total plant hydraulic resistance, from the root to the leaf.

Then, without soil moisture limitations ($\psi_s = 0$), the transpiration becomes:

$$E_{max} = D_p = \left(\frac{-\psi_L}{R_p}\right) \quad \text{(A15)}$$

Then, solving Eq. (A15) for $R_p$ and replacing in Eq. (A14):

$$S_W = D_p \left(1 - \frac{-\psi_s}{\psi_L}\right) \quad \text{(A16)}$$

Where, $\psi_L$ is found using the first law of thermodynamics: $dU = dQ - PdV$. Where $dU$ is the change in internal energy, $dQ$ the instantaneous heat input and $PdV$ the work done at constant pressure.

So, if $\psi = \frac{U}{Vw}$, and $PdV = \frac{-nRT}{P}dP$ from the ideal gas law, assuming $dQ = 0$ at pre-dawn, then:

$$\psi_L = \frac{1}{Vw} \int_{P1}^{P2} \frac{nRT}{P} dP = \frac{RT}{Vw} \ln \frac{e_L^a}{e_L^s} \quad \text{(A17)}$$

Where, $Vw$ is the molar volume, $e_L^a$ and $e_L^s$ are the actual and saturation vapour pressure at the leaf surface, its proportion was assumed as 0.98 (Nobel, 1983).

### A4.2 Water uptake from Campbell and Norman (1998)

$$S_W = D_p \left(1 - \frac{2}{3}\frac{\psi_s}{\psi_L}\right) \quad \text{(A18)}$$



### A4.3 Water stress function from Metselaar and de Jong van Lier (2007)

$$S_W = Sc \frac{\Theta^{a+1} - \Theta_W^{a+1}}{\Theta_l^{a+1} - \Theta_W^{a+1}} \tag{A19}$$

Where:

$$\Theta = \frac{\theta - \theta_r}{\theta_s - \theta_{wp}} \tag{A20}$$

$$\Theta_l = \frac{\theta_l - \theta_r}{\theta_s - \theta_{wp}} \tag{A21}$$

$$\Theta_W = \frac{\theta_{wp} - \theta_r}{\theta_s - \theta_{wp}} \tag{A22}$$

$$\theta_l = \theta_{wp} + A * (\theta_s - \theta_{wp}) \tag{A23}$$

$$\alpha = 3 + 2\lambda \tag{A24}$$

Here, $\lambda$ is a parameter for the BC water retention model, $A$ is an empirical parameter defining the curvature of the function, $\theta_{wp}, \theta_s, \theta, \theta_r$ are the volumetric water content at wilting point, saturation, actual and residual respectively, and $Sc = 1.05(mm\,h^{-1})$ after (Federer, 1982).

### A4.4 Rainfall event duration

Pareto distribution, higher frequency events last between 0-1 hrs 80% lower 6 rainy hours, 6hrs chosen parameter





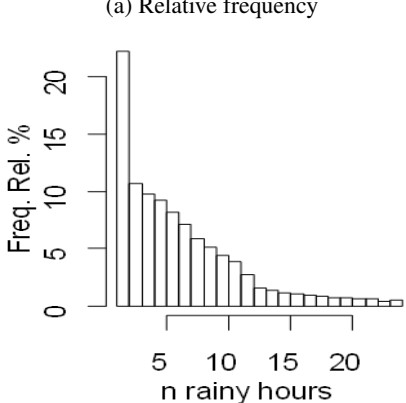
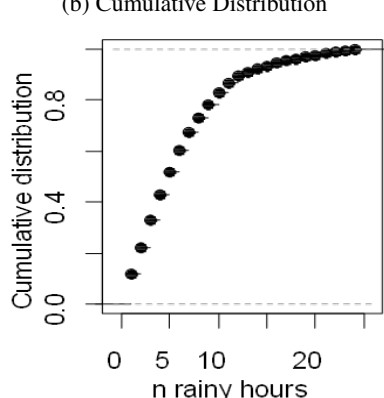

**Figure A1.** Distribution of daily rainy-hour counts from GsMap global hourly data 2000-2014. (a) Relative frequencies. (b) Cumulative probability

## A5  Pedotransfer functions

From the models tested (Table 2), the equations from Balland et al. (2008), which use the largest dataset and non-linear
formulations outperformed the other models with the explained variance exceeding 60% for permanent wilting point ($\theta_{1500}$), field capacity ($\theta_{33}$) and saturation ($\theta_{sat}$) (Fig.A2).





**Figure A2.** Evaluation of different pedotransfer functions to estimate $\theta_{1500}$, $\theta_{33}$ and $\theta_{sat}$, n = 68567.



The PTFs for computing $K_{sat}$ showed poor performances for all the models, most of the measurements seem to cluster around 0 to 50 $mm\,h^{-1}$ which is only captured by the Cosby et al. (1984) model. Overall, the explained variance of all the models didn't reach the 10%, (Fig. A3). Here, $K_{sat}$ values exceeding 526 $mm\,h^{-1}$ were considered as outliers (percentile

90%) and excluded from the analysis, similar maximum values were presented by Van Looy et al. (2017).

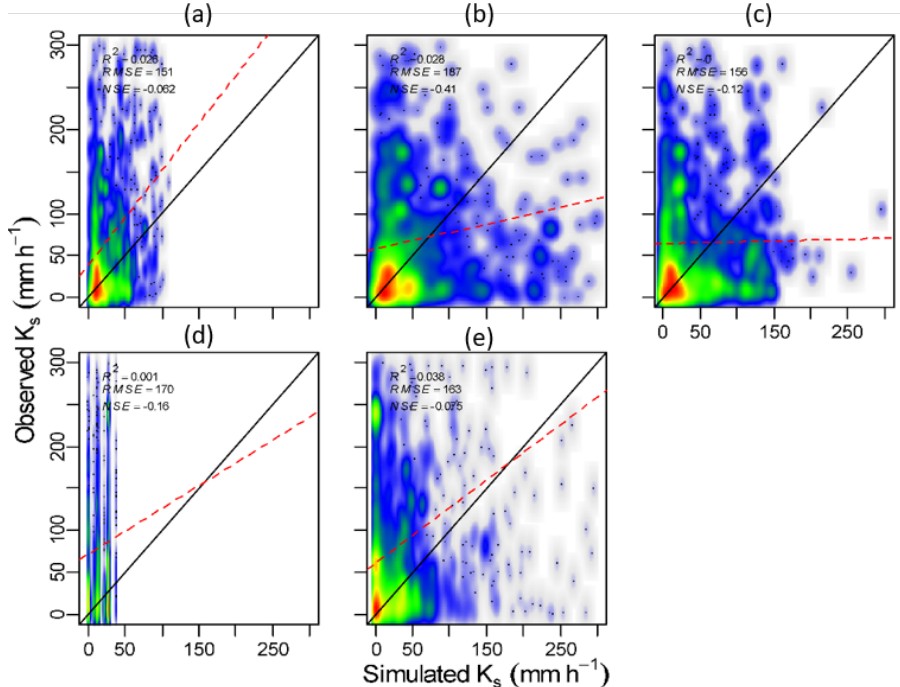

**Figure A3.** Evaluation of different pedotransfer functions to estimate $K_{sat}$. (a) Cosby et al. (1984). (b) Balland et al. (2008). (c) Saxton and Rawls (2006). (d) Tóth et al. (2015). (e) Rosetta 3 (Zhang and Schaap, 2017).

$K_{sat}$ estimated by the Saxton and Rawls (2006) PFT was the best performing model, however, it leads to unrealistic values when the drainable porosity ($\theta_{sat} - \theta_{33}$) is relatively high. So, a simple exponential saturating curve was adopted here, which yields similar estimations to Saxton and Rawls (2006) at the lower end of the drainable porosity but it flattens at a fitted $K_{sat}$ maximum of 623 $mm\,h^{-1}$.

Furthermore, to improve these estimations, the equations from Balland et al. (2008) (Eq. A25a,A25b,A25c,A25d) were optimized using the full dataset employed for this evaluation resulting in the parameters detailed in Table A1.

$$\theta_{1500} = \theta_{33}\left(c_{wp} + (d_{wp} - c_{wp})\,CLAY^{0.5}\right) \tag{A25a}$$

$$\theta_{33} = \theta_{sat}\left(c_{fc} + (d_{fc} - c_{fc})\,CLAY^{0.5}\right) e^{\frac{a_{fc}SAND - b_{fc}SOM}{\theta_{sat}}} \tag{A25b}$$

$$\theta_{sat} = 1 - \frac{\rho_b}{\rho_p} \tag{A25c}$$

$$K_{sat} = 10^{a_{ks} + b_{ks}\,log_{10}(\rho_p - \rho_b) + c_{ks}SAND} \tag{A25d}$$



Were $\theta_{1500}$ is wilting point (water held at 1500kPa), $\theta_{33}$ is field capacity (water held at 33kPa), $\theta_{sat}$ is saturation, $K_{sat}$ the saturated hydraulic conductivity, and, SAND, CLAY and SOM refer to sand, clay and organic matter contents (%). $a, b, c$ are constants with the subscripts referring to wilting point, field capacity or hydraulic saturated conductivity respectively, $\rho_b$ is the bulk density, and $\rho_p$ is the particle density, calculated as follows (Balland et al., 2008):

$\qquad \rho_p = \dfrac{1}{\frac{SOM}{1.3} + \frac{1-SOM}{2.65}}$ $\hfill$ (A26)

**Table A1.** Updated parameters for the Balland et al. (2008) PTFs

|           | $a_x$    | $b_x$    | $c_x$  | $d_x$  |
|-----------|----------|----------|--------|--------|
| $\theta_{1500}$ | –        | –        | 0.2018 | 0.7809 |
| $\theta_{33}$   | -0.0547  | -0.0010  | 0.4760 | 0.9402 |
| $K_{sat}$       | -2.6539  | 3.0924   | 4.2146 | –      |

**Table A1.** IGBP Biomes and their description

| Biome | Code | Description |
|-------|------|-------------|
| Evergreen Needleleaf Forests | ENF | Dominated by evergreen conifer trees (canopy >2m). Tree cover >60%. |
| Evergreen Broadleaf Forests | EBF | Dominated by evergreen broadleaf and palmate trees (canopy >2m). Tree cover >60%. |
| Deciduous Needleleaf Forests | DNF | Dominated by deciduous needleleaf (larch) trees (canopy >2m). Tree cover >60%. |
| Deciduous Broadleaf Forests | DBF | Dominated by deciduous broadleaf trees (canopy >2m). Tree cover >60%. |
| Mixed Forests | MF | Dominated by neither deciduous nor evergreen (40-60% of each) tree type (canopy >2m). Tree cover >60%. |
| Closed Shrublands | CSH | Dominated by >60% cover. woody perennials (1-2m height) |
| Open Shrublands | OSH | Dominated by 10-60% cover. woody perennials (1-2m height) |
| Woody Savannas | WSA | Tree cover 30-60% (canopy >2m). |
| Savannas | SAV | Tree cover 10-30% (canopy >2m). |
| Grasslands | GRA | Dominated by herbaceous annuals (<2m). |
| Permanent Wetlands | WET | Permanently inundated lands with 30-60% water cover and >10% vegetated cover. |
| Croplands | CRO | At least 60% of area is cultivated cropland. |
| Cropland/Natural Vegetation Mosaics | CNV | Mosaics of small-scale cultivation 40-60% with natural tree, shrub, or herbaceous vegetation. |
| Permanent Snow and Ice | SNO | At least 60% of area is covered by snow and ice for at least 10 months of the year. |
| Barren | BSV | At least 60% of area is non-vegetated barren (sand, rock, soil) areas with less than 10% vegetation. |





**Table A2.** Köppen-Geiger climate zones after Beck et al. (2018). Where, $T_{cold}$ is the air temperature of the coldest month (°C); $T_{hot}$ is the air temperature of the warmest month (°C); $T_{mon10}$ is the number of months with air temperature > 10 °C (unitless); $P_{dry}$ is the precipitation in the driest month ($mm\,mo^{-1}$); $P_{sdry}$ is the precipitation in the driest month in summer ($mm\,mo^{-1}$); $P_{wdry}$ is precipitation in the driest month in winter ($mm\,mo^{-1}$); $P_{swet}$ is precipitation in the wettest month in summer ($mm\,mo^{-1}$); $P_{wwet}$ is precipitation in the wettest month in winter ($mm\,mo^{-1}$). $P_{threshold}$ = 2×MAT if >70% of precipitation falls in winter, $P_{threshold}$ =2×MAT+28 if >70% of precipitation falls in summer, otherwise $P_{threshold}$ = 2×MAT+14.

| $1^{st}$ level | $2^{nd}$ level | $3^{rd}$ level | Description | Criteria |
|---|---|---|---|---|
| | | | Tropical | Not (B) and $T_{cold} \leq 18$ $P_{dry}$ |
| | f | | Rainforest | $P_{dry} \geq 60$ |
| A | m | | Monsoon | Not (Af) and $P_{dry} \geq 100 - MAP/25$ |
| | w | | Savanna (Dry winter) | Not (Af) and $P_{dry} < 100 - MAP/25$ |
| | s | | Savanna (Dry summer) | Not (Af) and $P_{dry} < P_{wwet}/3$ |
| | | | Arid | $MAP < 10 \times P_{threshold}$ |
| | W | | Desert | $MAP < 5 \times P_{threshold}$ |
| B | S | | Steppe | $MAP \geq 5 \times P_{threshold}$ |
| | | h | - Hot | $MAP \geq 18$ |
| | | k | - Cold | $MAP < 18$ |
| | | | Temperate | Not (B) $T_{hot} > 10$ and $0 < T_{cold} < 18$ |
| | s | | Dry summer | $P_{sdry} < 40$ and $P_{sdry} < P_{wwet}/3$ |
| | w | | Dry winter | $P_{sdry} < P_{swet}/10$ |
| C | f | | No dry season | Not (Cs) or (Cw) |
| | | a | - Hot summer | $T_{hot} \geq 22$ |
| | | b | - Warm summer | Not (a) and $T_{mon} \geq 4$ |
| | | c | - Cold summer | Not (a or b) and $1 \leq T_{mon10} < 4$ |
| | | | Continental | Not (B) $T_{hot} > 10$ and $0 < T_{cold} \leq 0$ |
| | s | | Dry summer | $P_{sdry} < 40$ and $P_{sdry} < P_{wwet}/3$ |
| | w | | Dry winter | $P_{wdry} < P_{swet}/10$ |
| D | f | | No dry season | Not (Ds) or (Dw) |
| | | a | - Hot summer | $T_{hot} \geq 22$ |
| | | b | - Warm summer | Not (a) and $T_{mon} \geq 4$ |
| | | c | - Cold summer | Not (a, b, or d) |
| | | d | - Very cold winter | Not (a or b) and $T_{cold} < -38$ |
| | | | Polar | Not (B) $T_{hot} \leq 10$ |
| E | T | | Tundra | $T_{hot} > 0$ |
| | F | | Frost | $T_{hot} \leq 0$ |



Table A3: Eddy covariance stations used in the performance evaluation. Where Lat. is latitude (°), Lon. is longitude (°), Net. is the network, here FLX, EUR and AME stand for Fluxnet, Europeflux and Ameriflux respectively. Climate refers to the Köppen-Geiger climate zone, Biome as defined by the IGBP (Table A1) (Friedl et al., 2019). Elev. is the elevation in masl, Slop. and Asp. are the slope (°) and aspect (°) respectively and Au. is the upslope draining area ($km^2$)

| Station | Lat. | Lon. | Net. | Climate | Biome | Period | Elev. | Slop. | Asp. | Au | Reference |
|---------|------|------|------|---------|-------|--------|-------|-------|------|-----|-----------|
| AT-Neu | 47.12 | 11.32 | FLX | Dfc | GRA | 2002-2012 | 981 | 10 | 135 | 0.50 | Wohlfahrt et al. (2008) |
| AU-Rob | -17.12 | 145.63 | FLX | Cfa | EBF | 2014-2014 | 712 | 1 | 126 | 1.88 | Bristow et al. (2016) |
| AU-Tum | -35.66 | 148.15 | FLX | Cfb | ENF | 2001-2014 | 1243 | 2 | 26 | 0.08 | Leuning et al. (2005) |
| AU-Wac | -37.43 | 145.19 | FLX | Cfb | EBF | 2005-2008 | 704 | 0 | 311 | 0.01 | Kilinc et al. (2013) |
| BE-Jal | 50.56 | 6.07 | EUR | Cfb | MF | 2006-2007 | 480 | 1 | 198 | 1.49 | – – – |
| CH-Cha | 47.21 | 8.41 | FLX | Cfb | SAV | 2005-2014 | 392 | 0 | 331 | 0.07 | Merbold et al. (2014) |
| CH-Dav | 46.82 | 9.86 | FLX | ET | MF | 1997-2014 | 1628 | 22 | 271 | 0.01 | Zielis et al. (2014) |
| CH-Fru | 47.12 | 8.54 | FLX | Cfb | GRA | 2005-2014 | 974 | 1 | 262 | 0.02 | Imer et al. (2013) |
| CH-Lae | 47.48 | 8.37 | FLX | Cfb | MF | 2004-2014 | 679 | 18 | 180 | 0.02 | Etzold et al. (2011) |
| CH-Oe1 | 47.29 | 7.73 | FLX | Cfb | CRO | 2002-2008 | 454 | 0 | 78 | 0.02 | Ammann et al. (2009) |
| CH-Oe2 | 47.29 | 7.73 | FLX | Cfb | CRO | 2004-2014 | 454 | 0 | 78 | 0.02 | Dietiker et al. (2010) |
| CN-Dan | 30.50 | 91.07 | FLX | ET | GRA | 2004-2005 | 4316 | 1 | 128 | 0.70 | Shi et al. (2006) |
| CN-Ha2 | 37.61 | 101.33 | FLX | ET | GRA | 2003-2005 | 3205 | 0 | 186 | 0.06 | – – – |
| CN-HaM | 37.37 | 101.18 | FLX | ET | GRA | 2002-2004 | 3985 | 9 | 97 | 0.01 | Kato et al. (2006) |
| CZ-BK1 | 49.50 | 18.54 | FLX | Dfb | ENF | 2004-2014 | 833 | 9 | 62 | 0.01 | Acosta et al. (2013) |
| CZ-BK2 | 49.49 | 18.54 | FLX | Dfb | MF | 2004-2012 | 859 | 5 | 125 | 0.03 | Sigut et al. |
| DE-Lkb | 49.10 | 13.30 | FLX | Cfb | GRA | 2009-2013 | 1305 | 5 | 160 | 0.01 | Lindauer et al. (2014) |
| DE-Obe | 50.78 | 13.72 | FLX | Cfb | ENF | 2008-2014 | 767 | 3 | 336 | 0.01 | – – – |
| DK-NuF | 64.13 | -51.39 | FLX | ET | GRA | 2008-2009 | 46 | 4 | 343 | 0.01 | Westergaard-Nielsen et al. (2013) |
| ES-LgS | 37.10 | -2.97 | FLX | Csa | SAV | 2007-2009 | 2250 | 3 | 322 | 0.06 | Reverter et al. (2010) |
| ES-LJu | 36.93 | -2.75 | EUR | Csa | OSH | 2004-2013 | 1610 | 3 | 309 | 0.01 | – – – |
| ES-Ln2 | 36.97 | -3.48 | FLX | Csa | SAV | 2009-2009 | 2180 | 9 | 186 | 0.01 | Serrano-Ortiz et al. (2011) |
| ES-VDA | 42.15 | 1.45 | EUR | Cfb | SAV | 2004-2008 | 1759 | 2 | 318 | 0.09 | – – – |
| FR-Lq1 | 45.64 | 2.74 | EUR | Cfb | SAV | 2004-2010 | 1031 | 4 | 257 | 0.96 | – – – |
| FR-Lq2 | 45.64 | 2.74 | EUR | Cfb | SAV | 2006-2010 | 1031 | 4 | 257 | 0.96 | – – – |
| GL-NuF | 64.13 | -51.39 | EUR | ET | GRA | 2008-2014 | 46 | 4 | 343 | 0.01 | López-Blanco et al. (2017) |
| HU-Mat | 47.85 | 19.73 | EUR | Cfb | SAV | 2004-2008 | 302 | 4 | 236 | 0.03 | – – – |
| IL-Yat | 31.35 | 35.05 | EUR | BSh | ENF | 2001-2009 | 647 | 1 | 245 | 0.03 | – – – |
| IT-Amp | 41.90 | 13.61 | EUR | Cfa | SAV | 2002-2008 | 836 | 1 | 202 | 0.15 | – – – |
| IT-Col | 41.85 | 13.59 | FLX | Cfa | SAV | 1996-2014 | 1571 | 11 | 222 | 0.12 | Valentini et al. (1996) |
| IT-La2 | 45.95 | 11.29 | FLX | Cfb | ENF | 2000-2002 | 1375 | 4 | 183 | 0.01 | Marcolla et al. (2003) |
| IT-Lav | 45.96 | 11.28 | FLX | Cfb | ENF | 2003-2014 | 1403 | 5 | 213 | 0.02 | – – – |
| IT-Mal | 46.11 | 11.70 | EUR | ET | MF | 2003-2006 | 1660 | 8 | 142 | 0.08 | – – – |
| IT-MBo | 46.01 | 11.05 | FLX | Dfb | GRA | 2003-2013 | 1557 | 3 | 112 | 0.65 | Marcolla et al. (2011) |
| IT-Ren | 46.59 | 11.43 | FLX | Dfc | WSA | 1998-2013 | 1761 | 11 | 221 | 0.03 | Montagnani et al. (2009) |
| IT-To1 | 42.19 | 11.92 | EUR | Csa | MF | 2004-2006 | 367 | 7 | 3 | 0.01 | – – – |
| IT-To2 | 42.19 | 11.92 | EUR | Csa | MF | 2004-2006 | 367 | 7 | 3 | 0.01 | – – – |
| IT-Tol | 42.19 | 11.92 | EUR | Csa | MF | 2005-2006 | 367 | 7 | 3 | 0.01 | – – – |
| IT-Tor | 45.84 | 7.58 | FLX | Dfc | GRA | 2008-2014 | 2133 | 15 | 231 | 0.03 | Galvagno et al. (2013) |
| JP-MBF | 44.39 | 142.32 | FLX | Dfb | WSA | 2003-2005 | 580 | 2 | 88 | 0.02 | – – – |
| SE-St1 | 68.35 | 19.05 | EUR | ET | WET | 2012-2014 | 352 | 0 | 101 | 0.33 | – – – |
| SK-Ta1 | 49.16 | 20.26 | EUR | Dfc | WSA | 2006-2007 | 1010 | 7 | 130 | 0.06 | – – – |
| SK-Ta2 | 49.13 | 20.11 | EUR | Dfc | WSA | 2007-2007 | 1268 | 9 | 187 | 0.58 | – – – |
| SK-Tat | 49.12 | 20.16 | EUR | Dfc | WSA | 2005-2007 | 1043 | 7 | 133 | 0.19 | – – – |



Continuation of Table A3

| Station | Lat. | Lon | Net. | Climate | Biome | Period | Elev. | Slop. | Asp | Au | Reference |
|---|---|---|---|---|---|---|---|---|---|---|---|
| UK-Gri | 56.61 | -3.80 | EUR | Cfc | ENF | 2000-2006 | 359 | 2 | 0 | 0.23 | − − − |
| US-ADR | 36.77 | -116.69 | AME | BWk | BSV | 2011-2017 | 852 | 0 | 158 | 0.11 | ? |
| US-Blk | 44.16 | -103.65 | AME | Dfb | WSA | 2004-2008 | 1762 | 4 | 135 | 0.09 | Novick et al. (2016) |
| US-Blo | 38.90 | -120.63 | FLX | Csa | ENF | 1997-2007 | 1330 | 2 | 251 | 0.01 | Goldstein et al. (2000) |
| US-CaV | 39.06 | -79.42 | AME | Cfb | WSA | 2004-2010 | 978 | 2 | 293 | 0.03 | Meyers (2016) |
| US-CPk | 41.07 | -106.12 | AME | Dfc | GRA | 2009-2013 | 2758 | 0 | 73 | 0.02 | Chu et al. (2018a) |
| US-CZ2 | 37.03 | -119.26 | AME | Csb | WSA | 2010-2016 | 1181 | 3 | 91 | 0.05 | Goulden (2018a) |
| US-CZ3 | 37.07 | -119.20 | AME | Csb | ENF | 2008-2016 | 2020 | 2 | 231 | 0.01 | Goulden (2018b) |
| US-CZ4 | 37.07 | -118.99 | AME | Dsc | WSA | 2009-2016 | 2712 | 1 | 158 | 0.05 | Goulden (2018c) |
| US-EML | 63.88 | -149.25 | AME | Dfc | OSH | 2008-2018 | 675 | 1 | 0 | 0.01 | Belshe et al. (2012) |
| US-Fmf | 35.14 | -111.73 | AME | Csb | GRA | 2005-2010 | 2198 | 0 | 81 | 0.03 | Amiro et al. (2010) |
| US-Fwf | 35.45 | -111.77 | AME | Csb | GRA | 2005-2010 | 2309 | 2 | 0 | 1.14 | Amiro et al. (2010) |
| US-GBT | 41.37 | -106.24 | FLX | Dfc | GRA | 1999-2006 | 3193 | 3 | 182 | 0.05 | Zeller and Nikolov (2000) |
| US-GLE | 41.37 | -106.24 | FLX | Dfc | GRA | 2004-2014 | 3193 | 3 | 182 | 0.05 | Frank et al. (2014) |
| US-GMF | 41.97 | -73.23 | AME | Dfb | MF | 1999-2004 | 500 | 3 | 73 | 0.02 | Chu et al. (2018b) |
| US-HBK | 43.94 | -71.72 | AME | Dfb | DBF | 2016-2019 | 396 | 6 | 356 | 0.01 | Kelsey and Green (2020) |
| US-ICh | 68.61 | -149.30 | AME | Dfc | OSH | 2007-2011 | 950 | 3 | 271 | 0.01 | Euskirchen et al. (2012a) |
| US-ICs | 68.61 | -149.31 | AME | Dfc | OSH | 2007-2011 | 898 | 1 | 281 | 0.57 | Euskirchen et al. (2017) |
| US-ICt | 68.61 | -149.30 | AME | Dfc | OSH | 2007-2011 | 918 | 5 | 261 | 0.01 | Euskirchen et al. (2012b) |
| US-Me1 | 44.58 | -121.50 | FLX | Csb | GRA | 2004-2005 | 893 | 4 | 79 | 0.02 | Irvine et al. (2007) |
| US-Me2 | 44.45 | -121.56 | FLX | Csb | ENF | 2002-2014 | 1259 | 1 | 166 | 0.01 | Campbell et al. (2004) |
| US-Me3 | 44.32 | -121.61 | AME | Csb | WSA | 2004-2009 | 1006 | 0 | 49 | 0.04 | Barr et al. (2013a) |
| US-Me4 | 44.50 | -121.62 | AME | Csb | ENF | 1996-2000 | 956 | 7 | 274 | 0.03 | Anthoni et al. (1999) |
| US-Me5 | 44.44 | -121.57 | AME | Csb | ENF | 2000-2002 | 1189 | 2 | 208 | 0.81 | Anthoni et al. (2002) |
| US-Me6 | 44.32 | -121.61 | FLX | Csb | WSA | 2010-2014 | 999 | 0 | 65 | 23.80 | Ruehr et al. (2012) |
| US-MRf | 44.65 | -123.55 | AME | Csb | ENF | 2006-2011 | 265 | 6 | 246 | 0.02 | Chu et al. (2018c) |
| US-NR1 | 40.03 | -105.55 | FLX | Dfc | WSA | 1998-2014 | 3039 | 7 | 83 | 0.03 | Monson et al. (2002) |
| US-Rls | 43.14 | -116.74 | AME | BSk | GRA | 2014-2018 | 1611 | 5 | 343 | 0.02 | Fellows et al. (2020) |
| US-Rms | 43.06 | -116.75 | AME | BSk | GRA | 2014-2018 | 2116 | 3 | 324 | 0.01 | Fellows et al. (2020) |
| US-Rws | 43.17 | -116.71 | AME | BSk | GRA | 2014-2018 | 1437 | 4 | 349 | 0.07 | Fellows et al. (2020) |
| US-SCf | 33.81 | -116.77 | AME | Csa | ENF | 2006-2015 | 1739 | 4 | 176 | 0.01 | Goulden (2018d) |
| US-SCg | 33.74 | -117.69 | AME | Csa | OSH | 2006-2016 | 440 | 3 | 9 | 0.01 | Goulden (2018e) |
| US-SCs | 33.73 | -117.70 | AME | Csa | GRA | 2006-2016 | 470 | 4 | 96 | 0.02 | Goulden (2018f) |
| US-SCw | 33.60 | -116.45 | AME | BWh | OSH | 2006-2016 | 1290 | 2 | 263 | 0.01 | Goulden (2018g) |
| US-SO2 | 33.37 | -116.62 | AME | Csa | SAV | 1997-2006 | 1423 | 5 | 188 | 0.39 | Barr et al. (2013b) |
| US-SO3 | 33.38 | -116.62 | AME | Csa | GRA | 1997-2006 | 1480 | 7 | 201 | 0.01 | Baldocchi et al. (2015) |
| US-SO4 | 33.38 | -116.64 | AME | Csa | GRA | 2004-2006 | 1416 | 3 | 206 | 0.01 | Baldocchi et al. (2015) |
| US-SRC | 31.91 | -110.84 | AME | BSk | OSH | 2008-2014 | 985 | 1 | 304 | 3.06 | Wolf et al. (2016) |
| US-SRG | 31.79 | -110.83 | FLX | BSk | GRA | 2008-2014 | 1288 | 3 | 308 | 0.39 | Scott et al. (2015) |
| US-SRM | 31.82 | -110.87 | FLX | BSk | GRA | 2004-2014 | 1112 | 1 | 295 | 0.05 | Barron-Gafford et al. (2013) |
| US-Vcm | 35.89 | -106.53 | AME | Cfb | GRA | 2007-2019 | 2984 | 7 | 334 | 0.01 | Anderson-Teixeira et al. (2011) |
| US-Vcp | 35.86 | -106.60 | AME | Cfb | GRA | 2007-2019 | 2503 | 9 | 255 | 0.19 | Anderson-Teixeira et al. (2011) |
| US-Vcs | 35.92 | -106.61 | AME | Cfb | WSA | 2016-2019 | 2775 | 3 | 165 | 0.01 | Remy et al. (2019) |



Table A4: SNOTEL stations used in the performance evaluation. Where Lat. is latitude (°), Lon. is longitude (°), Climate refers to the Köppen-Geiger climate zone, Biome as defined by the IGBP (Table A1) (Friedl et al., 2019). Elev. is the elevation in masl, Slop. and Asp. are the slope (°) and aspect (°) respectively, Au. is the upslope draining area ($km^2$), and SM sens. is the depth of the soil profile ($m$) where the soil moisture was measured.

| Station | Lat. | Lon | Climate | Biome | Period | Elev. | Slop. | Asp | Au | SM sens. | Site name |
|---|---|---|---|---|---|---|---|---|---|---|---|
| SNTL:1243 | 40.86 | -115.22 | BSk | GRA | 2014-2015 | 1997 | 4 | 200 | 0.53 | 0.49 | Dry Creek - NV |
| SNTL:1242 | 39.25 | -119.88 | Dsb | SAV | 2014-2015 | 1979 | 0 | 116 | 25.64 | 0.49 | Little Valley - NV |
| SNTL:1244 | 40.86 | -115.12 | Dsb | GRA | 2014-2015 | 2365 | 7 | 148 | 8.50 | 0.49 | Pole Canyon - NV |
| SNTL:1236 | 39.66 | -110.38 | BSk | GRA | 2014-2015 | 2501 | 11 | 162 | 0.04 | 0.49 | Corral - UT |
| SNTL:1248 | 37.60 | -112.93 | Dsb | GRA | 2014-2015 | 2453 | 8 | 208 | 0.04 | 0.49 | Suu Ranch - UT |
| SNTL:1214 | 40.46 | -112.25 | Dfb | GRA | 2013-2015 | 1987 | 10 | 318 | 0.04 | 0.98 | Bevans Cabin - UT |
| SNTL:1153 | 37.87 | -109.45 | Dsb | WSA | 2013-2015 | 2720 | 6 | 1 | 5.66 | 0.49 | Buckboard Flat - UT |
| SNTL:1215 | 38.48 | -109.29 | Dsb | GRA | 2013-2015 | 2677 | 10 | 299 | 0.04 | 0.49 | Lasal Mountain-Lower - UT |
| SNTL:1225 | 40.68 | -111.22 | Dfb | SAV | 2013-2015 | 2600 | 5 | 302 | 0.04 | 0.49 | Redden Mine Lwr - UT |
| SNTL:1217 | 39.50 | -111.73 | BSk | GRA | 2013-2015 | 2196 | 14 | 4 | 0.08 | 0.49 | Rees Flat - UT |
| SNTL:1155 | 39.46 | -114.65 | BSk | SAV | 2012-2015 | 2297 | 13 | 21 | 0.04 | 0.49 | Bird Creek - NV |
| SNTL:1207 | 41.89 | -115.86 | Dfb | GRA | 2012-2012 | 2105 | 7 | 329 | 0.20 | 0.49 | Merritt Mountain - NV |
| SNTL:1192 | 40.84 | -110.66 | Dfc | SAV | 2012-2015 | 2935 | 2 | 63 | 47.60 | 0.49 | Buck Pasture - UT |
| SNTL:1184 | 38.79 | -111.69 | BSk | SAV | 2012-2015 | 2559 | 6 | 82 | 0.13 | 0.49 | Gooseberry RS Up - UT |
| SNTL:992 | 40.89 | -110.83 | Dfc | GRA | 2011-2015 | 2675 | 3 | 281 | 0.40 | 0.49 | Bear River RS - UT |
| SNTL:1149 | 38.50 | -111.77 | BSk | GRA | 2011-2015 | 2681 | 0 | 192 | 159.55 | 0.49 | Fish Lake Utah - UT |
| SNTL:1145 | 41.25 | -111.41 | Dfb | SAV | 2011-2015 | 2200 | 10 | 180 | 0.32 | 0.49 | Kilfoil Creek - UT |
| SNTL:1159 | 48.95 | -118.99 | Dfc | WSA | 2011-2015 | 1633 | 7 | 307 | 0.04 | 0.49 | Gold Axe Camp - WA |
| SNTL:1144 | 47.98 | -114.35 | Dfc | WSA | 2010-2014 | 1722 | 9 | 208 | 0.04 | 0.98 | Blacktail Mtn - MT |
| SNTL:1117 | 40.84 | -110.01 | Dfc | WSA | 2010-2015 | 3119 | 4 | 30 | 0.20 | 0.49 | Spirit Lk - UT |
| SNTL:1113 | 41.89 | -111.57 | Dfb | GRA | 2010-2015 | 1929 | 10 | 100 | 0.12 | 0.49 | Tony Grove RS - UT |
| SNTL:1130 | 43.67 | -109.38 | Dfc | GRA | 2010-2014 | 2560 | 7 | 167 | 0.04 | 0.98 | Castle Creek - WY |
| SNTL:1123 | 40.51 | -105.77 | Dfc | WSA | 2009-2014 | 3041 | 7 | 118 | 0.12 | 0.49 | Long Draw Resv - CO |
| SNTL:1080 | 48.93 | -121.20 | Dsc | SAV | 2009-2014 | 1776 | 19 | 12 | 0.11 | 0.49 | Brown Top - WA |
| SNTL:1129 | 45.99 | -120.81 | Dsc | WSA | 2009-2015 | 1633 | 8 | 170 | 0.11 | 0.20 | Indian Rock - WA |
| SNTL:1107 | 47.71 | -123.46 | ET | ENF | 2008-2014 | 1484 | 13 | 231 | 0.11 | 0.20 | Buckinghorse - WA |
| SNTL:1085 | 46.87 | -121.53 | Dsc | WSA | 2007-2011 | 1597 | 23 | 182 | 0.11 | 0.49 | Cayuse Pass - WA |
| SNTL:1081 | 47.86 | -117.04 | Dsb | MF | 2006-2014 | 1283 | 17 | 46 | 0.18 | 0.49 | Ragged Mountain - ID |
| SNTL:999 | 48.76 | -121.70 | Dfc | ENF | 2006-2009 | 1072 | 14 | 220 | 0.04 | 0.98 | Marten Ridge - WA |
| SNTL:1056 | 41.36 | -111.49 | Dfb | GRA | 2005-2015 | 2503 | 3 | 194 | 0.04 | 0.98 | Lightning Ridge - UT |
| SNTL:1051 | 38.72 | -119.89 | Dsb | GRA | 2004-2015 | 2477 | 9 | 213 | 0.08 | 0.49 | Burnside Lake - CA |
| SNTL:1049 | 38.68 | -119.96 | Dsc | SAV | 2004-2015 | 2443 | 3 | 172 | 0.08 | 0.49 | Forestdale Creek - CA |
| SNTL:1043 | 48.86 | -118.40 | Dfb | ENF | 2003-2014 | 1426 | 13 | 325 | 0.11 | 0.20 | Sentinel Butte - WA |
| SNTL:1013 | 41.79 | -111.55 | Dfb | GRA | 2002-2015 | 2257 | 4 | 254 | 0.04 | 0.49 | Temple Fork - UT |
| SNTL:989 | 46.81 | -116.85 | Dsb | ENF | 2001-2015 | 1432 | 2 | 267 | 0.04 | 0.49 | Moscow Mountain - ID |
| SNTL:926 | 43.73 | -113.83 | Dfc | GRA | 2001-2014 | 2901 | 8 | 334 | 0.08 | 0.49 | Smiley Mountain - ID |
| SNTL:979 | 44.38 | -116.34 | Dsb | GRA | 2001-2014 | 1499 | 7 | 205 | 0.88 | 0.49 | Van Wyck - ID |
| SNTL:990 | 48.88 | -121.26 | Dfc | ENF | 2001-2008 | 1106 | 14 | 55 | 0.18 | 0.98 | Beaver Pass - WA |
| SNTL:985 | 46.24 | -117.39 | Dsb | GRA | 2000-2014 | 1219 | 10 | 339 | 0.04 | 0.49 | Sourdough Gulch - WA |
| SNTL:978 | 43.76 | -116.10 | Dsb | GRA | 1999-2014 | 1932 | 13 | 336 | 0.04 | 0.49 | Bogus Basin - ID |
| SNTL:2029 | 43.29 | -116.84 | BSk | GRA | 1999-2014 | 1706 | 7 | 151 | 0.04 | 0.98 | Reynolds Creek - ID |
| SNTL:942 | 47.04 | -121.94 | Dfc | ENF | 1999-2013 | 1271 | 17 | 205 | 0.07 | 0.49 | Burnt Mountain - WA |
| SNTL:974 | 47.94 | -123.43 | Dsc | ENF | 1999-2014 | 1527 | 24 | 216 | 0.07 | 0.49 | Waterhole - WA |
| SNTL:939 | 39.03 | -106.08 | Dfc | GRA | 1998-2008 | 3157 | 5 | 338 | 0.12 | 0.98 | Rough And Tumble - CO |
| SNTL:921 | 35.70 | -105.81 | Dfb | GRA | 1997-2000 | 2502 | 7 | 290 | 0.91 | 0.49 | Elk Cabin - NM |



Continuation of Table A4

| Station | Lat. | Lon | Climate | Biome | Period | Elev. | Slop. | Asp | Au | SM sens. | Site name |
|---------|------|-----|---------|-------|--------|-------|-------|-----|-----|----------|-----------|
| SNTL:914 | 37.85 | -105.44 | Dfc | SAV | 1996-2001 | 2941 | 9 | 66 | 0.04 | 0.49 | Medano Pass - CO |
| SNTL:907 | 37.52 | -112.27 | Dfb | GRA | 1995-2004 | 2712 | 12 | 202 | 0.04 | 0.49 | Agua Canyon - UT |
| SNTL:896 | 40.87 | -111.72 | Dsb | SAV | 1994-2009 | 2209 | 8 | 283 | 0.12 | 0.49 | Hardscrabble - UT |
| SNTL:895 | 43.77 | -114.42 | Dfb | SAV | 1993-2005 | 1923 | 8 | 136 | 1.66 | 0.49 | Chocolate Gulch - ID |
| SNTL:633 | 38.67 | -119.61 | Dsb | GRA | 1991-2001 | 2531 | 6 | 10 | 0.17 | 0.49 | Monitor Pass - CA |
| SNTL:387 | 37.66 | -107.80 | Dfb | SAV | 1990-1995 | 2718 | 6 | 169 | 0.21 | 0.49 | Cascade #2 - CO |
| SNTL:623 | 47.15 | -116.27 | Dsb | ENF | 1990-1998 | 1374 | 11 | 330 | 0.04 | 0.49 | Mica Creek - ID |
| SNTL:871 | 42.01 | -115.00 | BSk | GRA | 1990-2002 | 2170 | 3 | 229 | 0.04 | 0.49 | Wilson Creek - ID |
| SNTL:599 | 46.36 | -121.08 | Dsc | GRA | 1990-2004 | 1560 | 3 | 44 | 0.04 | 0.20 | Lost Horse - WA |
| SNTL:783 | 47.18 | -114.33 | Dfc | WSA | 1989-1997 | 1874 | 11 | 27 | 0.47 | 0.98 | Sleeping Woman - MT |
| SNTL:707 | 47.88 | -117.09 | Dsb | ENF | 1986-1998 | 1432 | 9 | 272 | 0.04 | 0.49 | Quartz Peak - WA |
| SNTL:522 | 40.91 | -109.96 | Dfc | GRA | 1985-1995 | 2780 | 2 | 191 | 0.57 | 0.49 | Hickerson Park - UT |
| SNTL:694 | 38.88 | -112.25 | Dfb | SAV | 1985-1995 | 2662 | 27 | 10 | 0.08 | 0.49 | Pine Creek - UT |
| SNTL:778 | 38.67 | -119.82 | Dsb | SAV | 1984-1996 | 1848 | 11 | 97 | 0.04 | 0.49 | Spratt Creek - CA |
| SNTL:425 | 46.56 | -115.29 | Dsc | WSA | 1984-1988 | 1816 | 1 | 25 | 0.04 | 0.49 | Crater Meadows - ID |
| SNTL:340 | 39.45 | -119.94 | Dsb | SAV | 1984-1994 | 2510 | 8 | 18 | 0.12 | 0.49 | Big Meadow - NV |
| SNTL:454 | 41.66 | -115.32 | BSk | GRA | 1984-1997 | 2220 | 13 | 46 | 0.04 | 0.49 | Draw Creek - NV |
| SNTL:734 | 47.38 | -121.06 | Dsc | WSA | 1984-1998 | 1322 | 4 | 23 | 0.04 | 0.49 | Sasse Ridge - WA |
| SNTL:735 | 46.47 | -114.63 | Dsc | ENF | 1983-1988 | 1886 | 7 | 251 | 0.04 | 0.49 | Savage Pass - ID |
| SNTL:356 | 38.61 | -119.92 | Dsc | SAV | 1981-1991 | 2458 | 3 | 316 | 0.04 | 0.49 | Blue Lakes - CA |
| SNTL:463 | 38.85 | -120.08 | Dsb | WSA | 1981-1993 | 2332 | 19 | 182 | 0.42 | 0.49 | Echo Peak - CA |
| SNTL:724 | 39.00 | -120.13 | Dsb | ENF | 1981-1994 | 2322 | 8 | 37 | 0.08 | 0.49 | Rubicon #2 - CA |
| SNTL:834 | 39.30 | -120.18 | Dsb | ENF | 1981-1994 | 1983 | 6 | 52 | 0.25 | 0.49 | Truckee #2 - CA |
| SNTL:445 | 41.97 | -118.19 | BSk | GRA | 1981-1992 | 1908 | 18 | 135 | 0.12 | 0.49 | Disaster Peak - NV |
| SNTL:652 | 39.32 | -119.89 | Dsc | GRA | 1981-1990 | 2682 | 18 | 343 | 0.04 | 0.49 | Mt Rose Ski Area - NV |
| SNTL:849 | 39.13 | -114.96 | BSk | GRA | 1981-1991 | 2804 | 21 | 234 | 0.04 | 0.49 | Ward Mountain - NV |
| SNTL:784 | 39.19 | -120.27 | Dsb | GRA | 1980-1991 | 2442 | 8 | 117 | 0.17 | 0.49 | Squaw Valley G.C. - CA |
| SNTL:848 | 39.14 | -120.22 | Dsb | ENF | 1980-1992 | 2055 | 3 | 128 | 3.36 | 0.49 | Ward Creek #3 - CA |
| SNTL:531 | 39.36 | -106.06 | ET | GRA | 1980-1990 | 3474 | 7 | 225 | 0.04 | 0.98 | Hoosier Pass - CO |
| SNTL:688 | 40.40 | -105.85 | Dfc | GRA | 1980-1985 | 2752 | 2 | 79 | 63.23 | 0.49 | Phantom Valley - CO |
| SNTL:312 | 44.30 | -115.23 | Dsc | WSA | 1980-1988 | 2145 | 1 | 48 | 0.04 | 0.49 | Banner Summit - ID |
| SNTL:424 | 44.44 | -111.99 | Dfc | WSA | 1980-1988 | 2103 | 5 | 46 | 0.19 | 0.49 | Crab Creek - ID |
| SNTL:550 | 44.05 | -115.44 | Dsc | WSA | 1980-1998 | 2154 | 7 | 323 | 0.04 | 0.49 | Jackson Peak - ID |
| SNTL:752 | 46.95 | -116.34 | Dsb | WSA | 1980-1981 | 975 | 2 | 42 | 0.69 | 0.49 | Sherwin - ID |
| SNTL:395 | 43.23 | -121.81 | Dsb | WSA | 1980-1992 | 1478 | 1 | 104 | 0.04 | 0.98 | Chemult Alternate - OR |
| SNTL:529 | 43.67 | -122.57 | Csb | ENF | 1980-1985 | 1502 | 3 | 267 | 0.04 | 0.98 | Holland Meadows - OR |
| SNTL:647 | 45.27 | -117.69 | Dsc | ENF | 1980-1992 | 1755 | 5 | 43 | 0.04 | 0.98 | Moss Springs - OR |
| SNTL:653 | 45.27 | -117.17 | Dsc | SAV | 1980-1992 | 2410 | 9 | 16 | 0.04 | 0.98 | Mt. Howard - OR |
| SNTL:706 | 42.32 | -120.83 | Dsb | SAV | 1980-1994 | 1743 | 3 | 278 | 0.67 | 0.98 | Quartz Mountain - OR |
| SNTL:721 | 44.01 | -118.84 | BSk | GRA | 1980-1999 | 1612 | 1 | 208 | 0.69 | 0.20 | Rock Springs - OR |
| SNTL:729 | 43.61 | -122.12 | Dsb | ENF | 1980-1985 | 1286 | 1 | 189 | 0.12 | 0.49 | Salt Creek Falls - OR |
| SNTL:756 | 42.96 | -121.18 | Dsb | WSA | 1980-1994 | 1749 | 3 | 39 | 0.04 | 0.20 | Silver Creek - OR |
| SNTL:557 | 38.48 | -112.39 | Dsc | SAV | 1980-1992 | 2773 | 14 | 352 | 0.84 | 0.49 | Kimberly Mine - UT |
| SNTL:679 | 46.78 | -121.75 | Dfc | ENF | 1980-1983 | 1563 | 9 | 216 | 0.15 | 0.49 | Paradise - WA |
| SNTL:711 | 48.52 | -120.74 | Dsc | ENF | 1980-1987 | 1490 | 19 | 41 | 0.04 | 0.49 | Rainy Pass - WA |
| SNTL:824 | 46.12 | -117.85 | Dsb | WSA | 1980-1994 | 1685 | 2 | 30 | 0.04 | 0.49 | Touchet - WA |
| SNTL:508 | 38.85 | -119.94 | Dsb | WSA | 1979-1992 | 2359 | 8 | 10 | 0.08 | 0.49 | Hagans Meadow - CA |
| SNTL:539 | 39.45 | -120.29 | Dsb | WET | 1979-1991 | 2127 | 0 | 130 | 0.08 | 0.49 | Independence Camp - CA |
| SNTL:541 | 39.43 | -120.31 | Dsb | WSA | 1979-1992 | 2541 | 12 | 48 | 0.04 | 0.49 | Independence Lake - CA |
| SNTL:587 | 38.44 | -119.37 | Dsb | GRA | 1979-1990 | 2819 | 0 | 300 | 9.89 | 0.49 | Lobdell Lake - CA |



Continuation of Table A4

| Station | Lat. | Lon | Climate | Biome | Period | Elev. | Slop. | Asp | Au | SM sens. | Site name |
|---------|------|-----|---------|-------|--------|-------|-------|-----|-----|----------|-----------|
| SNTL:771 | 38.31 | -119.60 | Dsc | SAV | 1979-1991 | 2673 | 12 | 28 | 0.04 | 0.49 | Sonora Pass - CA |
| SNTL:846 | 38.07 | -119.23 | Dsc | GRA | 1979-1991 | 2878 | 8 | 152 | 0.04 | 0.49 | Virginia Lakes Ridge - CA |
| SNTL:682 | 39.05 | -107.87 | Dfc | SAV | 1979-1984 | 3035 | 5 | 260 | 0.17 | 0.49 | Park Reservoir - CO |
| SNTL:484 | 42.05 | -111.60 | Dfc | GRA | 1979-1992 | 2481 | 5 | 271 | 0.28 | 0.49 | Franklin Basin - ID |
| SNTL:490 | 43.87 | -114.71 | Dsc | SAV | 1979-1984 | 2676 | 7 | 107 | 0.04 | 0.49 | Galena Summit - ID |
| SNTL:594 | 47.46 | -115.70 | Dsb | WSA | 1979-1988 | 1581 | 10 | 10 | 0.04 | 0.49 | Lookout - ID |
| SNTL:645 | 48.06 | -116.23 | Dsc | ENF | 1979-1984 | 1603 | 8 | 202 | 0.07 | 0.49 | Mosquito Ridge - ID |
| SNTL:749 | 43.21 | -111.69 | Dfb | MF | 1979-1988 | 2026 | 10 | 39 | 0.08 | 0.49 | Sheep Mtn. - ID |
| SNTL:770 | 42.95 | -111.36 | Dfb | WSA | 1979-1989 | 2072 | 9 | 36 | 0.20 | 0.49 | Somsen Ranch - ID |
| SNTL:774 | 42.76 | -116.90 | Dsb | GRA | 1979-1988 | 1981 | 13 | 37 | 0.28 | 0.49 | South Mtn. - ID |
| SNTL:532 | 36.72 | -106.26 | Dfc | GRA | 1979-1982 | 3048 | 1 | 63 | 0.04 | 0.49 | Hopewell - NM |
| SNTL:615 | 39.16 | -119.90 | Dsb | WSA | 1979-1992 | 2403 | 9 | 41 | 0.04 | 0.49 | Marlette Lake - NV |
| SNTL:344 | 42.41 | -122.27 | Dsc | ENF | 1979-1992 | 1609 | 4 | 125 | 0.16 | 0.98 | Billie Creek Divide - OR |
| SNTL:361 | 44.83 | -118.19 | Dsc | WSA | 1979-1987 | 1783 | 19 | 164 | 0.08 | 0.49 | Bourne - OR |
| SNTL:523 | 45.70 | -118.11 | Dsb | ENF | 1979-1991 | 1499 | 7 | 12 | 0.15 | 0.98 | High Ridge - OR |
| SNTL:759 | 42.75 | -118.69 | Dsb | GRA | 1979-1998 | 2130 | 2 | 277 | 0.04 | 0.98 | Silvies - OR |
| SNTL:821 | 44.66 | -118.43 | Dsb | WSA | 1979-1991 | 1569 | 6 | 334 | 0.04 | 0.49 | Tipton - OR |
| SNTL:339 | 38.30 | -112.36 | Dfc | SAV | 1979-1992 | 3154 | 2 | 311 | 0.21 | 0.49 | Big Flat - UT |
| SNTL:364 | 38.51 | -112.02 | Dfc | GRA | 1979-1992 | 3003 | 1 | 148 | 0.04 | 0.49 | Box Creek - UT |
| SNTL:455 | 41.41 | -111.54 | Dfc | GRA | 1979-1992 | 2530 | 1 | 90 | 0.12 | 0.49 | Dry Bread Pond - UT |
| SNTL:495 | 38.80 | -111.68 | BSk | GRA | 1979-1992 | 2421 | 6 | 282 | 0.04 | 0.49 | Gooseberry RS - UT |
| SNTL:832 | 47.23 | -120.29 | Dsc | SAV | 1979-1993 | 1670 | 4 | 55 | 0.58 | 0.20 | Trough - WA |
| SNTL:358 | 44.68 | -107.58 | Dfc | GRA | 1979-1986 | 2849 | 6 | 301 | 0.11 | 0.98 | Bone Springs Div - WY |
| SNTL:613 | 48.80 | -113.67 | Dfc | WSA | 1977-1993 | 1493 | 16 | 172 | 0.04 | 0.20 | Many Glacier - MT |
| SNTL:781 | 46.78 | -110.62 | Dfc | WSA | 1967-1972 | 2468 | 4 | 339 | 0.04 | 0.98 | Spur Park - MT |
| SNTL:554 | 42.27 | -110.80 | Dfc | GRA | 1964-1965 | 2493 | 6 | 122 | 0.20 | 0.49 | Kelley R.S. - WY |
| SNTL:342 | 42.65 | -109.26 | Dfc | GRA | 1963-1977 | 2767 | 3 | 251 | 0.87 | 0.49 | Big Sandy Opening - WY |
| SNTL:822 | 43.75 | -110.06 | Dfc | GRA | 1961-1974 | 2919 | 5 | 199 | 0.23 | 0.49 | Togwotee Pass - WY |
| SCAN:2160 | 41.78 | -113.82 | BSk | GRA | 2010-2014 | 1778 | 2 | 172 | 0.16 | 0.98 | Grouse Creek - UT |
| SCAN:2149 | 37.78 | -118.42 | BWk | GRA | 2009-2014 | 1884 | 4 | 249 | 0.13 | 0.98 | Marble Creek - CA |
| SCAN:2142 | 36.37 | -115.78 | BSk | OSH | 2008-2015 | 2391 | 8 | 201 | 0.04 | 0.98 | Trough Springs - NV |
| SCAN:2074 | 42.02 | -121.39 | BSk | CRO | 2003-2014 | 1247 | 0 | 198 | 0.28 | 0.98 | Lynhart Ranch - OR |



Table A5: GSIM hydrometric stations used in the streamflow performance evaluation. Where Lat. is latitude (°), Lon. is longitude (°), Climate refers to the three largest Köppen-Geiger climate zones in the watershed, Biome refers to the three largest biomes in the watershed, as defined by the IGBP (Table A1) (Friedl et al., 2019). Elev. is the elevation range in masl. Area is the watershed area (km²) and River lists the river name and country.

| Station | Lat | Lon | Climate | Biome | Period | Elev | Area | River |
|---|---|---|---|---|---|---|---|---|
| CA_0000351 | 48.52 | -89.18 | Dfb: 98.9%; Dfc: 1.1% | MF: 56.8%; WSA: 37.8%; DBF: 5.4% | 1980-2014 | 341-514 | 104.70 | North Current River - CA |
| CA_0000373 | 48.85 | -86.61 | Dfc: 77.4%; Dfb: 22.6% | MF: 83.1%; WSA: 10.1%; ENF: 6.7% | 1980-2014 | 211-536 | 1324.17 | Little Pic River - CA |
| CA_0000391 | 47.00 | -84.52 | Dfb: 100% | MF: 56%; DBF: 39.5%; WSA: 3.7% | 1980-2014 | 239-601 | 1227.64 | Batchawana River - CA |
| CA_0000395 | 47.06 | -84.43 | Dfb: 100% | DBF: 100% | 1980-2014 | 370-521 | 10.36 | Norberg Creek - CA |
| CA_0000396 | 47.05 | -84.41 | Dfb: 100% | DBF: 100% | 1980-2014 | 373-521 | 8.03 | Norberg Creek - CA |
| CA_0000397 | 47.04 | -84.41 | Dfb: 100% | DBF: 100% | 1980-2014 | 387-521 | 5.05 | Norberg Creek - CA |
| CA_0001119 | 46.37 | -74.50 | Dfb: 52.3%; Dfc: 47.7% | MF: 100% | 1980-2013 | 447-732 | 39.90 | Saintlouis Ruisseau A 03 Km De La River - CA |
| CA_0001273 | 46.44 | -73.46 | Dfb: 97.2%; Dfc: 2.8% | MF: 96.2%; WSA: 3.2%; DBF: 0.6% | 1980-2013 | 275-663 | 186.00 | Mastigouche River - CA |
| CA_0001411 | 47.26 | -71.14 | Dfc: 100% | MF: 89.2%; WSA: 9.6%; ENF: 1.2% | 1980-2013 | 581-1065 | 269.00 | Montmorency River - CA |
| CA_0001418 | 47.27 | -71.14 | Dfc: 100% | MF: 85.7%; WSA: 14.3% | 1980-2013 | 581-904 | 9.17 | Eaux Volees Ruisseau Des Pres De La River - CA |
| CA_0001423 | 47.12 | -70.82 | Dfc: 88.2%; Dfb: 11.8% | MF: 76.2%; WSA: 15.5%; DBF: 4.5% | 1980-1994 | 288-1106 | 974.00 | Sainteanne Du Nord River - CA |
| CA_0001527 | 49.88 | -70.93 | Dfc: 100% | WSA: 53.3%; ENF: 31.6%; SAV: 8.5% | 1980-2007 | 248-975 | 1717.00 | Manouane River - CA |
| CA_0001556 | 49.33 | -70.98 | Dfc: 100% | MF: 71%; ENF: 23.7%; WSA: 5.4% | 1980-1994 | 383-603 | 277.00 | Shipshaw River - CA |
| CA_0001566 | 47.94 | -71.38 | Dfc: 100% | MF: 81.2%; ENF: 9.1%; WSA: 8.8% | 1980-2013 | 559-1037 | 495.00 | Pikauba River - CA |
| CA_0001679 | 48.81 | -57.78 | Dfc: 100% | WSA: 68.8%; SAV: 31.2% | 1980-2014 | 327-584 | 12.90 | Copper Pond Brook - CA |
| CA_0002036 | 49.66 | -114.13 | Dfc: 75%; Dfb: 24.6%; ET: 0.4% | GRA: 65.4%; ENF: 19.9%; WSA: 12.6% | 1980-1993 | 1201-2293 | 144.00 | Todd Creek - CA |
| CA_0002038 | 49.60 | -114.41 | Dfc: 97.3%; ET: 2.7% | WSA: 37.5%; ENF: 37.2%; GRA: 23.4% | 1980-2012 | 1282-2458 | 402.70 | Crowsnest River - CA |
| CA_0002041 | 49.47 | -114.13 | Dfc: 93.4%; Dfb: 6.3%; ET: 0.3% | WSA: 32.4%; ENF: 26.5%; GRA: 24% | 1980-2012 | 1236-2356 | 179.00 | Mill Creek - CA |
| CA_0002047 | 49.49 | -114.14 | Dfc: 93.9%; Dfb: 5.1%; ET: 0.9% | ENF: 31.7%; GRA: 31.2%; WSA: 26.4% | 1980-2013 | 1198-2498 | 820.70 | Castle River - CA |
| CA_0002048 | 49.81 | -114.18 | Dfc: 91.2%; ET: 8.6%; Dfb: 0.2% | WSA: 44.5%; ENF: 36.6%; GRA: 17.4% | 1980-2008 | 1280-2757 | 1446.10 | Oldman River - CA |
| CA_0002051 | 49.90 | -114.43 | Dfc: 89.4%; ET: 10.6% | WSA: 42.6%; ENF: 40.4%; GRA: 16.2% | 1980-1995 | 1505-2669 | 143.00 | Dutch Creek - CA |
| CA_0002052 | 49.84 | -114.42 | Dfc: 96.7%; ET: 3.3% | ENF: 53.1%; WSA: 34.6%; GRA: 11.7% | 1980-2013 | 1466-2511 | 217.60 | Racehorse Creek - CA |
| CA_0002053 | 49.40 | -114.34 | Dfc: 99.1%; ET: 0.9% | ENF: 45.6%; SAV: 18.5%; GRA: 18.4% | 1980-2014 | 1354-2420 | 375.30 | Castle River - CA |



Continuation of Table A5

| Station | Lat | Lon | Climate | Biome | Period | Elev | Area | River |
|---|---|---|---|---|---|---|---|---|
| CA_0002055 | 49.60 | -114.40 | Dfc: 99.1%; ET: 0.9% | WSA: 40.4%; ENF: 32.3%; GRA: 24.2% | 1980-2014 | 1335-2322 | 63.30 | Gold Creek - CA |
| CA_0002057 | 49.31 | -114.08 | Dfc: 100% | GRA: 50%; SAV: 28.9%; WSA: 18.4% | 1980-2012 | 1589-2235 | 24.00 | Pincher Creek - CA |
| CA_0002059 | 49.76 | -114.24 | Dfc: 99.3%; ET: 0.7% | GRA: 46.1%; ENF: 33%; WSA: 20.9% | 1980-2013 | 1406-2293 | 74.00 | Todd Creek - CA |
| CA_0002064 | 49.64 | -113.80 | Dfc: 52.1%; Dfb: 47.9% | GRA: 61.4%; ENF: 16.4%; WSA: 11.4% | 1980-2012 | 1102-1770 | 255.80 | Beaver Creek - CA |
| CA_0002072 | 50.02 | -113.71 | Dfc: 72.9%; Dfb: 27%; ET: 0.1% | GRA: 54.2%; WSA: 21.2%; ENF: 10.8% | 1980-2013 | 1001-2368 | 1180.60 | Willow Creek - CA |
| CA_0002079 | 50.20 | -114.21 | Dfc: 99.7%; ET: 0.3% | WSA: 46.3%; ENF: 36.5%; GRA: 7.8% | 1980-1995 | 1315-2229 | 162.00 | Willow Creek - CA |
| CA_0002087 | 50.14 | -113.94 | Dfc: 99.4%; Dfb: 0.4%; ET: 0.1% | GRA: 45.2%; WSA: 31.9%; ENF: 16.6% | 1980-1997 | 1172-2368 | 727.00 | Willow Creek - CA |
| CA_0002088 | 50.24 | -114.35 | Dfc: 99.2%; ET: 0.8% | ENF: 54.2%; WSA: 30.8%; GRA: 9.3% | 1980-2014 | 1478-2229 | 65.30 | Willow Creek - CA |
| CA_0002089 | 50.13 | -113.85 | Dfc: 96%; Dfb: 3.9%; ET: 0.1% | GRA: 49.8%; WSA: 29.3%; ENF: 15.2% | 1980-2012 | 1079-2368 | 832.90 | Willow Creek - CA |
| CA_0002091 | 50.12 | -113.75 | Dfb: 93.8%; Dfc: 6.2% | GRA: 52%; CRO: 48% | 1980-2013 | 1038-1381 | 85.70 | Na - CA |
| CA_0002133 | 49.11 | -113.84 | Dfc: 93.1%; ET: 4%; Dfb: 2.9% | GRA: 39.3%; WSA: 24.4%; ENF: 16.3% | 1980-2012 | 1274-2699 | 612.70 | Waterton River - CA |
| CA_0002135 | 49.10 | -113.70 | Dfc: 94.8%; ET: 4.9%; Dfb: 0.4% | GRA: 41.8%; WSA: 22.3%; ENF: 19.5% | 1980-2014 | 1340-2940 | 319.20 | Belly River - CA |
| CA_0002166 | 49.23 | -113.96 | Dfc: 92.1%; ET: 7.9% | GRA: 71.6%; ENF: 10.8%; SAV: 9.5% | 1980-2013 | 1460-2483 | 47.90 | Yarrow Creek - CA |
| CA_0002198 | 48.83 | -113.52 | Dfc: 93%; ET: 7% | GRA: 65.7%; WSA: 15%; ENF: 9.8% | 1980-1994 | 1454-2757 | 168.00 | Swiftcurrent Creek - CA |
| CA_0002206 | 49.03 | -113.54 | Dfc: 100% | WSA: 44.4%; ENF: 34%; GRA: 14.6% | 1980-1992 | 1436-2267 | 93.80 | Lee Creek - CA |
| CA_0002264 | 49.74 | -110.04 | Dfb: 74.1%; Dfc: 25.9% | CRO: 88%; GRA: 5.4%; ENF: 2.2% | 1980-1993 | 998-1390 | 75.10 | Mackay Creek - CA |
| CA_0002279 | 51.43 | -116.19 | ET: 67.8%; Dfc: 32.2% | GRA: 29.1%; BSV: 26.5%; ENF: 17.2% | 1980-2014 | 1541-2978 | 422.40 | Bow River - CA |
| CA_0002280 | 51.43 | -116.17 | ET: 80.2%; Dfc: 19.8% | GRA: 46.7%; BSV: 20.9%; ENF: 12.8% | 1980-2014 | 1618-3093 | 306.10 | Pipestone River - CA |
| CA_0002284 | 51.24 | -115.84 | ET: 78%; Dfc: 22% | GRA: 49.5%; WSA: 17.3%; BSV: 16.8% | 1980-1996 | 1449-2904 | 124.00 | Johnston Creek - CA |
| CA_0002292 | 51.10 | -115.67 | ET: 69.7%; Dfc: 30.3% | GRA: 53.9%; WSA: 33.9%; ENF: 7.8% | 1980-1996 | 1622-2762 | 109.00 | Brewster Creek - CA |
| CA_0002293 | 51.22 | -115.81 | ET: 69.2%; Dfc: 30.8% | GRA: 33.3%; WSA: 32.9%; ENF: 17.3% | 1980-1996 | 1449-3155 | 147.00 | Redearth Creek - CA |
| CA_0002294 | 51.16 | -115.55 | ET: 50.1%; Dfc: 49.9% | GRA: 44.1%; WSA: 36.5%; ENF: 11.4% | 1980-2014 | 1432-3079 | 750.60 | Spray River - CA |
| CA_0002299 | 51.06 | -115.43 | Dfc: 65.3%; ET: 34.7% | WSA: 62.7%; GRA: 28.8%; ENF: 8.5% | 1980-2012 | 1643-2693 | 40.90 | Goat Creek - CA |
| CA_0002303 | 51.29 | -115.53 | ET: 77%; Dfc: 23% | GRA: 55.5%; WSA: 20.9%; ENF: 14.8% | 1980-1996 | 1571-3043 | 454.00 | Cascade River - CA |
| CA_0002314 | 50.70 | -115.12 | ET: 56.1%; Dfc: 43.9% | GRA: 37.1%; ENF: 24.8%; BSV: 20% | 1980-2012 | 1621-3026 | 362.00 | Kananaskis River - CA |



Continuation of Table A5

| Station | Lat | Lon | Climate | Biome | Period | Elev | Area | River |
|---|---|---|---|---|---|---|---|---|
| CA_0002319 | 50.79 | -115.30 | ET: 70.5%; Dfc: 29.5% | GRA: 59.5%; BSV: 21.6%; WSA: 18.9% | 1980-1992 | 1898-2876 | 29.00 | Mud Lake - CA |
| CA_0002321 | 50.95 | -115.15 | Dfc: 60%; ET: 40% | ENF: 50%; WSA: 31.2%; GRA: 18.8% | 1980-2012 | 1711-2473 | 9.10 | Marmot Creek - CA |
| CA_0002328 | 51.04 | -115.03 | Dfc: 53.9%; ET: 46.1% | GRA: 36.7%; ENF: 26.9%; WSA: 22.6% | 1980-2012 | 1342-3026 | 899.00 | Kananaskis River - CA |
| CA_0002330 | 51.30 | -115.18 | ET: 73.4%; Dfc: 26.6% | GRA: 67.3%; BSV: 11.9%; WSA: 11% | 1980-1993 | 1625-2834 | 211.00 | Ghost River - CA |
| CA_0002334 | 51.28 | -114.84 | Dfc: 82.9%; ET: 17.1% | ENF: 51.7%; WSA: 22.7%; GRA: 20.4% | 1980-2012 | 1292-2764 | 332.50 | Waiparous Creek - CA |
| CA_0002336 | 51.27 | -114.93 | Dfc: 81.9%; ET: 18.1% | ENF: 36.3%; GRA: 31.7%; WSA: 26.5% | 1980-2012 | 1328-2660 | 484.50 | Ghost River - CA |
| CA_0002345 | 51.00 | -114.94 | Dfc: 91%; ET: 9% | ENF: 79.3%; GRA: 12.1%; WSA: 8.6% | 1980-2012 | 1654-2320 | 36.90 | Jumpingpound Creek - CA |
| CA_0002355 | 50.86 | -114.79 | ET: 50.3%; Dfc: 49.7% | GRA: 54%; WSA: 22.8%; ENF: 18.2% | 1980-1995 | 1534-2911 | 437.00 | Elbow River - CA |
| CA_0002356 | 50.79 | -114.92 | ET: 71.9%; Dfc: 28.1% | GRA: 74.3%; WSA: 20.6%; BSV: 3.3% | 1980-1995 | 1757-2778 | 129.00 | Little Elbow River - CA |
| CA_0002375 | 50.41 | -114.50 | Dfc: 76%; ET: 24% | ENF: 46.9%; WSA: 28%; GRA: 21.6% | 1980-2012 | 1426-2939 | 773.60 | Highwood River - CA |
| CA_0002378 | 50.29 | -114.59 | Dfc: 83.7%; ET: 16.3% | ENF: 53.3%; WSA: 35.2%; GRA: 11.5% | 1980-2012 | 1680-2629 | 165.50 | Cataract Creek - CA |
| CA_0002379 | 50.47 | -114.21 | Dfc: 98.1%; ET: 1.9% | GRA: 29.5%; WSA: 25.7%; CRO: 21.1% | 1980-2014 | 1208-2353 | 231.90 | Pekisko Creek - CA |
| CA_0002382 | 50.48 | -114.43 | Dfc: 78.9%; ET: 21.1% | GRA: 41.7%; ENF: 28%; WSA: 19.7% | 1980-2012 | 1365-2606 | 137.40 | Trap Creek - CA |
| CA_0002424 | 51.66 | -115.13 | Dfc: 100% | WSA: 66.7%; GRA: 33.3% | 1980-1995 | 1432-1620 | 5.44 | Deer Creek - CA |
| CA_0002425 | 51.66 | -115.41 | ET: 70.4%; Dfc: 29.6% | GRA: 46.8%; WSA: 24.5%; BSV: 16.6% | 1980-2012 | 1513-3052 | 941.40 | Red Deer River - CA |
| CA_0002506 | 52.04 | -116.38 | ET: 81.9%; Dfc: 18.1% | GRA: 45.2%; BSV: 34.3%; WSA: 13.3% | 1980-1996 | 1406-3027 | 515.00 | Siffleur River - CA |
| CA_0002511 | 51.88 | -116.69 | ET: 73.2%; Dfc: 26.8% | BSV: 38.5%; GRA: 25.7%; WSA: 13.3% | 1980-2013 | 1635-2968 | 248.00 | Mistaya River - CA |
| CA_0002513 | 52.00 | -116.47 | ET: 69.3%; Dfc: 30.7% | GRA: 34.5%; BSV: 32.8%; SNO: 13.1% | 1980-2014 | 1355-3333 | 1923.20 | North Saskatchewan River - CA |
| CA_0002514 | 51.80 | -116.58 | ET: 91.5%; Dfc: 8.5% | BSV: 48.8%; GRA: 32.6%; WSA: 11.6% | 1980-2013 | 1728-2842 | 21.00 | Silverhorn Creek - CA |
| CA_0002517 | 51.99 | -115.43 | ET: 51.3%; Dfc: 48.7% | GRA: 34.4%; ENF: 30.9%; WSA: 22.3% | 1980-1992 | 1365-3136 | 1340.00 | Clearwater River - CA |
| CA_0002527 | 52.37 | -115.42 | Dfc: 66.5%; ET: 33.5% | ENF: 61.2%; GRA: 20.2%; WSA: 16% | 1980-2013 | 1083-2986 | 1853.60 | Ram River - CA |
| CA_0002531 | 52.28 | -116.00 | Dfc: 60.9%; ET: 39.1% | ENF: 67.9%; GRA: 21.3%; WSA: 10.2% | 1980-2013 | 1463-2596 | 347.30 | North Ram River - CA |
| CA_0002536 | 52.76 | -116.36 | Dfc: 98.4%; ET: 1.6% | ENF: 87.7%; WSA: 12%; SAV: 0.3% | 1980-2013 | 1316-2149 | 218.70 | Brown Creek - CA |
| CA_0002539 | 52.87 | -116.60 | Dfc: 59.6%; ET: 40.4% | ENF: 61.3%; GRA: 18.1%; WSA: 14.2% | 1980-1990 | 1341-2682 | 495.00 | Cardinal River - CA |
| CA_0002950 | 52.60 | -101.04 | Dfc: 91.5%; Dfb: 8.5% | WSA: 44.9%; MF: 29.2%; ENF: 18.5% | 1980-1993 | 316-762 | 170.00 | Bell River - CA |



Continuation of Table A5

| Station | Lat | Lon | Climate | Biome | Period | Elev | Area | River |
|---------|-----|-----|---------|-------|--------|------|------|-------|
| CA_0003014 | 50.48 | -99.48 | Dfb: 85.7%; Dfc: 14.3% | DBF: 44.4%; CRO: 16.7%; WSA: 16.7% | 1980-2014 | 469-704 | 9.20 | Pelican Creek - CA |
| CA_0003558 | 52.86 | -118.11 | ET: 61.1%; Dfc: 38.9% | ENF: 39.4%; GRA: 32.5%; WSA: 16% | 1980-2012 | 1051-2769 | 628.50 | Miette River - CA |
| CA_0003561 | 52.93 | -118.03 | ET: 73.9%; Dfc: 26.1% | GRA: 47.1%; ENF: 21.7%; BSV: 16.3% | 1980-1997 | 1032-3151 | 908.00 | Maligne River - CA |
| CA_0003562 | 52.22 | -117.23 | ET: 100% | SNO: 57.4%; BSV: 42.6% | 1980-2012 | 1943-3390 | 29.30 | Sunwapta River - CA |
| CA_0003564 | 52.72 | -117.92 | ET: 63.1%; Dfc: 36.9% | GRA: 31.9%; BSV: 29%; WSA: 14.9% | 1980-1996 | 1148-3094 | 598.00 | Whirlpool River - CA |
| CA_0003566 | 53.16 | -118.03 | ET: 68.1%; Dfc: 31.9% | GRA: 43.3%; ENF: 28.5%; WSA: 17.5% | 1980-1993 | 1118-2752 | 1580.00 | Snake Indian River - CA |
| CA_0003567 | 53.52 | -117.95 | Dfc: 56.8%; ET: 43.2% | ENF: 70.4%; WSA: 16.6%; GRA: 12.4% | 1980-2013 | 1270-2512 | 959.80 | Wildhay River - CA |
| CA_0003574 | 53.68 | -118.24 | Dfc: 54.9%; ET: 45.1% | ENF: 55.8%; WSA: 23.1%; GRA: 19.9% | 1980-2012 | 1414-2415 | 93.00 | Little Berland River - CA |
| CA_0003590 | 53.16 | -117.26 | Dfc: 100% | ENF: 100% | 1980-2014 | 1289-1627 | 25.90 | Wampus Creek - CA |
| CA_0003591 | 53.16 | -117.24 | Dfc: 100% | ENF: 100% | 1980-1990 | 1280-1616 | 14.00 | Deerlick Creek - CA |
| CA_0003592 | 53.15 | -117.23 | Dfc: 100% | ENF: 100% | 1980-1992 | 1309-1636 | 17.10 | Eunice Creek - CA |
| CA_0003598 | 53.08 | -117.20 | ET: 60%; Dfc: 40% | ENF: 42.5%; GRA: 36.4%; WSA: 13.9% | 1980-2012 | 1415-2611 | 329.60 | Mcleod River - CA |
| CA_0003600 | 53.25 | -117.36 | Dfc: 83.4%; ET: 16.6% | ENF: 79.6%; WSA: 10.5%; GRA: 9.6% | 1980-2012 | 1235-2366 | 384.00 | Gregg River - CA |
| CA_0003619 | 53.00 | -116.66 | Dfc: 100% | WSA: 92.6%; ENF: 7.4% | 1980-2013 | 1338-1555 | 102.70 | Lovett River - CA |
| CA_0003647 | 55.42 | -114.81 | Dfc: 100% | WSA: 88.9%; WET: 11.1% | 1980-2012 | 590-595 | 23.70 | Lily Creek - CA |
| CA_0003651 | 54.80 | -115.47 | Dfc: 100% | ENF: 52.9%; WSA: 43.3%; MF: 3.8% | 1980-2014 | 973-1348 | 155.10 | Swan River - CA |
| CA_0003714 | 57.19 | -124.90 | ET: 58.4%; Dfc: 27.5%; Dsc: 14.1% | ENF: 35.3%; GRA: 27.9%; WSA: 27.1% | 1980-2012 | 766-2661 | 1690.00 | Akie River - CA |
| CA_0003720 | 56.13 | -124.80 | Dfc: 59.7%; ET: 35.2%; Dsc: 5.1% | WSA: 48.6%; ENF: 30.4%; GRA: 18.8% | 1980-2013 | 809-2274 | 1950.00 | Osilinka River - CA |
| CA_0003728 | 54.53 | -122.61 | Dfc: 100% | ENF: 47.3%; MF: 37.5%; DBF: 7.5% | 1980-2012 | 760-1267 | 310.00 | Chuchinka Creek - CA |
| CA_0003732 | 55.95 | -122.66 | Dfc: 90.3%; ET: 9.7% | ENF: 79.8%; WSA: 12.6%; GRA: 7.5% | 1980-2013 | 769-1936 | 741.00 | Carbon Creek - CA |
| CA_0003742 | 55.54 | -121.60 | Dfc: 100% | ENF: 51%; DBF: 20.4%; MF: 20.4% | 1980-2014 | 650-1602 | 82.40 | Dickebusch Creek - CA |
| CA_0003743 | 55.15 | -120.92 | Dfc: 100% | ENF: 86.2%; MF: 6.9%; WSA: 6.9% | 1980-2001 | 946-1254 | 29.50 | Quality Creek - CA |
| CA_0003747 | 55.09 | -120.94 | Dfc: 97.8%; ET: 2.2% | ENF: 87.1%; WSA: 6.7%; GRA: 3% | 1980-2012 | 865-1922 | 486.00 | Flatbed Creek - CA |
| CA_0003749 | 55.70 | -121.63 | Dfc: 100% | ENF: 33.3%; MF: 31%; WSA: 31% | 1980-1998 | 747-1295 | 22.90 | Windrem Creek - CA |
| CA_0003753 | 56.27 | -120.95 | Dfc: 100% | WSA: 47.8%; SAV: 16.6%; GRA: 12.7% | 1980-1992 | 689-866 | 298.00 | Stoddart Creek - CA |
| CA_0003773 | 56.17 | -117.60 | Dfc: 100% | CRO: 100% | 1980-2011 | 589-653 | 8.10 | Na - CA |
| CA_0003779 | 53.93 | -118.82 | Dfc: 73.3%; ET: 26.7% | ENF: 62.5%; WSA: 25.2%; GRA: 11.3% | 1980-2014 | 1151-2581 | 702.90 | Muskeg River - CA |



Continuation of Table A5

| Station | Lat | Lon | Climate | Biome | Period | Elev | Area | River |
|---|---|---|---|---|---|---|---|---|
| CA_0003780 | 54.52 | -118.96 | Dfc: 100% | ENF: 53.4%; WSA: 28.4%; MF: 17.4% | 1980-2013 | 825-1599 | 842.20 | Cutbank River - CA |
| CA_0003784 | 54.84 | -119.39 | Dfc: 100% | MF: 72.5%; WSA: 16.4%; ENF: 6.7% | 1980-2011 | 720-1285 | 493.80 | Pinto Creek - CA |
| CA_0003927 | 61.19 | -136.98 | Dfc: 66.7%; Dsc: 33.3% | WSA: 85.7%; SAV: 14.3% | 1980-2014 | 945-1044 | 190.00 | Giltana Creek - CA |
| CA_0003932 | 60.11 | -136.93 | ET: 81.5%; Dsc: 18.5% | GRA: 65.7%; OSH: 18.6%; WSA: 10.9% | 1980-2014 | 726-1934 | 375.00 | Takhanne River - CA |
| CA_0003933 | 60.12 | -137.08 | ET: 72.2%; Dsc: 27.8% | GRA: 54.6%; OSH: 12.8%; BSV: 10.7% | 1980-2014 | 592-2035 | 1750.00 | Tatshenshini River - CA |
| CA_0003943 | 57.22 | -129.11 | ET: 96.5%; Dsc: 3.5% | GRA: 40.4%; BSV: 23.4%; SNO: 21.3% | 1980-1996 | 1212-2015 | 29.20 | Unnamed Creek - CA |
| CA_0003944 | 57.25 | -129.05 | ET: 100% | GRA: 72.7%; BSV: 27.3% | 1980-1996 | 1499-1982 | 16.60 | Klappan River - CA |
| CA_0003951 | 57.53 | -130.18 | ET: 63.8%; Dsc: 36.2% | GRA: 39.2%; ENF: 36.3%; WSA: 14.1% | 1980-1996 | 821-2330 | 1250.00 | Iskut River - CA |
| CA_0003953 | 57.04 | -130.40 | ET: 69.5%; Dsc: 20.4%; Dfc: 10.1% | GRA: 37%; SNO: 25.5%; BSV: 17.2% | 1980-1995 | 427-2284 | 844.00 | More Creek - CA |
| CA_0003954 | 56.92 | -130.72 | ET: 69.5%; Dfc: 30.5% | SNO: 64.8%; GRA: 15.4%; ENF: 10.1% | 1980-1994 | 486-2060 | 311.00 | Forrest Kerr Creek - CA |
| CA_0003956 | 56.11 | -129.48 | ET: 50.2%; Dfc: 49.1%; Dsc: 0.7% | GRA: 46.9%; SNO: 20.6%; BSV: 11.9% | 1980-2012 | 342-2314 | 218.00 | Surprise Creek - CA |
| CA_0003960 | 56.97 | -129.47 | ET: 87.1%; Dsc: 12.9% | GRA: 58.5%; WSA: 15.2%; ENF: 14.7% | 1980-1996 | 991-1984 | 118.00 | Craven Creek - CA |
| CA_0003963 | 56.29 | -129.23 | Dsc: 70%; Dfc: 15%; ET: 15% | ENF: 75%; GRA: 25% | 1980-2011 | 532-1473 | 8.64 | Kelly Creek - CA |
| CA_0003978 | 56.04 | -129.93 | ET: 53%; Dfc: 46.5%; Dfb: 0.5% | SNO: 43.2%; GRA: 24.3%; ENF: 21.2% | 1980-1999 | 87-2317 | 350.00 | Bear River - CA |
| CA_0003979 | 56.35 | -130.69 | ET: 50.6%; Dfc: 46.4%; Dsc: 2.2% | SNO: 28.1%; ENF: 27.8%; GRA: 26% | 1980-1996 | 123-2366 | 1480.00 | Unuk River - CA |
| CA_0003982 | 55.43 | -127.72 | Dsc: 78.3%; ET: 12.3%; Dsb: 5.6% | ENF: 72.6%; GRA: 11.6%; MF: 10.4% | 1980-2014 | 286-1839 | 1880.00 | Kispiox River - CA |
| CA_0003984 | 55.46 | -127.85 | Dsc: 81.1%; ET: 18.9% | ENF: 79.4%; GRA: 17.6%; BSV: 2.9% | 1980-2014 | 579-1717 | 19.10 | Compass Creek - CA |
| CA_0003987 | 54.41 | -125.43 | Dsc: 82%; Dfc: 18% | ENF: 61.3%; WSA: 34.2%; GRA: 4.2% | 1980-2013 | 863-1574 | 808.00 | Pinkut Creek - CA |
| CA_0003993 | 53.93 | -127.45 | Dfc: 65.8%; ET: 34.2%; Dsc: 0.1% | ENF: 43.7%; GRA: 23.1%; WSA: 15.1% | 1980-2013 | 912-2243 | 732.00 | Nanika River - CA |
| CA_0003994 | 54.12 | -127.43 | Dfc: 59.9%; ET: 33.9%; Dsc: 6.2% | ENF: 40.6%; GRA: 23.3%; WSA: 17.5% | 1980-2013 | 761-2469 | 1900.00 | Morice River - CA |
| CA_0004004 | 54.65 | -127.12 | Dsc: 72.5%; ET: 27.5% | ENF: 66.7%; GRA: 11.7%; WSA: 11.3% | 1980-2012 | 674-2116 | 125.00 | Goathorn Creek - CA |
| CA_0004007 | 54.81 | -127.20 | Dsc: 64.5%; ET: 35.5% | ENF: 40.7%; GRA: 29.6%; BSV: 18.5% | 1980-2012 | 560-2114 | 13.20 | Simpson Creek - CA |
| CA_0004008 | 54.40 | -126.65 | Dsc: 86.3%; Dfc: 12.4%; ET: 1.4% | ENF: 54.2%; WSA: 41.7%; GRA: 2.4% | 1980-2015 | 611-1550 | 565.00 | Buck Creek - CA |
| CA_0004009 | 54.80 | -127.11 | Dsc: 71.1%; ET: 16.5%; Dfc: 12.4% | ENF: 66.7%; WSA: 23.2%; GRA: 6% | 1980-1996 | 540-1859 | 256.00 | Canyon Creek - CA |
| CA_0004015 | 54.61 | -127.50 | ET: 45.5%; Dsc: 42.9%; Dfc: 11.6% | ENF: 52.3%; GRA: 20.4%; WSA: 10.1% | 1980-2014 | 710-2355 | 367.00 | Telkwa River - CA |



Continuation of Table A5

| Station | Lat | Lon | Climate | Biome | Period | Elev | Area | River |
|---------|-----|-----|---------|-------|--------|------|------|-------|
| CA_0004020 | 55.30 | -127.62 | Dsc: 91.2%; Dfc: 5.9%; Dsb: 2.9% | ENF: 89.3%; MF: 10.7% | 1980-2014 | 438-1418 | 21.20 | Two Mile Creek - CA |
| CA_0004022 | 55.23 | -127.57 | Dsc: 59.1%; ET: 40.9% | GRA: 47.1%; ENF: 35.3%; BSV: 17.6% | 1980-1996 | 473-1949 | 10.80 | Station Creek - CA |
| CA_0004027 | 54.78 | -127.47 | Dsc: 66.7%; ET: 33.3% | ENF: 52.9%; WSA: 41.2%; GRA: 5.9% | 1980-2013 | 954-1721 | 9.97 | M3 Creek - CA |
| CA_0004042 | 54.99 | -128.81 | Dfc: 87.5%; Dsc: 6.2%; ET: 6.2% | ENF: 78.6%; WSA: 14.3%; GRA: 7.1% | 1980-2013 | 563-1534 | 7.16 | Clarence Creek - CA |
| CA_0004058 | 52.36 | -126.81 | Dfc: 65.1%; ET: 25.3%; Dfb: 9.6% | ENF: 58.2%; GRA: 19.2%; SNO: 8.9% | 1980-1996 | 186-1984 | 92.50 | Clayton Falls Creek - CA |
| CA_0004069 | 53.56 | -127.95 | Dfc: 56.3%; ET: 35.3%; Dfb: 8.4% | GRA: 34.9%; ENF: 23.9%; SNO: 22.1% | 1980-2013 | 68-2144 | 556.00 | Kemano River - CA |
| CA_0004074 | 54.22 | -128.22 | Dfc: 95.8%; Dfb: 4.2% | ENF: 57.9%; GRA: 26.3%; WSA: 15.8% | 1980-2009 | 404-1495 | 12.70 | Kilometre 189 Creek - CA |
| CA_0004101 | 49.45 | -123.11 | Dfc: 50.3%; Dfb: 45.5%; Cfb: 4.2% | ENF: 96.9%; WSA: 2.3%; GRA: 0.8% | 1980-2003 | 310-1501 | 69.90 | Capilano River - CA |
| CA_0004102 | 49.45 | -123.10 | Dfb: 51.3%; Dfc: 46.1%; Cfb: 2.6% | ENF: 98.5%; WSA: 1.5% | 1980-2003 | 364-1503 | 41.40 | Eastcap Creek - CA |
| CA_0004135 | 50.11 | -123.43 | Dfc: 57.4%; ET: 29.1%; Dfb: 13.5% | SNO: 26%; GRA: 25.3%; ENF: 25.2% | 1980-2014 | 283-2640 | 1200.00 | Elaho River - CA |
| CA_0004136 | 50.08 | -123.04 | Dfc: 64.1%; ET: 28.7%; Dfb: 4.4% | ENF: 38.1%; GRA: 23.2%; BSV: 21.5% | 1980-2012 | 694-2499 | 297.00 | Cheakamus River - CA |
| CA_0004139 | 49.52 | -123.00 | Dfc: 57%; Dfb: 41.3%; Cfb: 1.7% | ENF: 94.1%; GRA: 2.9%; WSA: 2.9% | 1980-2011 | 271-1471 | 63.00 | Seymour River - CA |
| CA_0004141 | 49.50 | -122.97 | Dfc: 51.7%; Dfb: 43%; Cfb: 5.3% | ENF: 95.4%; GRA: 2.3%; WSA: 2.3% | 1980-2012 | 233-1471 | 82.90 | Seymour River - CA |
| CA_0004149 | 49.79 | -123.42 | Dfc: 73.5%; Dfb: 21.8%; Cfb: 3.1% | ENF: 48.2%; GRA: 25.9%; BSV: 13.6% | 1980-2013 | 74-2301 | 147.00 | Clowhom River - CA |
| CA_0004159 | 51.37 | -124.76 | ET: 45.9%; Dsc: 33.5%; Dfc: 18.5% | ENF: 31.2%; BSV: 24.2%; WSA: 19.2% | 1980-1995 | 632-3051 | 1960.00 | Homathko River - CA |
| CA_0004161 | 51.41 | -124.93 | ET: 57.8%; Dfc: 20.2%; Dsc: 16.8% | BSV: 28.2%; GRA: 27%; ENF: 24.5% | 1980-1995 | 704-3659 | 1550.00 | Mosley Creek - CA |
| CA_0004162 | 51.67 | -124.41 | Dsc: 45.5%; Dfc: 28.6%; ET: 20.4% | WSA: 56.8%; ENF: 31%; BSV: 7.9% | 1980-2014 | 854-2615 | 486.00 | Homathko River - CA |
| CA_0004163 | 51.42 | -124.51 | ET: 42.5%; Dsc: 37%; Dfc: 18.1% | ENF: 34.1%; WSA: 22.7%; BSV: 22.6% | 1980-1995 | 817-3051 | 1550.00 | Homathko River - CA |
| CA_0004169 | 51.07 | -126.36 | Dfc: 78.6%; Dfb: 15.3%; Cfb: 5.1% | ENF: 47%; GRA: 27.7%; WSA: 12% | 1980-2010 | 158-1669 | 54.30 | Kippan Creek - CA |
| CA_0004219 | 48.64 | -124.29 | Cfb: 100% | ENF: 100% | 1980-2012 | 394-842 | 8.12 | Renfrew Creek - CA |
| CA_0004220 | 48.72 | -124.23 | Cfb: 58.2%; Csb: 40%; Dfc: 1.8% | ENF: 100% | 1980-2012 | 299-986 | 28.00 | Harris Creek - CA |
| CA_0004227 | 49.29 | -124.58 | Dfc: 47.6%; Csb: 32.3%; Dsc: 7.7% | ENF: 91%; WSA: 6.2%; GRA: 2.8% | 1980-2001 | 185-1603 | 135.00 | Little Qualicum River - CA |
| CA_0004257 | 49.02 | -124.19 | Csb: 36.9%; Dfc: 27.2%; Dsb: 18.4% | ENF: 100% | 1980-2014 | 344-1328 | 62.20 | Jump Creek - CA |
| CA_0004284 | 49.25 | -125.58 | Cfb: 54.3%; Cfc: 24.3%; Dfc: 21.4% | ENF: 98.3%; GRA: 1.7% | 1980-2013 | 75-1174 | 38.60 | Tofino Creek - CA |
| CA_0004289 | 49.06 | -124.13 | Csb: 31.4%; Dsb: 30.4%; Dfc: 15.7% | ENF: 100% | 1980-2014 | 283-1346 | 211.00 | South Nanaimo River - CA |



Continuation of Table A5

| Station | Lat | Lon | Climate | Biome | Period | Elev | Area | River |
|---|---|---|---|---|---|---|---|---|
| CA_0004301 | 49.82 | -125.99 | Dfc: 71.6%; Cfb: 25.7%; Cfc: 2.7% | ENF: 100% | 1980-2013 | 220-1419 | 60.00 | Heber River - CA |
| CA_0004324 | 49.86 | -125.81 | Dfc: 75.4%; Cfb: 16%; ET: 6.4% | ENF: 71.7%; WSA: 16.4%; GRA: 11.9% | 1980-2013 | 301-1845 | 132.00 | Elk River - CA |
| CA_0004326 | 49.93 | -125.51 | Cfb: 44.2%; Dfb: 29.2%; Dfc: 26.6% | ENF: 83%; WSA: 14.8%; GRA: 2.2% | 1980-2013 | 311-1513 | 84.20 | Quinsam River - CA |
| CA_0004331 | 49.75 | -125.34 | Dfc: 100% | ENF: 100% | 1980-2005 | 1007-1337 | 8.37 | Piggott Creek - CA |
| CA_0004339 | 50.14 | -126.17 | Dfc: 74.1%; ET: 14.8%; Dfb: 11.1% | ENF: 68%; WSA: 24%; GRA: 8% | 1980-2012 | 577-1630 | 14.60 | Zeballos River - CA |
| CA_0004345 | 50.44 | -126.58 | Dfc: 58%; Cfb: 28.1%; Cfc: 10.7% | ENF: 98.8%; GRA: 0.6%; WSA: 0.4% | 1980-2012 | 109-1539 | 365.00 | Tsitika River - CA |
| CA_0004349 | 50.42 | -126.58 | Dfc: 76.1%; Cfc: 9.1%; Cfb: 8% | ENF: 97.5%; WSA: 2.5% | 1980-1999 | 206-1448 | 46.10 | Catherine Creek - CA |
| CA_0004351 | 50.73 | -127.88 | Cfc: 74.5%; Cfb: 25.5% | ENF: 97.6%; EBF: 2.4% | 1980-2012 | 236-503 | 22.30 | Pugh Creek - CA |
| CA_0004364 | 53.26 | -125.41 | Dfc: 81.7%; Dsc: 12.3%; ET: 6% | WSA: 78.8%; ENF: 19.3%; GRA: 1.9% | 1980-2006 | 1022-1826 | 150.00 | Van Tine Creek - CA |
| CA_0004365 | 53.65 | -127.54 | ET: 62.2%; Dfc: 37.8% | GRA: 50.4%; ENF: 26.6%; WSA: 10.1% | 1980-2013 | 880-2026 | 86.50 | Laventie Creek - CA |
| CA_0004366 | 53.80 | -126.36 | ET: 56.7%; Dsc: 43.3% | GRA: 35.6%; WSA: 34.5%; ENF: 28.7% | 1980-1995 | 896-2078 | 53.40 | Macivor Creek - CA |
| CA_0004375 | 53.66 | -126.99 | Dfc: 73.3%; ET: 26.7% | ENF: 57.1%; GRA: 28.6%; WSA: 14.3% | 1980-2013 | 1001-1637 | 7.72 | Whitesail Middle Creek - CA |
| CA_0004380 | 53.90 | -126.95 | Dfc: 84%; ET: 12.6%; Dsc: 3.4% | ENF: 49.5%; WSA: 38.1%; GRA: 7.9% | 1980-2015 | 927-2147 | 369.00 | Nadina River - CA |
| CA_0004381 | 54.18 | -125.49 | Dsc: 96.8%; Dfc: 3.2% | WSA: 61.1%; ENF: 33.9%; GRA: 2.4% | 1980-2004 | 697-1415 | 771.00 | Endako River - CA |
| CA_0004397 | 55.98 | -126.68 | Dfc: 65.8%; ET: 19.8%; Dsc: 14.4% | ENF: 71.6%; WSA: 14.6%; GRA: 12% | 1980-2012 | 779-2083 | 403.00 | Driftwood River - CA |
| CA_0004402 | 53.31 | -120.25 | ET: 64.5%; Dfc: 35.5% | WSA: 31.3%; GRA: 27.3%; BSV: 25.2% | 1980-2013 | 782-2722 | 409.00 | Dore River - CA |
| CA_0004406 | 52.99 | -119.01 | ET: 61.4%; Dfc: 38.6% | GRA: 40%; WSA: 20.3%; ENF: 18.5% | 1980-2013 | 1030-3108 | 1710.00 | Fraser River - CA |
| CA_0004407 | 52.92 | -118.80 | ET: 66.9%; Dfc: 33.1% | GRA: 46.4%; BSV: 23.4%; WSA: 16% | 1980-1995 | 1082-2985 | 458.00 | Moose River - CA |
| CA_0004408 | 53.44 | -120.22 | ET: 61.6%; Dfc: 38.4% | GRA: 43%; WSA: 26.8%; ENF: 25.4% | 1980-2013 | 983-2296 | 253.00 | Mckale River - CA |
| CA_0004411 | 52.84 | -119.27 | ET: 62.8%; Dfc: 36.8%; Dfb: 0.4% | GRA: 56.6%; WSA: 20.5%; ENF: 17.8% | 1980-1998 | 806-2339 | 132.00 | Swift Creek - CA |
| CA_0004412 | 53.68 | -120.59 | Dfc: 55.1%; ET: 44.9% | WSA: 38.3%; GRA: 32.8%; ENF: 25.4% | 1980-2013 | 704-2360 | 1260.00 | Morkill River - CA |
| CA_0004416 | 54.30 | -120.98 | Dfc: 87%; ET: 13% | WSA: 45%; GRA: 27.2%; ENF: 22.2% | 1980-2013 | 954-2194 | 103.00 | Muller Creek - CA |
| CA_0004419 | 54.61 | -123.24 | Dfc: 98.5%; Dsc: 1.5% | WSA: 62.2%; ENF: 34%; GRA: 3.9% | 1980-1998 | 811-1059 | 303.00 | Muskeg River - CA |
| CA_0004420 | 53.26 | -121.41 | Dfc: 87.4%; ET: 12.6% | ENF: 64.3%; WSA: 26.1%; GRA: 9.5% | 1980-1995 | 909-2150 | 458.00 | Bowron River - CA |
| CA_0004433 | 53.16 | -122.48 | Dfc: 95.7%; Dfb: 3.4%; ET: 0.9% | WSA: 49.8%; ENF: 41.3%; MF: 8.4% | 1980-1999 | 589-1862 | 1910.00 | Cottonwood River - CA |
| CA_0004436 | 52.97 | -122.51 | Dfc: 95.2%; Dfb: 4.8% | WSA: 81.9%; ENF: 12.6%; GRA: 3.4% | 1980-2013 | 483-1471 | 1550.00 | Baker Creek - CA |



Continuation of Table A5

| Station | Lat | Lon | Climate | Biome | Period | Elev | Area | River |
|---|---|---|---|---|---|---|---|---|
| CA_0004443 | 52.91 | -121.77 | Dfc: 93.5%; ET: 6.5% | WSA: 64.8%; ENF: 35.2% | 1980-2013 | 1088-1955 | 127.00 | Little Swift River - CA |
| CA_0004451 | 53.86 | -122.57 | Dfc: 97.5%; Dfb: 2.5% | MF: 51%; ENF: 34.3%; WSA: 14.7% | 1980-1999 | 681-1203 | 60.20 | Tabor Creek - CA |
| CA_0004452 | 52.64 | -122.41 | Dfc: 73.7%; Dfb: 26.3% | ENF: 59.5%; WSA: 40.5% | 1980-1995 | 823-1394 | 37.30 | Alix Creek - CA |
| CA_0004453 | 52.83 | -122.54 | Dfc: 91.8%; Dfb: 8.2% | WSA: 68.8%; ENF: 24.2%; MF: 6.4% | 1980-1993 | 796-1401 | 99.10 | Deserters Creek - CA |
| CA_0004459 | 52.99 | -123.80 | Dfc: 98.8%; ET: 1.2% | WSA: 86.1%; ENF: 10.8%; GRA: 3.1% | 1980-1995 | 964-1706 | 992.00 | Baezaeko River - CA |
| CA_0004469 | 52.29 | -121.07 | Dfc: 80.5%; ET: 19.5% | ENF: 52.7%; WSA: 25.9%; GRA: 16.7% | 1980-2013 | 889-2400 | 790.00 | Horsefly River - CA |
| CA_0004474 | 52.32 | -121.41 | Dfc: 97.7%; Dfb: 1.3%; ET: 1% | WSA: 74.7%; ENF: 17.4%; GRA: 8% | 1980-2012 | 803-2064 | 548.00 | Moffat Creek - CA |
| CA_0004475 | 52.28 | -121.00 | Dfc: 95.6%; Dfb: 2.4%; ET: 2.1% | ENF: 75.1%; WSA: 21.2%; MF: 1.9% | 1980-2014 | 863-2098 | 431.00 | Mckinley Creek - CA |
| CA_0004478 | 52.95 | -122.40 | Dfb: 100% | WSA: 42.9%; ENF: 28.6%; GRA: 14.3% | 1980-1994 | 617-834 | 7.25 | Dragon Creek - CA |
| CA_0004480 | 52.37 | -121.36 | Dfc: 60.1%; Dfb: 38.3%; ET: 1.6% | ENF: 74.5%; WSA: 17.4%; MF: 6.1% | 1980-1990 | 783-2121 | 416.00 | Little Horsefly River - CA |
| CA_0004501 | 51.61 | -121.23 | Dfc: 99.1%; Dfb: 0.9% | WSA: 69.5%; ENF: 29.5%; GRA: 0.7% | 1980-1997 | 991-1614 | 912.00 | Bridge Creek - CA |
| CA_0004502 | 51.73 | -120.01 | Dfc: 69.4%; ET: 28.1%; Dfb: 2.5% | ENF: 64%; WSA: 19%; GRA: 15% | 1980-1994 | 847-2431 | 52.50 | Spahats Creek - CA |
| CA_0004509 | 51.62 | -120.67 | Dfc: 100% | ENF: 60%; WSA: 40% | 1980-2011 | 1175-1861 | 79.40 | Windy Creek - CA |
| CA_0004521 | 50.72 | -120.03 | Dfb: 54%; Dfc: 44.4%; BSk: 1.6% | WSA: 50%; ENF: 44.4%; GRA: 2.8% | 1980-2009 | 870-1438 | 55.70 | Paul Creek - CA |
| CA_0004527 | 51.18 | -120.13 | Dfc: 69%; Dfb: 28.1%; ET: 2.3% | ENF: 64.7%; WSA: 29.8%; GRA: 4.8% | 1980-2014 | 381-2217 | 1140.00 | Barriere River - CA |
| CA_0004531 | 51.12 | -120.21 | Dfc: 80.5%; Dfb: 19.5% | WSA: 74.4%; ENF: 20.5%; GRA: 5.1% | 1980-2013 | 617-1623 | 135.00 | Fishtrap Creek - CA |
| CA_0004541 | 52.12 | -119.30 | Dfc: 81.1%; ET: 13.5%; Dfb: 5.4% | WSA: 38.1%; ENF: 30.4%; GRA: 25.4% | 1980-2013 | 685-2331 | 272.00 | Blue River - CA |
| CA_0004567 | 51.25 | -119.94 | Dfc: 79.6%; Dfb: 16.1%; ET: 4.4% | ENF: 58.1%; WSA: 33.7%; GRA: 7.8% | 1980-2013 | 571-2217 | 624.00 | Barriere River - CA |
| CA_0004570 | 51.13 | -120.12 | Dfc: 70.7%; Dfb: 29.3% | ENF: 54.6%; WSA: 40.5%; MF: 2.7% | 1980-1996 | 388-2075 | 515.00 | Louis Creek - CA |
| CA_0004572 | 51.35 | -119.88 | Dfc: 88.5%; ET: 8.6%; Dfb: 3% | ENF: 59.2%; WSA: 29.2%; GRA: 10.9% | 1980-2013 | 803-2217 | 166.00 | Harper Creek - CA |
| CA_0004574 | 51.43 | -120.20 | Dfc: 85.7%; Dfb: 14.3% | ENF: 50.6%; WSA: 45.8%; MF: 3.4% | 1980-2013 | 399-1679 | 443.00 | Lemieux Creek - CA |
| CA_0004579 | 50.30 | -118.82 | Dfc: 70.2%; Dfb: 26%; ET: 3.8% | ENF: 67.4%; WSA: 19.9%; GRA: 10.5% | 1980-2013 | 455-2687 | 2000.00 | Shuswap River - CA |
| CA_0004590 | 50.35 | -118.55 | Dfc: 75.1%; Dfb: 18.7%; ET: 6.2% | ENF: 60.3%; WSA: 19.3%; GRA: 17.1% | 1980-2013 | 600-2687 | 1130.00 | Shuswap River - CA |
| CA_0004608 | 50.30 | -118.86 | Dfc: 56.9%; Dfb: 43.1% | ENF: 58.3%; WSA: 33.9%; GRA: 6.4% | 1980-2013 | 483-1934 | 769.00 | Bessette Creek - CA |
| CA_0004609 | 50.28 | -118.95 | Dfb: 62.3%; Dfc: 37.7% | ENF: 76.7%; WSA: 17.2%; MF: 5.2% | 1980-2013 | 505-1844 | 70.90 | Vance Creek - CA |



Continuation of Table A5

| Station | Lat | Lon | Climate | Biome | Period | Elev | Area | River |
|---------|-----|-----|---------|-------|--------|------|------|-------|
| CA_0004610 | 50.25 | -118.96 | Dfc: 65.2%; Dfb: 34.8% | ENF: 57.2%; WSA: 35.6%; GRA: 6.2% | 1980-2011 | 495-1934 | 632.00 | Bessette Creek - CA |
| CA_0004613 | 50.26 | -118.63 | Dfc: 74.2%; Dfb: 24.8%; ET: 1.1% | ENF: 80.1%; WSA: 15%; GRA: 3.8% | 1980-1990 | 525-2352 | 503.00 | Cherry Creek - CA |
| CA_0004614 | 50.53 | -118.97 | Dfb: 63.6%; Dfc: 36.4% | ENF: 86.9%; WSA: 9.5%; SAV: 2.5% | 1980-1990 | 431-1760 | 191.00 | Trinity Creek - CA |
| CA_0004616 | 50.89 | -119.73 | Dfc: 78%; Dfb: 22% | ENF: 92.6%; WSA: 7.4% | 1980-2002 | 634-1765 | 69.70 | Hiuihill Creek - CA |
| CA_0004620 | 51.43 | -119.47 | Dfc: 81.1%; Dfb: 18.9% | ENF: 85.7%; WSA: 14.3% | 1980-1998 | 422-1580 | 14.70 | Fisher Creek - CA |
| CA_0004640 | 50.50 | -119.56 | Dfc: 56%; BSk: 25.5%; Dfb: 18.5% | WSA: 53.7%; ENF: 34.4%; GRA: 10.3% | 1980-2013 | 588-1914 | 1030.00 | Salmon River - CA |
| CA_0004641 | 50.69 | -119.33 | Dfc: 47.7%; Dfb: 34.7%; BSk: 17.6% | WSA: 46.7%; ENF: 40.6%; GRA: 10.6% | 1980-2011 | 353-1914 | 1550.00 | Salmon River - CA |
| CA_0004644 | 50.94 | -118.80 | Dfc: 68.1%; Dfb: 25.9%; ET: 6% | ENF: 55.6%; WSA: 20.4%; GRA: 18.5% | 1980-2014 | 367-2708 | 932.00 | Eagle River - CA |
| CA_0004647 | 51.26 | -118.95 | Dfc: 66.2%; Dfb: 24.4%; ET: 9.4% | ENF: 57.4%; WSA: 19.4%; GRA: 15.8% | 1980-2013 | 365-2568 | 805.00 | Seymour River - CA |
| CA_0004687 | 50.29 | -119.96 | Dfc: 85%; Dfb: 8.6%; BSk: 6.4% | WSA: 70.8%; ENF: 22.9%; GRA: 6.2% | 1980-2002 | 1000-1914 | 143.00 | Salmon River - CA |
| CA_0004688 | 50.91 | -119.53 | Dfc: 60%; Dfb: 40% | ENF: 94.1%; WSA: 5.9% | 1980-2010 | 400-1719 | 26.20 | Corning Creek - CA |
| CA_0004707 | 50.52 | -119.85 | Dfc: 73.4%; BSk: 14.5%; Dfb: 12.1% | WSA: 61.3%; ENF: 28.3%; SAV: 7.5% | 1980-1994 | 708-1705 | 64.30 | Monte Creek - CA |
| CA_0004710 | 50.93 | -118.47 | Dfc: 58%; Dfb: 38%; ET: 4% | ENF: 90%; DBF: 5%; GRA: 2.5% | 1980-1994 | 556-2189 | 30.20 | South Pass Creek - CA |
| CA_0004712 | 50.70 | -119.20 | Dfb: 73.7%; Dfc: 26.3% | ENF: 100% | 1980-2014 | 626-1299 | 20.80 | East Canoe Creek - CA |
| CA_0004714 | 50.82 | -119.68 | Dfc: 64.9%; Dfb: 35.1% | ENF: 59%; WSA: 36.1%; MF: 3.3% | 1980-2013 | 384-1777 | 297.00 | Chase Creek - CA |
| CA_0004721 | 50.88 | -120.97 | Dfc: 75.5%; Dfb: 17%; BSk: 7.6% | WSA: 66.4%; ENF: 28.9%; GRA: 4.7% | 1980-2014 | 546-1855 | 479.00 | Criss Creek - CA |
| CA_0004728 | 50.88 | -121.42 | Dfc: 70.1%; Dfb: 17.3%; BSk: 10% | WSA: 52.4%; ENF: 35.8%; GRA: 10.9% | 1980-1994 | 591-2254 | 658.00 | Hat Creek - CA |
| CA_0004739 | 50.90 | -120.98 | Dfc: 80.4%; Dfb: 14.5%; BSk: 5.1% | WSA: 77.5%; ENF: 14.5%; GRA: 7.9% | 1980-2014 | 554-1741 | 878.00 | Deadman River - CA |
| CA_0004771 | 51.04 | -121.44 | Dfc: 67.8%; Dfb: 22.4%; BSk: 9.7% | WSA: 67.4%; ENF: 29.6%; GRA: 2.7% | 1980-1996 | 728-1697 | 479.00 | Loon Creek - CA |
| CA_0004773 | 51.26 | -121.68 | Dfc: 89.6%; ET: 10.4% | WSA: 69.2%; GRA: 16.9%; ENF: 12.3% | 1980-1996 | 1143-2168 | 36.40 | Fiftynine Creek - CA |
| CA_0004774 | 50.74 | -121.58 | Dfc: 90.6%; Dfb: 9.4% | WSA: 66.7%; GRA: 18.8%; ENF: 14.6% | 1980-1998 | 1131-1914 | 32.90 | Ambusten Creek - CA |
| CA_0004777 | 50.73 | -121.64 | Dfc: 91.9%; ET: 8.1% | ENF: 73.1%; WSA: 19.2%; GRA: 3.8% | 1980-1998 | 1193-2110 | 31.90 | Anderson Creek - CA |
| CA_0004778 | 50.68 | -120.57 | Dfc: 53%; BSk: 47% | WSA: 62.5%; ENF: 20%; GRA: 17.5% | 1980-1996 | 628-1860 | 143.00 | Cherry Creek - CA |
| CA_0004780 | 50.98 | -121.40 | Dfc: 91.7%; Dfb: 4.3%; BSk: 4% | WSA: 68.2%; GRA: 18.4%; ENF: 13.5% | 1980-1994 | 885-1749 | 141.00 | Scottie Creek - CA |
| CA_0004782 | 51.01 | -120.75 | Dfc: 100% | WSA: 93.8%; GRA: 6.2% | 1980-1994 | 1120-1855 | 46.60 | Heller Creek - CA |





Continuation of Table A5

| Station | Lat | Lon | Climate | Biome | Period | Elev | Area | River |
|---|---|---|---|---|---|---|---|---|
| CA_0004785 | 51.15 | -120.86 | Dfc: 100% | WSA: 85.5%; GRA: 13.9%; ENF: 0.6% | 1980-2015 | 1078-1525 | 98.90 | Joe Ross Creek - CA |
| CA_0004786 | 51.43 | -120.62 | Dfc: 100% | WSA: 83.9%; ENF: 16.1% | 1980-1994 | 1149-1489 | 20.40 | Mcdonald Creek - CA |
| CA_0004789 | 50.84 | -121.24 | Dfc: 80.2%; Dfb: 10.5%; BSk: 9.3% | WSA: 67.6%; ENF: 27%; GRA: 5.4% | 1980-2013 | 770-1661 | 50.50 | Arrowstone Creek - CA |
| CA_0004790 | 50.84 | -121.15 | Dfc: 78.9%; Dfb: 10.5%; BSk: 10.5% | WSA: 70.6%; ENF: 23.5%; GRA: 5.9% | 1980-2011 | 943-1628 | 10.60 | Dairy Creek - CA |
| CA_0004798 | 50.14 | -121.03 | Dsc: 73.1%; Dsb: 24.4%; BSk: 1.8% | ENF: 64.8%; WSA: 32.4%; GRA: 2.7% | 1980-2014 | 542-2150 | 775.00 | Spius Creek - CA |
| CA_0004799 | 50.49 | -120.86 | Dfc: 100% | WSA: 66.5%; GRA: 30.2%; BSV: 1.7% | 1980-1996 | 1120-1829 | 139.00 | Witches Brook - CA |
| CA_0004800 | 50.11 | -120.80 | Dsc: 51.4%; Dsb: 35.1%; BSk: 6.4% | ENF: 48.9%; WSA: 47.6%; GRA: 3% | 1980-2014 | 598-1989 | 917.00 | Coldwater River - CA |
| CA_0004805 | 49.96 | -120.13 | Dfc: 100% | WSA: 72.3%; ENF: 22.3%; GRA: 5.4% | 1980-2013 | 1458-1929 | 87.60 | Pennask Creek - CA |
| CA_0004824 | 50.36 | -120.81 | Dfc: 95.1%; BSk: 4.4%; Dfb: 0.5% | WSA: 73.5%; GRA: 17.8%; ENF: 8% | 1980-2014 | 955-1907 | 871.00 | Guichon Creek - CA |
| CA_0004829 | 49.85 | -120.91 | Dsc: 83.7%; Dsb: 13.3%; Dfc: 2.7% | ENF: 66.5%; WSA: 30.3%; GRA: 2.9% | 1980-2014 | 919-1989 | 316.00 | Coldwater River - CA |
| CA_0004830 | 50.18 | -120.38 | Dfc: 58.4%; BSk: 30%; Dfb: 11.6% | WSA: 50.4%; GRA: 36.5%; ENF: 12.3% | 1980-2014 | 736-1929 | 1500.00 | Nicola River - CA |
| CA_0004836 | 50.61 | -120.91 | Dfc: 100% | WSA: 81.9%; GRA: 17.4%; ENF: 0.6% | 1980-2014 | 1189-1931 | 78.20 | Guichon Creek - CA |
| CA_0004840 | 50.14 | -120.28 | Dfc: 60%; BSk: 33.5%; Dfb: 6.6% | WSA: 52.3%; GRA: 34.4%; ENF: 11.4% | 1980-1996 | 826-1929 | 241.00 | Spahomin Creek - CA |
| CA_0004843 | 50.11 | -119.98 | Dfc: 97.1%; BSk: 2.9% | WSA: 81.8%; GRA: 9.5%; ENF: 8.8% | 1980-2001 | 1110-1820 | 85.00 | Beak Creek - CA |
| CA_0004845 | 50.33 | -120.92 | Dfc: 100% | GRA: 77.8%; WSA: 22.2% | 1980-1998 | 1313-1719 | 32.20 | Chataway Creek - CA |
| CA_0004846 | 50.15 | -120.88 | Dfc: 86.5%; BSk: 9.3%; Dfb: 4.2% | WSA: 74.3%; GRA: 16.5%; ENF: 8.6% | 1980-2015 | 580-1907 | 1230.00 | Guichon Creek - CA |
| CA_0004847 | 49.95 | -121.10 | Dsc: 87.9%; Dsb: 12.1% | ENF: 69.2%; WSA: 27.7%; GRA: 3.1% | 1980-2011 | 932-2019 | 178.00 | Spius Creek - CA |
| CA_0004850 | 51.38 | -123.63 | ET: 66%; Dfc: 34% | BSV: 32.6%; ENF: 24.5%; GRA: 21.4% | 1980-2013 | 1323-2929 | 1520.00 | Taseko River - CA |
| CA_0004851 | 51.67 | -124.15 | ET: 45.1%; Dsc: 44.5%; Dfc: 10.4% | ENF: 62.7%; WSA: 22.9%; GRA: 12.4% | 1980-2014 | 1256-2140 | 98.80 | Lingfield Creek - CA |
| CA_0004859 | 51.25 | -123.10 | ET: 91.2%; Dfc: 8.8% | GRA: 48.9%; WSA: 26.5%; BSV: 23% | 1980-2013 | 1702-2737 | 232.00 | Big Creek - CA |
| CA_0004894 | 52.11 | -121.94 | Dfc: 71.7%; Dfb: 28.3% | WSA: 66.5%; ENF: 25.5%; GRA: 8% | 1980-2010 | 736-1081 | 192.00 | Borland Creek - CA |
| CA_0004895 | 52.08 | -121.99 | Dfb: 56.7%; Dfc: 43.3% | ENF: 60.5%; WSA: 36.1%; GRA: 3.2% | 1980-2013 | 615-1609 | 1990.00 | San Jose River - CA |
| CA_0004898 | 52.15 | -122.56 | Dfc: 100% | WSA: 52.3%; ENF: 46.5%; SAV: 1.3% | 1980-1994 | 990-1280 | 82.70 | Meldrum Creek - CA |
| CA_0004900 | 52.43 | -122.29 | Dfc: 85.6%; Dfb: 14.4% | WSA: 55%; ENF: 45% | 1980-2014 | 725-1312 | 99.00 | Sheridan Creek - CA |
| CA_0004925 | 50.91 | -121.72 | Dfc: 98.5%; ET: 1.5% | ENF: 83.9%; WSA: 16.1% | 1980-1998 | 1326-2020 | 37.10 | Pavilion Creek - CA |
| CA_0004929 | 50.84 | -121.90 | Dfc: 89.4%; Dfb: 10.6% | ENF: 50%; WSA: 50% | 1980-1994 | 904-1919 | 36.30 | Lee Creek - CA |



Continuation of Table A5

| Station | Lat | Lon | Climate | Biome | Period | Elev | Area | River |
|---|---|---|---|---|---|---|---|---|
| CA_0004933 | 50.67 | -121.97 | ET: 42.1%; Dsc: 34.6%; Dfc: 17.5% | GRA: 32%; WSA: 30.6%; ENF: 24.7% | 1980-2014 | 275-2712 | 885.00 | Cayoosh Creek - CA |
| CA_0004934 | 50.67 | -121.97 | Dsc: 32.7%; Dsb: 24.9%; ET: 23% | ENF: 37.1%; WSA: 29.2%; GRA: 24.1% | 1980-2013 | 244-2668 | 1020.00 | Seton River - CA |
| CA_0004947 | 50.86 | -123.45 | ET: 63.1%; Dfc: 36.9% | SNO: 64%; GRA: 18.4%; BSV: 14.6% | 1980-2013 | 1381-2815 | 144.00 | Bridge River - CA |
| CA_0004949 | 50.91 | -122.24 | Dfc: 43.2%; ET: 40%; Dsc: 14.5% | WSA: 49.2%; GRA: 20.9%; ENF: 14.9% | 1980-2013 | 795-2711 | 581.00 | Yalakom River - CA |
| CA_0004950 | 50.96 | -122.30 | Dfc: 93.2%; ET: 4.3%; Dsc: 2.5% | WSA: 53.9%; ENF: 25.5%; GRA: 20.6% | 1980-1994 | 1013-2101 | 86.40 | Junction Creek - CA |
| CA_0004951 | 50.73 | -122.94 | ET: 46.6%; Dsc: 40.4%; Dfc: 13% | GRA: 38.4%; BSV: 25.2%; ENF: 17.3% | 1980-2013 | 1029-2621 | 312.00 | Hurley River - CA |
| CA_0004952 | 50.82 | -123.20 | ET: 61.1%; Dfc: 38%; Dfb: 0.8% | GRA: 28.6%; SNO: 26.6%; BSV: 24.9% | 1980-2011 | 763-2815 | 708.00 | Bridge River - CA |
| CA_0004991 | 49.54 | -121.12 | Dsc: 78%; Dfc: 14.3%; Dsb: 7.7% | ENF: 52.3%; WSA: 29.5%; GRA: 16.8% | 1980-2014 | 881-1886 | 85.50 | Coquihalla River - CA |
| CA_0004993 | 49.95 | -121.86 | Dsc: 43.9%; Dfc: 28.4%; Dsb: 17.1% | ENF: 36%; GRA: 32.3%; WSA: 20% | 1980-2014 | 322-2559 | 712.00 | Nahatlatch River - CA |
| CA_0005014 | 50.12 | -122.95 | Dfc: 49.4%; ET: 28.8%; Dsc: 11.2% | ENF: 40.4%; GRA: 24.5%; BSV: 14.6% | 1980-2012 | 657-2536 | 89.70 | Fitzsimmons Creek - CA |
| CA_0005027 | 49.08 | -121.46 | Dfc: 76.3%; Dfb: 19.7%; Dsb: 2.5% | ENF: 64.4%; GRA: 15.9%; WSA: 6.9% | 1980-2014 | 619-2384 | 335.00 | Chilliwack River - CA |
| CA_0005044 | 49.07 | -121.70 | Dfc: 75.8%; Dfb: 22.1%; Cfb: 2.1% | ENF: 66%; GRA: 20.1%; SAV: 5.7% | 1980-2014 | 312-2079 | 160.00 | Slesse Creek - CA |
| CA_0005045 | 49.19 | -122.16 | Dfb: 82%; Cfb: 10.2%; Dfc: 7.8% | ENF: 98.3%; MF: 1.7% | 1980-2007 | 78-1261 | 117.00 | Norrish Creek - CA |
| CA_0005070 | 49.10 | -121.66 | Dfc: 74.6%; Dfb: 23%; Dsb: 1.3% | ENF: 67.1%; GRA: 15.8%; WSA: 5.8% | 1980-2013 | 357-2384 | 650.00 | Chilliwack River - CA |
| CA_0005083 | 49.49 | -122.79 | Dfc: 67%; Dfb: 33% | ENF: 73.8%; GRA: 16.2%; WSA: 7.5% | 1980-2012 | 335-1723 | 52.50 | Coquitlam River - CA |
| CA_0005086 | 49.56 | -122.32 | Dfc: 73.7%; Dfb: 23.6%; ET: 2.7% | ENF: 37%; GRA: 29.7%; WSA: 13.8% | 1980-2011 | 194-2097 | 290.00 | Stave River - CA |
| CA_0005087 | 49.24 | -122.13 | Dfb: 89.9%; Dfc: 8.6%; Cfb: 1.4% | ENF: 100% | 1980-2006 | 350-1261 | 78.20 | Norrish Creek - CA |
| CA_0005101 | 51.30 | -116.97 | Dfc: 56.5%; ET: 43.1%; Dfb: 0.4% | ENF: 30.3%; GRA: 27.7%; WSA: 20.2% | 1980-2014 | 787-3161 | 1850.00 | Kicking Horse River - CA |
| CA_0005106 | 50.90 | -116.41 | Dfc: 54.2%; ET: 43.6%; Dfb: 2.2% | WSA: 29%; GRA: 28.5%; ENF: 24.9% | 1980-2014 | 804-2968 | 1460.00 | Spillimacheen River - CA |
| CA_0005128 | 51.14 | -116.74 | Dfc: 62.5%; ET: 25%; Dfb: 12.5% | ENF: 35.7%; WSA: 35.7%; SAV: 21.4% | 1980-1998 | 914-2272 | 8.03 | Carbonate Creek - CA |
| CA_0005134 | 50.32 | -115.86 | Dfc: 56.1%; ET: 27.2%; Dfb: 16.7% | WSA: 45.4%; ENF: 25.8%; GRA: 22.2% | 1980-1996 | 807-2837 | 891.00 | Columbia River - CA |
| CA_0005142 | 51.44 | -116.37 | ET: 72.5%; Dfc: 27.5% | GRA: 37%; BSV: 37%; WSA: 13.2% | 1980-1996 | 1610-3161 | 119.00 | Kicking Horse River - CA |
| CA_0005163 | 51.48 | -116.97 | ET: 51.4%; Dfc: 48.2%; Dfb: 0.4% | GRA: 32.1%; WSA: 19.9%; BSV: 19.5% | 1980-2015 | 914-3030 | 587.00 | Blaeberry River - CA |
| CA_0005164 | 51.61 | -117.74 | ET: 63.7%; Dfc: 36.3% | GRA: 30.8%; BSV: 26.5%; SNO: 17.1% | 1980-1995 | 1112-3225 | 135.00 | Gold River - CA |
| CA_0005165 | 51.68 | -117.72 | ET: 58.5%; Dfc: 41.3%; Dfb: 0.1% | GRA: 34.3%; BSV: 20%; ENF: 15.9% | 1980-2013 | 837-3225 | 429.00 | Gold River - CA |



Continuation of Table A5

| Station | Lat | Lon | Climate | Biome | Period | Elev | Area | River |
|---|---|---|---|---|---|---|---|---|
| CA_0005166 | 51.65 | -116.74 | ET: 64.1%; Dfc: 35.9% | GRA: 29.9%; BSV: 27%; WSA: 25.1% | 1980-1996 | 1154-2944 | 230.00 | Blaeberry River - CA |
| CA_0005167 | 51.53 | -116.90 | ET: 51.6%; Dfc: 48.4% | GRA: 43.2%; WSA: 22%; ENF: 20.5% | 1980-2014 | 1035-2641 | 80.50 | Split Creek - CA |
| CA_0005169 | 51.51 | -117.47 | Dfc: 57.2%; ET: 41.2%; Dfb: 1.6% | ENF: 30.4%; GRA: 29.8%; BSV: 16.2% | 1980-2014 | 796-3105 | 1150.00 | Beaver River - CA |
| CA_0005173 | 52.73 | -119.38 | ET: 57.8%; Dfc: 42.2% | GRA: 29.4%; SNO: 22.9%; WSA: 18.8% | 1980-2013 | 975-2952 | 305.00 | Canoe River - CA |
| CA_0005184 | 51.67 | -118.60 | Dfc: 66.2%; ET: 24.8%; Dfb: 9% | ENF: 34.7%; WSA: 26%; GRA: 24.9% | 1980-2014 | 622-2745 | 934.00 | Goldstream River - CA |
| CA_0005185 | 51.01 | -118.09 | Dfc: 65.1%; ET: 27.7%; Dfb: 7.3% | ENF: 34.2%; GRA: 27.6%; WSA: 19.5% | 1980-2014 | 511-2921 | 1150.00 | Illecillewaet River - CA |
| CA_0005189 | 51.66 | -118.10 | ET: 54.4%; Dfc: 45.6% | GRA: 43%; BSV: 24.2%; SNO: 13.4% | 1980-1998 | 1028-2745 | 139.00 | Stitt Creek - CA |
| CA_0005190 | 51.64 | -118.67 | Dfc: 77.7%; ET: 16.6%; Dfb: 5.7% | ENF: 42%; GRA: 26%; WSA: 16.6% | 1980-2005 | 744-2253 | 112.00 | Kirbyville Creek - CA |
| CA_0005195 | 50.77 | -117.68 | Dfc: 57.4%; ET: 32.1%; Dfb: 10.6% | GRA: 35.2%; ENF: 18.7%; WSA: 17.1% | 1980-1996 | 474-3045 | 1020.00 | Incomappleux River - CA |
| CA_0005199 | 50.28 | -117.73 | Dfc: 91.9%; Dfb: 6.7%; ET: 1.4% | GRA: 34.1%; ENF: 32.9%; WSA: 30.1% | 1980-2013 | 737-2416 | 330.00 | Kuskanax Creek - CA |
| CA_0005201 | 50.73 | -117.73 | Dfc: 59.7%; Dfb: 40.3% | ENF: 87.3%; WSA: 11.1%; SAV: 1.6% | 1980-2013 | 545-2138 | 96.70 | Beaton Creek - CA |
| CA_0005226 | 49.01 | -117.95 | Dfc: 53.9%; Dfb: 35.1%; Dsb: 8.3% | ENF: 60.8%; WSA: 38.3%; MF: 0.8% | 1980-2015 | 687-2224 | 347.00 | Big Sheep Creek - CA |
| CA_0005255 | 49.05 | -117.29 | Dfc: 59.5%; Dfb: 29.3%; Dsc: 6.5% | ENF: 59.9%; WSA: 32.7%; GRA: 4.2% | 1980-2014 | 609-2234 | 1240.00 | Salmo River - CA |
| CA_0005258 | 49.91 | -118.13 | Dfc: 81.8%; Dfb: 18.2% | ENF: 79.3%; WSA: 16.3%; GRA: 1.8% | 1980-2013 | 623-2052 | 204.00 | Barnes Creek - CA |
| CA_0005265 | 49.45 | -118.04 | Dfc: 58.7%; Dfb: 41.3% | ENF: 72.1%; WSA: 27%; GRA: 0.9% | 1980-2015 | 720-2130 | 81.60 | Deer Creek - CA |
| CA_0005281 | 49.90 | -118.19 | Dfc: 82.3%; Dfb: 17.7% | ENF: 53.2%; WSA: 42.5%; GRA: 3.8% | 1980-2013 | 555-2167 | 298.00 | Inonoaklin Creek - CA |
| CA_0005285 | 49.23 | -117.24 | Dfc: 82.7%; Dfb: 17.3% | ENF: 62.9%; WSA: 25.8%; GRA: 5.6% | 1980-2013 | 898-2048 | 56.70 | Hidden Creek - CA |
| CA_0005288 | 50.34 | -117.52 | Dfc: 97%; ET: 3% | GRA: 56.6%; WSA: 24.9%; ENF: 17.9% | 1980-1996 | 1088-2425 | 113.00 | Kuskanax Creek - CA |
| CA_0005293 | 50.79 | -118.08 | Dfc: 77.9%; ET: 17.9%; Dfb: 4.2% | GRA: 26.4%; ENF: 23.3%; WSA: 22.1% | 1980-2011 | 818-2674 | 99.60 | Cranberry Creek - CA |
| CA_0005299 | 50.89 | -116.05 | Dfc: 70.4%; ET: 29.6% | ENF: 36%; WSA: 31.4%; GRA: 25.1% | 1980-2014 | 1179-2862 | 416.00 | Kootenay River - CA |
| CA_0005303 | 50.66 | -115.53 | Dfc: 55.1%; ET: 44.9% | GRA: 38.5%; WSA: 29.4%; ENF: 14.7% | 1980-1999 | 1330-2909 | 69.70 | Albert River - CA |
| CA_0005304 | 50.53 | -115.62 | Dfc: 61.7%; ET: 38.3% | GRA: 33.6%; WSA: 28.2%; ENF: 25.4% | 1980-1995 | 1104-3204 | 653.00 | Palliser River - CA |
| CA_0005306 | 49.49 | -115.37 | Dfc: 84.3%; ET: 12.2%; Dfb: 3.5% | WSA: 37.2%; ENF: 33.4%; GRA: 19.1% | 1980-2014 | 894-3043 | 1520.00 | Bull River - CA |
| CA_0005345 | 49.61 | -116.17 | Dfc: 80.2%; ET: 16.1%; Dfb: 3.7% | WSA: 37.2%; ENF: 31.3%; GRA: 26.4% | 1980-1995 | 972-2759 | 1480.00 | St Mary River - CA |
| CA_0005369 | 49.71 | -115.90 | Dfc: 83.5%; ET: 8.7%; Dfb: 7.9% | WSA: 73.7%; GRA: 12.7%; SAV: 7% | 1980-2014 | 1048-2516 | 135.00 | Mather Creek - CA |



Continuation of Table A5

| Station | Lat | Lon | Climate | Biome | Period | Elev | Area | River |
|---|---|---|---|---|---|---|---|---|
| CA_0005370 | 49.74 | -116.44 | Dfc: 88.8%; ET: 11.2% | WSA: 36.2%; GRA: 31%; ENF: 27.7% | 1980-2014 | 1155-2530 | 208.00 | St Mary River - CA |
| CA_0005371 | 49.19 | -115.44 | Dfc: 81%; Dfb: 19% | ENF: 71.1%; WSA: 25.9%; GRA: 2.3% | 1980-1995 | 1041-2085 | 313.00 | Caven Creek - CA |
| CA_0005376 | 49.70 | -116.03 | Dfc: 90.7%; ET: 9.3% | WSA: 51%; ENF: 30.2%; SAV: 12.8% | 1980-1999 | 1360-2421 | 102.00 | Mark Creek - CA |
| CA_0005377 | 49.66 | -116.07 | Dfc: 82.5%; ET: 17.5% | WSA: 46.7%; ENF: 25.8%; GRA: 14% | 1980-1998 | 1234-2539 | 148.00 | Matthew Creek - CA |
| CA_0005383 | 49.09 | -116.46 | Dfc: 75.4%; Dfb: 24.3%; ET: 0.3% | ENF: 55.7%; WSA: 37.6%; GRA: 4.5% | 1980-1995 | 643-2412 | 1180.00 | Goat River - CA |
| CA_0005384 | 49.91 | -116.95 | Dfc: 79.6%; ET: 11.9%; Dfb: 8.5% | ENF: 41.3%; WSA: 28.1%; GRA: 24.4% | 1980-2014 | 739-2684 | 442.00 | Kaslo River - CA |
| CA_0005385 | 49.00 | -116.18 | Dfc: 73.8%; Dfb: 25.8%; Dsc: 0.4% | ENF: 56.7%; WSA: 38.6%; GRA: 3.7% | 1980-2013 | 822-2284 | 1480.00 | Moyie River - CA |
| CA_0005386 | 50.26 | -116.97 | Dfc: 73.3%; ET: 13.6%; Dfb: 13.1% | ENF: 40.3%; GRA: 31.8%; WSA: 20% | 1980-2014 | 564-2692 | 1640.00 | Lardeau River - CA |
| CA_0005395 | 49.20 | -116.53 | Dfc: 78.5%; Dfb: 21.5% | ENF: 59.5%; WSA: 36.7%; MF: 2.5% | 1980-2014 | 825-2082 | 57.00 | Duck Creek - CA |
| CA_0005400 | 49.00 | -116.57 | Dfc: 78.6%; Dfb: 19.8%; Dsc: 0.9% | ENF: 85.4%; WSA: 10.5%; GRA: 2.2% | 1980-2013 | 539-1967 | 242.00 | Boundary Creek - CA |
| CA_0005444 | 49.16 | -116.45 | Dfc: 80.9%; Dfb: 19.1% | ENF: 83.8%; WSA: 16.2% | 1980-2014 | 846-2040 | 78.30 | Arrow Creek - CA |
| CA_0005450 | 49.70 | -116.92 | Dfc: 71.7%; ET: 9.4%; Dsb: 8.8% | WSA: 34.8%; ENF: 26.5%; GRA: 25.8% | 1980-1992 | 759-2602 | 87.30 | Coffee Creek - CA |
| CA_0005460 | 49.10 | -116.44 | Dfc: 58.3%; Dfb: 41.7% | ENF: 90%; SAV: 10% | 1980-2014 | 811-1794 | 6.22 | Sullivan Creek - CA |
| CA_0005462 | 49.34 | -116.71 | Dfc: 73.9%; Dfb: 26.1% | ENF: 61.1%; WSA: 38.9% | 1980-1999 | 767-1978 | 12.70 | Twin Bays Creek - CA |
| CA_0005464 | 50.64 | -117.05 | Dfc: 58.1%; ET: 38%; Dfb: 3.8% | GRA: 36.6%; WSA: 19.9%; BSV: 19.8% | 1980-2014 | 597-3124 | 1310.00 | Duncan River - CA |
| CA_0005465 | 49.42 | -115.94 | Dfc: 99.6%; Dfb: 0.2%; ET: 0.2% | ENF: 53.6%; WSA: 42.4%; GRA: 3.6% | 1980-2014 | 1189-2284 | 239.00 | Moyie River - CA |
| CA_0005472 | 50.08 | -116.78 | Dfc: 63.2%; ET: 35.9%; Dfb: 0.9% | GRA: 40.6%; WSA: 29.6%; ENF: 15% | 1980-2014 | 887-2978 | 585.00 | Fry Creek - CA |
| CA_0005473 | 50.16 | -116.58 | ET: 50.2%; Dfc: 49.8% | GRA: 37.4%; BSV: 26.2%; WSA: 23% | 1980-2004 | 1371-2906 | 119.00 | Carney Creek - CA |
| CA_0005474 | 49.87 | -117.12 | Dfc: 75.8%; ET: 24.2% | GRA: 44.9%; ENF: 25.4%; WSA: 24.6% | 1980-2014 | 1242-2684 | 92.30 | Keen Creek - CA |
| CA_0005477 | 50.63 | -117.54 | Dfc: 83.3%; ET: 11.1%; Dfb: 5.6% | ENF: 38.5%; WSA: 30.8%; GRA: 23.1% | 1980-2013 | 1006-2503 | 8.84 | Humphries Creek - CA |
| CA_0005496 | 49.64 | -117.00 | Dfc: 85.2%; Dfb: 14.8% | ENF: 43.5%; WSA: 43.5%; GRA: 13% | 1980-2000 | 747-2136 | 15.00 | Laird Creek - CA |
| CA_0005502 | 49.59 | -117.24 | Dfc: 77.8%; Dsc: 11.1%; Dfb: 10.1% | WSA: 57.1%; ENF: 38.1%; GRA: 4.8% | 1980-2014 | 803-2106 | 52.90 | Duhamel Creek - CA |
| CA_0005503 | 49.59 | -117.06 | Dfc: 84.2%; Dfb: 11.8%; Dsc: 3.9% | ENF: 60.3%; WSA: 39.7% | 1980-1994 | 672-2183 | 42.20 | Harrop Creek - CA |
| CA_0005534 | 49.62 | -117.06 | Dfc: 93.6%; Dfb: 6.4% | WSA: 70.7%; ENF: 26.8%; GRA: 2.4% | 1980-2014 | 897-2175 | 27.20 | Redfish Creek - CA |
| CA_0005571 | 49.57 | -117.65 | Dfc: 53.8%; Dfb: 46.2% | ENF: 81.8%; WSA: 18.2% | 1980-1994 | 523-1933 | 5.70 | Mcfayden Creek - CA |
| CA_0005573 | 49.50 | -117.26 | Dfc: 70.6%; Dfb: 29.4% | ENF: 75%; WSA: 25% | 1980-2015 | 968-1794 | 9.07 | Anderson Creek - CA |



Continuation of Table A5

| Station | Lat | Lon | Climate | Biome | Period | Elev | Area | River |
|---|---|---|---|---|---|---|---|---|
| CA_0005598 | 49.70 | -117.45 | Dfc: 86.7%; Dfb: 9.6%; ET: 3.7% | ENF: 47.9%; WSA: 36.4%; GRA: 14% | 1980-2014 | 782-2462 | 181.00 | Lemon Creek - CA |
| CA_0005600 | 49.47 | -117.52 | Dfb: 62.5%; Dfc: 37.5% | WSA: 57.1%; ENF: 42.9% | 1980-1993 | 848-1637 | 5.59 | Smoky Creek - CA |
| CA_0005602 | 50.11 | -117.49 | Dfc: 84.6%; Dfb: 15.4% | ENF: 76.9%; WSA: 23.1% | 1980-1998 | 1074-1948 | 6.70 | Cadden Creek - CA |
| CA_0005605 | 49.48 | -117.36 | Dfc: 73.7%; Dfb: 26.3% | ENF: 87.5%; WSA: 12.5% | 1980-1998 | 1002-1890 | 11.10 | Sandy Creek - CA |
| CA_0005606 | 49.52 | -117.21 | Dfc: 100% | ENF: 45.7%; WSA: 42.9%; GRA: 11.4% | 1980-2014 | 1283-2180 | 47.70 | Five Mile Creek - CA |
| CA_0005622 | 49.87 | -114.87 | Dfc: 72.7%; ET: 27.3% | WSA: 42%; GRA: 28.6%; ENF: 21.4% | 1980-2015 | 1202-3040 | 1840.00 | Elk River - CA |
| CA_0005624 | 49.89 | -114.87 | Dfc: 76.5%; ET: 23.5% | WSA: 49.2%; GRA: 25.7%; ENF: 15.7% | 1980-2014 | 1235-2813 | 621.00 | Fording River - CA |
| CA_0005625 | 49.84 | -114.86 | Dfc: 97.5%; ET: 2.5% | WSA: 42.5%; ENF: 39.6%; GRA: 8.2% | 1980-1999 | 1214-2337 | 83.90 | Grave Creek - CA |
| CA_0005626 | 49.73 | -114.86 | Dfc: 96%; ET: 3.9%; Dfb: 0.1% | WSA: 44.8%; ENF: 33.8%; GRA: 15.2% | 1980-1996 | 1151-2509 | 637.00 | Michel Creek - CA |
| CA_0005627 | 50.20 | -114.88 | Dfc: 61.1%; ET: 38.9% | WSA: 46%; GRA: 35.4%; ENF: 10.6% | 1980-1995 | 1688-2813 | 104.00 | Fording River - CA |
| CA_0005628 | 49.89 | -114.83 | Dfc: 78.3%; ET: 21.7% | WSA: 39%; GRA: 33.3%; ENF: 22.1% | 1980-2014 | 1290-2725 | 138.00 | Line Creek - CA |
| CA_0005630 | 49.58 | -114.95 | Dfc: 100% | ENF: 88.9%; WSA: 11.1% | 1980-2014 | 1531-2036 | 6.40 | Hosmer Creek - CA |
| CA_0005631 | 50.38 | -114.92 | Dfc: 56.2%; ET: 43.8% | GRA: 35.9%; ENF: 26.7%; WSA: 21.8% | 1980-1996 | 1566-3040 | 334.00 | Elk River - CA |
| CA_0005632 | 49.51 | -114.68 | Dfc: 100% | ENF: 51%; GRA: 21.6%; WSA: 19.6% | 1980-1995 | 1532-2270 | 35.90 | Michel Creek - CA |
| CA_0005633 | 50.17 | -114.86 | Dfc: 61.4%; ET: 38.6% | BSV: 36.1%; WSA: 31.9%; GRA: 27.8% | 1980-1995 | 1630-2602 | 43.00 | Kilmarnock Creek - CA |
| CA_0005635 | 49.21 | -119.99 | Dfc: 62.2%; Dsc: 17.1%; ET: 14.3% | WSA: 39%; ENF: 37.1%; GRA: 23% | 1980-2013 | 488-2542 | 1050.00 | Ashnola River - CA |
| CA_0005638 | 49.46 | -120.50 | Dsc: 76.7%; Dsb: 11.6%; Dfc: 9.4% | ENF: 52.1%; WSA: 29.8%; GRA: 15.9% | 1980-2013 | 645-2390 | 1810.00 | Similkameen River - CA |
| CA_0005653 | 49.46 | -120.52 | Dsc: 59.5%; Dsb: 23%; Dfc: 9.8% | WSA: 48%; ENF: 45.8%; GRA: 5.9% | 1980-2013 | 660-2085 | 1780.00 | Tulameen River - CA |
| CA_0005658 | 49.37 | -120.57 | Dsc: 74.1%; Dsb: 22.5%; Dfc: 3.4% | ENF: 61.8%; WSA: 34%; GRA: 3.6% | 1980-1998 | 882-1917 | 185.00 | Whipsaw Creek - CA |
| CA_0005661 | 49.66 | -120.34 | Dfc: 79.6%; Dsc: 11.2%; BSk: 9.2% | WSA: 67.2%; GRA: 21.6%; ENF: 11.2% | 1980-2013 | 947-1759 | 263.00 | Siwash Creek - CA |
| CA_0005666 | 49.26 | -119.83 | BSk: 54.5%; Dfc: 40.5%; Dfb: 5% | WSA: 69.3%; ENF: 25.1%; GRA: 5% | 1980-2013 | 514-2156 | 181.00 | Keremeos Creek - CA |
| CA_0005668 | 49.36 | -120.07 | Dfc: 87.1%; BSk: 12.9% | WSA: 51.5%; ENF: 41.9%; GRA: 6.6% | 1980-2013 | 676-2105 | 388.00 | Hedley Creek - CA |
| CA_0005678 | 49.10 | -120.58 | Dsc: 92%; ET: 3.4%; Dfc: 2.8% | GRA: 38.1%; WSA: 29.7%; ENF: 29.2% | 1980-2013 | 1128-2390 | 566.00 | Pasayten River - CA |
| CA_0005679 | 49.09 | -120.67 | Dsc: 97.7%; Dsb: 1.6%; Dfc: 0.4% | ENF: 74.9%; WSA: 15.6%; GRA: 6.2% | 1980-2013 | 1116-2254 | 408.00 | Similkameen River - CA |
| CA_0005680 | 49.47 | -120.98 | Dsc: 88.8%; Dfc: 11.2% | ENF: 86.1%; WSA: 11.3%; GRA: 2.1% | 1980-2011 | 1141-2085 | 253.00 | Tulameen River - CA |



Continuation of Table A5

| Station | Lat | Lon | Climate | Biome | Period | Elev | Area | River |
|---|---|---|---|---|---|---|---|---|
| CA_0005703 | 50.30 | -119.21 | Dfb: 57.4%; Dfc: 42.6% | ENF: 62.4%; WSA: 31.8%; GRA: 3.5% | 1980-1999 | 638-1831 | 55.70 | Bx Creek - CA |
| CA_0005705 | 50.04 | -119.24 | Dfc: 100% | WSA: 74.7%; ENF: 20.7%; GRA: 4.6% | 1980-1996 | 1346-1620 | 62.40 | Vernon Creek - CA |
| CA_0005719 | 49.43 | -119.75 | Dfc: 57.6%; BSk: 41.9%; Dfb: 0.6% | ENF: 64.4%; WSA: 33.6%; GRA: 2.1% | 1980-2013 | 817-2079 | 101.00 | Shatford Creek - CA |
| CA_0005723 | 49.83 | -119.79 | Dfc: 64.4%; BSk: 20.1%; Dfb: 15.5% | ENF: 50.8%; WSA: 45.8%; GRA: 3.4% | 1980-2011 | 631-1860 | 182.00 | Trepanier Creek - CA |
| CA_0005733 | 49.89 | -119.42 | Dfc: 49.9%; BSk: 26.8%; Dfb: 23.3% | ENF: 37.3%; WSA: 30.5%; GRA: 20.2% | 1980-1996 | 386-1650 | 221.00 | Kelowna Creek - CA |
| CA_0005743 | 50.24 | -119.27 | Dfb: 39.8%; Dfc: 36.1%; BSk: 24.1% | WSA: 44.3%; ENF: 34.3%; GRA: 13.1% | 1980-2013 | 392-1620 | 569.00 | Vernon Creek - CA |
| CA_0005754 | 49.88 | -119.41 | Dfc: 76.3%; Dfb: 19.1%; BSk: 4.5% | ENF: 52.4%; WSA: 41.9%; GRA: 4.5% | 1980-2013 | 387-2060 | 795.00 | Mission Creek - CA |
| CA_0005768 | 49.71 | -120.01 | Dfc: 87.7%; BSk: 7.7%; Dfb: 4.6% | ENF: 58.9%; WSA: 41.1% | 1980-2014 | 1065-1841 | 34.60 | Camp Creek - CA |
| CA_0005772 | 50.07 | -119.67 | Dfc: 100% | WSA: 59.2%; ENF: 40.8% | 1980-1994 | 1363-1677 | 31.30 | Terrace Creek - CA |
| CA_0005776 | 50.26 | -119.08 | Dfb: 65.2%; Dfc: 34.8% | ENF: 67%; WSA: 25.8%; MF: 7.2% | 1980-2011 | 690-1620 | 60.60 | Coldstream Creek - CA |
| CA_0005777 | 50.01 | -119.25 | Dfc: 100% | WSA: 45%; ENF: 30%; GRA: 25% | 1980-2004 | 1314-1456 | 12.70 | Bulman Creek - CA |
| CA_0005780 | 49.34 | -119.58 | Dfc: 71.3%; BSk: 18%; Dfb: 10.7% | ENF: 48.7%; WSA: 43.3%; GRA: 6.7% | 1980-2010 | 373-1814 | 89.90 | Shuttleworth Creek - CA |
| CA_0005791 | 50.26 | -119.31 | Dfb: 43.9%; Dfc: 31.7%; BSk: 24.3% | WSA: 39.8%; ENF: 33.8%; GRA: 15.3% | 1980-1999 | 360-1771 | 751.00 | Vernon Creek - CA |
| CA_0005795 | 49.99 | -119.61 | Dfc: 84.4%; BSk: 15.6% | ENF: 68.7%; WSA: 31.3% | 1980-1996 | 922-1822 | 76.10 | Lambly Creek - CA |
| CA_0005798 | 49.62 | -119.42 | Dfc: 100% | WSA: 61.2%; ENF: 38.8% | 1980-1999 | 1571-1957 | 35.50 | Penticton Creek - CA |
| CA_0005800 | 49.25 | -119.32 | Dfc: 99.5%; BSk: 0.5% | ENF: 66.1%; WSA: 33.9% | 1980-2014 | 1277-2166 | 117.00 | Vaseux Creek - CA |
| CA_0005802 | 49.79 | -119.85 | Dfc: 57.5%; BSk: 39.7%; Dfb: 2.7% | ENF: 83.1%; WSA: 16.9% | 1980-2011 | 956-1645 | 40.70 | Greata Creek - CA |
| CA_0005803 | 50.21 | -119.54 | Dfc: 89.2%; Dfb: 9.8%; BSk: 1% | ENF: 74.4%; WSA: 23.9%; GRA: 1.7% | 1980-2013 | 735-1984 | 114.00 | Whiteman Creek - CA |
| CA_0005815 | 49.08 | -119.50 | Dfc: 46.1%; BSk: 32.3%; Dfb: 21.6% | WSA: 53.3%; ENF: 37.1%; GRA: 7.6% | 1980-2015 | 302-2134 | 227.00 | Inkaneep Creek - CA |
| CA_0005826 | 49.99 | -118.87 | Dfc: 100% | WSA: 65.4%; ENF: 34.6% | 1980-1998 | 1831-2060 | 16.10 | Loch Katrine Creek - CA |
| CA_0005827 | 50.00 | -119.07 | Dfc: 100% | WSA: 67%; ENF: 28.2%; GRA: 4.9% | 1980-2010 | 1276-1823 | 70.70 | Belgo Creek - CA |
| CA_0005839 | 49.17 | -118.98 | Dfc: 69.4%; Dfb: 23.3%; BSk: 7.3% | ENF: 51.4%; WSA: 47.7%; GRA: 0.9% | 1980-2014 | 646-2181 | 1890.00 | West Kettle River - CA |
| CA_0005851 | 49.70 | -119.09 | Dfc: 98.6%; Dfb: 0.9%; ET: 0.5% | WSA: 56.4%; ENF: 41.4%; GRA: 2.2% | 1980-2013 | 1043-2181 | 233.00 | West Kettle River - CA |
| CA_0005854 | 49.57 | -119.05 | Dfc: 78.3%; Dfb: 21.7% | ENF: 69.7%; WSA: 25.4%; GRA: 4.8% | 1980-2013 | 934-2170 | 145.00 | Trapping Creek - CA |
| CA_0005857 | 49.48 | -119.11 | Dfc: 81.4%; Dfb: 16.7%; BSk: 1.8% | ENF: 53.8%; WSA: 44.8%; GRA: 1.4% | 1980-1996 | 827-2181 | 1170.00 | West Kettle River - CA |



Continuation of Table A5

| Station | Lat | Lon | Climate | Biome | Period | Elev | Area | River |
|---|---|---|---|---|---|---|---|---|
| CA_0005858 | 49.59 | -118.31 | Dfc: 81.3%; Dfb: 18.7% | ENF: 58.6%; WSA: 39.7%; GRA: 1.7% | 1980-2012 | 924-2166 | 221.00 | Burrell Creek - CA |
| CA_0005861 | 49.37 | -118.85 | Dfc: 60.4%; Dfb: 39.6% | ENF: 65%; WSA: 35% | 1980-2011 | 808-2085 | 28.50 | Lost Horse Creek - CA |
| CA_0005862 | 49.00 | -114.48 | Dfc: 99.3%; Dfb: 0.4%; ET: 0.3% | ENF: 37.7%; WSA: 32%; GRA: 22.3% | 1980-2013 | 1219-2427 | 1110.00 | Flathead River - CA |
| CA_0005863 | 49.03 | -114.58 | Dfc: 100% | ENF: 44.3%; WSA: 31%; GRA: 14.6% | 1980-1992 | 1403-2175 | 118.00 | Couldrey Creek - CA |
| CA_0005864 | 49.09 | -114.54 | Dfc: 98.8%; ET: 1.2% | GRA: 47.3%; ENF: 25%; WSA: 24.1% | 1980-1996 | 1320-2336 | 145.00 | Howell Creek - CA |
| CA_0005865 | 49.09 | -114.55 | Dfc: 100% | ENF: 54.7%; GRA: 29.5%; WSA: 12.9% | 1980-2014 | 1338-2236 | 92.80 | Cabin Creek - CA |
| CA_0005880 | 60.08 | -133.86 | Dsc: 93.6%; ET: 4.7%; BSk: 1.7% | WSA: 79%; ENF: 17.3%; GRA: 2.4% | 1980-1993 | 645-1866 | 1770.00 | Lubbock River - CA |
| CA_0005883 | 59.84 | -135.01 | ET: 53.8%; Dsc: 46.2% | GRA: 42.8%; SNO: 30.8%; WSA: 9.9% | 1980-1993 | 669-1944 | 240.00 | Lindeman Creek - CA |
| CA_0005885 | 60.13 | -134.88 | ET: 61.6%; Dsc: 38.4% | GRA: 48.9%; BSV: 17.4%; WSA: 10.9% | 1980-2014 | 640-2139 | 864.00 | Wheaton River - CA |
| CA_0005886 | 59.95 | -134.33 | ET: 56%; Dsc: 44% | GRA: 43.7%; WSA: 18.7%; BSV: 12.9% | 1980-2014 | 715-1986 | 989.00 | Tutshi River - CA |
| CA_0005887 | 59.59 | -134.39 | ET: 70.2%; Dsc: 29.8% | GRA: 26.9%; SNO: 25.7%; BSV: 22.5% | 1980-1993 | 697-2103 | 717.00 | Fantail River - CA |
| CA_0005888 | 59.43 | -134.21 | ET: 83.2%; Dsc: 16.8% | GRA: 56.4%; ENF: 16.8%; WSA: 14.9% | 1980-1993 | 808-1931 | 269.00 | Wann River - CA |
| CA_0005891 | 60.61 | -134.46 | Dsc: 59.8%; ET: 19.5%; BSk: 19.1% | WSA: 66.3%; ENF: 24%; GRA: 5.8% | 1980-1995 | 656-1844 | 1700.00 | Mclintock River - CA |
| CA_0005895 | 60.73 | -135.49 | ET: 68.3%; Dsc: 31.7% | GRA: 44.4%; WSA: 29.5%; OSH: 22.5% | 1980-2014 | 859-2024 | 648.00 | Ibex River - CA |
| CA_0005900 | 59.91 | -132.91 | Dsc: 54.2%; ET: 45.8% | WSA: 43.2%; GRA: 30.5%; ENF: 19.5% | 1980-1993 | 872-1934 | 1910.00 | Gladys River - CA |
| CA_0005901 | 60.01 | -132.14 | Dsc: 96.7%; Dfb: 3.3% | ENF: 52%; WSA: 48% | 1980-2014 | 812-1238 | 1580.00 | Morely River - CA |
| CA_0005904 | 61.39 | -134.37 | ET: 51.6%; Dsc: 39.7%; Dfc: 8.7% | WSA: 60.3%; GRA: 25.6%; OSH: 9.3% | 1980-1996 | 780-1965 | 515.00 | South Big Salmon River - CA |
| CA_0005906 | 62.57 | -137.01 | Dfc: 88%; ET: 12% | WSA: 47.6%; SAV: 34.1%; OSH: 17.1% | 1980-2014 | 506-1876 | 1800.00 | Big Creek - CA |
| CA_0005908 | 62.20 | -134.39 | Dsc: 53.7%; ET: 30.8%; Dfc: 15.5% | WSA: 66.4%; ENF: 11.1%; GRA: 10.4% | 1980-2009 | 633-1934 | 552.00 | Drury Creek - CA |
| CA_0005911 | 62.92 | -130.54 | ET: 60.9%; Dfc: 36.2%; Dsc: 2.9% | SAV: 38.3%; GRA: 22.3%; OSH: 18% | 1980-1996 | 939-2290 | 997.00 | South Macmillan River - CA |
| CA_0005920 | 61.35 | -139.17 | ET: 95.1%; Dfc: 4.9% | GRA: 41.1%; BSV: 37.9%; OSH: 8.7% | 1980-2014 | 901-2916 | 654.00 | Duke River - CA |
| CA_0005935 | 64.00 | -137.57 | Dfc: 89.7%; ET: 9.6%; Dsc: 0.6% | WSA: 52.1%; SAV: 32.3%; OSH: 13.4% | 1980-1994 | 658-1736 | 860.00 | Little South Klondike River - CA |
| CA_0005950 | 60.29 | -129.02 | Dfc: 99.6%; ET: 0.4% | ENF: 75.8%; WSA: 24.2% | 1980-1993 | 725-1529 | 435.00 | Tom Creek - CA |
| CA_0005957 | 59.76 | -129.13 | Dsc: 58.4%; ET: 38.6%; Dfc: 2.9% | ENF: 37.6%; GRA: 29.6%; WSA: 27.5% | 1980-1995 | 719-2101 | 1700.00 | Blue River - CA |
| CA_0005958 | 59.12 | -129.83 | ET: 58.6%; Dsc: 41.4% | GRA: 47.8%; WSA: 34.5%; ENF: 5.3% | 1980-2013 | 815-1998 | 882.00 | Cottonwood River - CA |
| CA_0005972 | 59.34 | -125.94 | ET: 48%; Dfc: 47.9%; Dsc: 4.1% | WSA: 41.7%; GRA: 34.7%; ENF: 15.7% | 1980-2014 | 541-2142 | 1170.00 | Trout River - CA |



Continuation of Table A5

| Station | Lat | Lon | Climate | Biome | Period | Elev | Area | River |
|---|---|---|---|---|---|---|---|---|
| CA_0005973 | 59.60 | -126.67 | Dfc: 100% | WSA: 79.6%; ENF: 19.7%; MF: 0.7% | 1980-1996 | 568-1176 | 77.80 | Geddes Creek - CA |
| CA_0005974 | 59.45 | -126.23 | Dfc: 100% | MF: 45.9%; ENF: 35.9%; WSA: 17% | 1980-2014 | 475-1473 | 209.00 | Teeter Creek - CA |
| CA_0005976 | 59.37 | -125.07 | Dfc: 98.9%; ET: 1.1% | ENF: 67.7%; WSA: 18.9%; MF: 9.2% | 1980-1995 | 390-1685 | 1780.00 | Grayling River - CA |
| CA_0005986 | 58.03 | -122.72 | Dfc: 100% | ENF: 78.5%; MF: 19.8%; DBF: 1.6% | 1980-2012 | 529-1093 | 335.00 | Bougie Creek - CA |
| CA_0005987 | 58.11 | -122.72 | Dfc: 100% | ENF: 73.2%; MF: 23.5%; DBF: 2.3% | 1980-2014 | 552-1077 | 109.00 | Adsett Creek - CA |
| CA_0005994 | 62.21 | -128.76 | ET: 84%; Dfc: 16% | GRA: 41.2%; OSH: 34.3%; SAV: 20.8% | 1980-1992 | 1020-2138 | 216.00 | Mac Creek - CA |
| CA_0005995 | 62.37 | -128.68 | ET: 98.9%; Dfc: 1.1% | GRA: 46.2%; BSV: 30%; SAV: 11.2% | 1980-1992 | 1210-2089 | 34.30 | Lened Creek - CA |
| CA_0005997 | 61.56 | -124.81 | Dfc: 54.1%; ET: 45.9% | GRA: 61.2%; SAV: 19.8%; OSH: 17.3% | 1980-2014 | 896-1840 | 495.00 | Prairie Creek - CA |
| CA_0006022 | 63.30 | -129.79 | ET: 100% | OSH: 43.1%; GRA: 40.6%; BSV: 15.9% | 1980-1992 | 1247-2121 | 219.00 | Tsichu River - CA |
| CA_0006027 | 62.77 | -126.69 | ET: 90.2%; Dfc: 9.8% | GRA: 42.4%; BSV: 38.4%; OSH: 16.6% | 1980-1990 | 901-2413 | 1420.00 | Silverberry River - CA |
| CA_0006072 | 64.90 | -138.28 | ET: 57.1%; Dfc: 41.5%; Dsc: 1.4% | OSH: 71.1%; GRA: 25.6%; BSV: 3.1% | 1980-2014 | 940-2108 | 1180.00 | Blackstone River - CA |
| CA_0006242 | 49.65 | -110.01 | Dfc: 100% | GRA: 48.2%; ENF: 32.7%; CRO: 12.5% | 1980-2014 | 1186-1436 | 111.00 | Battle Creek - CA |
| US_0000044 | 45.23 | -70.20 | Dfb: 96.3%; Dfc: 3.7% | MF: 56.2%; DBF: 41.7%; ENF: 1.4% | 1980-2016 | 344-1192 | 516.00 | Dead River - US |
| US_0000045 | 45.31 | -70.24 | Dfb: 94.9%; Dfc: 5.1% | MF: 66.5%; DBF: 32.1%; WSA: 1.3% | 1980-2016 | 326-1024 | 193.00 | Spencer Stream - US |
| US_0000051 | 44.86 | -70.49 | Dfb: 100% | DBF: 69.8%; MF: 30.2% | 1980-2016 | 310-900 | 25.30 | Sandy River - US |
| US_0000057 | 44.88 | -71.06 | Dfb: 98.3%; Dfc: 1.7% | DBF: 83%; MF: 17% | 1980-2016 | 412-1004 | 152.00 | Diamond River - US |
| US_0000058 | 44.78 | -71.13 | Dfb: 96.9%; Dfc: 3.1% | DBF: 50%; MF: 45.6%; ENF: 3.7% | 1980-2016 | 378-1092 | 1046.00 | Androscoggin River - US |
| US_0000059 | 44.67 | -71.18 | Dfb: 97.2%; Dfc: 2.8% | DBF: 53.1%; MF: 42.8%; ENF: 3.3% | 1980-2016 | 363-1092 | 1177.00 | Androscoggin River Below Bog Brook - US |
| US_0000060 | 44.44 | -71.19 | Dfb: 97.5%; Dfc: 2.5% | DBF: 54.9%; MF: 40.5%; ENF: 2.8% | 1980-2016 | 275-1101 | 1361.00 | Androscoggin River - US |
| US_0000314 | 42.08 | -73.07 | Dfb: 100% | DBF: 80.8%; MF: 19.2% | 1980-2016 | 250-582 | 91.70 | West Branch Farmington River - US |
| US_0000338 | 42.47 | -73.20 | Dfb: 100% | DBF: 79%; WSA: 12.7%; MF: 4.9% | 1980-2016 | 321-672 | 57.60 | East Branch Housatonic River - US |
| US_0000433 | 42.32 | -74.44 | Dfb: 100% | DBF: 86.1%; WSA: 9%; MF: 4.9% | 1980-2016 | 359-1167 | 237.00 | Schoharie Creek - US |
| US_0000437 | 42.41 | -74.45 | Dfb: 100% | DBF: 72.5%; MF: 15%; WSA: 12.5% | 1980-2016 | 387-668 | 10.90 | Platter Kill - US |
| US_0000438 | 42.43 | -74.47 | Dfb: 100% | DBF: 91.8%; WSA: 6.6%; MF: 1.6% | 1980-2016 | 369-796 | 16.20 | Mine Kill - US |
| US_0000458 | 41.87 | -74.49 | Dfb: 100% | DBF: 100% | 1980-2016 | 328-1084 | 38.30 | Rondout Creek - US |
| US_0000585 | 42.14 | -74.65 | Dfb: 100% | DBF: 95.8%; WSA: 4.2% | 1980-2016 | 405-1084 | 163.00 | East Branch Delaware River - US |
| US_0000587 | 42.11 | -74.73 | Dfb: 100% | DBF: 100% | 1980-2016 | 437-1018 | 25.20 | Mill Brook - US |



Continuation of Table A5

| Station | Lat | Lon | Climate | Biome | Period | Elev | Area | River |
|---|---|---|---|---|---|---|---|---|
| US_0000588 | 42.12 | -74.82 | Dfb: 100% | DBF: 92.3%; WSA: 7.7% | 1980-2016 | 421-942 | 33.20 | Tremper Kill - US |
| US_0000827 | 42.39 | -77.36 | Dfb: 100% | WSA: 51.6%; DBF: 38.2%; SAV: 4.1% | 1980-1995 | 367-656 | 66.80 | Fivemile Creek - US |
| US_0000987 | 39.27 | -79.26 | Dfb: 100% | DBF: 60.5%; WSA: 33.3%; GRA: 5.6% | 1980-2010 | 804-1236 | 48.70 | Stony River - US |
| US_0000988 | 39.37 | -79.18 | Dfb: 100% | DBF: 71.1%; WSA: 28.1%; SAV: 0.7% | 1980-2016 | 620-1015 | 42.60 | Abram Creek - US |
| US_0001004 | 38.98 | -79.23 | Dfb: 92.9%; Dfa: 7.1% | DBF: 91.7%; WSA: 5.5%; SAV: 1.8% | 1980-2016 | 378-1444 | 310.00 | N F South Branch Potomac River - US |
| US_0001005 | 38.99 | -79.18 | Dfb: 88%; Dfa: 12% | DBF: 85.1%; WSA: 7.2%; SAV: 4.4% | 1980-2016 | 308-1444 | 651.00 | South Branch Potomac River - US |
| US_0001007 | 39.01 | -78.96 | Dfb: 55.2%; Dfa: 44.8% | DBF: 88.3%; WSA: 7%; SAV: 4.4% | 1980-2016 | 273-1213 | 277.00 | S F South Branch Potomac River - US |
| US_0001568 | 34.81 | -83.31 | Cfa: 74.7%; Cfb: 24.4%; Dfb: 0.9% | MF: 58.4%; DBF: 39.6%; WSA: 2% | 1980-2016 | 410-1361 | 207.00 | Chattooga River - US |
| US_0001569 | 34.89 | -83.53 | Cfa: 65.9%; Cfb: 27.3%; Dfb: 6.8% | DBF: 86.8%; MF: 12.7%; WSA: 0.5% | 1980-2016 | 607-1571 | 58.40 | Tallulah River - US |
| US_0001570 | 34.73 | -83.38 | Cfa: 85.9%; Cfb: 11.6%; Dfb: 2.5% | DBF: 68%; MF: 27.6%; WSA: 4.2% | 1980-2016 | 448-1571 | 184.40 | Tallulah River - US |
| US_0001942 | 34.70 | -83.73 | Cfa: 90.2%; Cfb: 9.8% | DBF: 55.3%; MF: 43.3%; WSA: 1.4% | 1980-2016 | 461-1135 | 44.70 | Chattahoochee River - US |
| US_0002093 | 34.57 | -84.47 | Cfa: 100% | DBF: 83.3%; WSA: 16.7% | 1980-2015 | 432-755 | 9.99 | Fausett Creek - US |
| US_0002122 | 34.26 | -84.60 | Cfa: 100% | DBF: 67.6%; WSA: 29.8%; SAV: 1.6% | 1980-2016 | 291-674 | 56.50 | Shoal Creek - US |
| US_0002372 | 38.81 | -79.88 | Dfb: 100% | DBF: 91%; GRA: 4.3%; WSA: 4.2% | 1980-2016 | 596-1406 | 185.00 | Tygart Valley River - US |
| US_0002373 | 38.92 | -79.88 | Dfb: 100% | DBF: 86.3%; WSA: 7.5%; GRA: 4.7% | 1980-2004 | 583-1406 | 271.00 | Tygart Valley River - US |
| US_0002374 | 39.03 | -79.94 | Dfb: 100% | DBF: 83.3%; WSA: 11.6%; GRA: 3.9% | 1980-2016 | 522-1406 | 406.00 | Tygart Valley River - US |
| US_0002375 | 39.04 | -80.07 | Dfb: 98.2%; Dfa: 1.8% | DBF: 98.6%; WSA: 1.4% | 1980-2016 | 554-1104 | 148.00 | Middle Fork River - US |
| US_0002390 | 39.07 | -79.62 | Dfb: 100% | DBF: 88.8%; WSA: 6.5%; MF: 2.7% | 1980-2016 | 587-1422 | 349.00 | Dry Fork - US |
| US_0002391 | 39.14 | -79.42 | Dfb: 100% | DBF: 62.4%; WSA: 31.4%; MF: 3.1% | 1980-2016 | 958-1298 | 54.70 | Blackwater River - US |
| US_0002392 | 39.13 | -79.47 | Dfb: 100% | DBF: 55.2%; WSA: 39.3%; MF: 3.1% | 1980-2016 | 942-1298 | 85.90 | Blackwater River - US |
| US_0002395 | 39.10 | -79.68 | Dfb: 99.9%; Dfa: 0.1% | DBF: 86.1%; MF: 13%; WSA: 0.9% | 1980-1993 | 503-1436 | 213.00 | Shavers Fork - US |
| US_0002396 | 39.12 | -79.68 | Dfb: 99.8%; Dfa: 0.2% | DBF: 81.8%; WSA: 10.1%; MF: 6.7% | 1980-2016 | 493-1436 | 722.00 | Cheat River - US |
| US_0002397 | 39.35 | -79.67 | Dfb: 98.5%; Dfa: 1.5% | DBF: 84.8%; WSA: 9.1%; MF: 5.1% | 1980-1996 | 443-1436 | 939.00 | Cheat River - US |
| US_0002548 | 38.54 | -79.83 | Dfb: 100% | DBF: 96.7%; MF: 1.5%; GRA: 0.9% | 1980-2015 | 838-1337 | 133.00 | Greenbrier River - US |
| US_0002549 | 38.19 | -80.13 | Dfb: 100% | DBF: 92.7%; WSA: 3.4%; SAV: 2.2% | 1980-2016 | 645-1448 | 540.00 | Greenbrier River - US |



Continuation of Table A5

| Station | Lat | Lon | Climate | Biome | Period | Elev | Area | River |
|---|---|---|---|---|---|---|---|---|
| US_0002550 | 37.68 | -80.46 | Dfb: 73.2%; Dfa: 26.8% | DBF: 69%; WSA: 13.1%; GRA: 10.1% | 1980-1998 | 572-1078 | 80.80 | Second Creek - US |
| US_0002556 | 38.38 | -80.48 | Dfb: 100% | DBF: 93.7%; MF: 5.3%; GRA: 0.7% | 1980-2016 | 696-1374 | 128.00 | Williams River - US |
| US_0002558 | 38.30 | -80.53 | Dfb: 100% | DBF: 90.2%; MF: 9.8% | 1980-2016 | 735-1369 | 80.40 | Cranberry River - US |
| US_0002560 | 38.29 | -80.64 | Dfb: 100% | DBF: 95%; MF: 2.9%; WSA: 1.2% | 1980-2016 | 619-1374 | 529.00 | Gauley River - US |
| US_0002561 | 38.22 | -80.89 | Dfb: 92.9%; Dfa: 7.1% | DBF: 91.8%; WSA: 4.5%; MF: 1.9% | 1980-2003 | 479-1374 | 806.00 | Gauley River - US |
| US_0002564 | 38.26 | -81.02 | Dfa: 88.6%; Dfb: 11.4% | DBF: 90.9%; WSA: 7.7%; GRA: 0.7% | 1980-2016 | 387-727 | 40.20 | Peters Creek - US |
| US_0002565 | 38.23 | -81.18 | Dfb: 79.5%; Dfa: 20.5% | DBF: 90.4%; WSA: 6.3%; GRA: 1.7% | 1980-2016 | 257-1374 | 1317.00 | Gauley River - US |
| US_0002567 | 38.60 | -80.49 | Dfb: 93.1%; Dfa: 6.9% | DBF: 99.3%; WSA: 0.4%; MF: 0.2% | 1980-2016 | 370-1434 | 266.00 | Elk River - US |
| US_0002573 | 37.97 | -81.52 | Dfa: 77.3%; Dfb: 22.7% | DBF: 89.8%; GRA: 9.3%; WSA: 0.9% | 1980-2016 | 321-902 | 62.80 | Clear Fork - US |
| US_0002593 | 37.43 | -82.35 | Cfa: 100% | DBF: 100% | 1980-2016 | 329-580 | 6.20 | Grapevine Creek - US |
| US_0003020 | 36.75 | -83.26 | Cfa: 92.7%; Dfb: 7.3% | DBF: 98.9%; WSA: 1.1% | 1980-2004 | 406-984 | 55.80 | Martins Fork - US |
| US_0003103 | 35.96 | -83.17 | Dfb: 39.1%; Cfa: 33%; Cfb: 27.9% | DBF: 79.4%; WSA: 10.2%; SAV: 6.5% | 1980-2016 | 330-1883 | 666.00 | Pigeon River - US |
| US_0003106 | 36.18 | -82.46 | Dfb: 46.4%; Cfa: 34%; Cfb: 19.5% | DBF: 87.7%; WSA: 9.1%; SAV: 1.8% | 1980-2016 | 483-1938 | 805.00 | Nolichucky River - US |
| US_0003165 | 36.43 | -83.40 | Cfa: 65.5%; Dfb: 18.3%; Dfa: 16.2% | DBF: 70.8%; WSA: 16.3%; SAV: 10.9% | 1980-2016 | 352-1386 | 1474.00 | Clinch River - US |
| US_0003169 | 36.54 | -83.63 | Cfa: 87.3%; Dfb: 7.9%; Dfa: 4.8% | DBF: 53.5%; WSA: 31.2%; SAV: 12% | 1980-2016 | 350-1137 | 685.00 | Powell River - US |
| US_0003181 | 34.88 | -83.72 | Cfa: 78.4%; Cfb: 21.6% | DBF: 76.8%; MF: 19.2%; WSA: 4% | 1980-2016 | 613-1223 | 39.50 | Hiwassee River - US |
| US_0003186 | 34.84 | -83.94 | Cfa: 91%; Cfb: 8.6%; Dfb: 0.3% | DBF: 64.1%; MF: 19.2%; WSA: 16.3% | 1980-2016 | 573-1319 | 74.80 | Nottely River - US |
| US_0004649 | 44.08 | -104.06 | Dfb: 93.3%; Dfc: 6.7% | WSA: 52.6%; GRA: 23.7%; SAV: 23.7% | 1980-2016 | 1918-2117 | 10.20 | Beaver Creek - US |
| US_0004650 | 43.86 | -104.11 | Dfb: 98.1%; Dfc: 1.7%; BSk: 0.2% | GRA: 53.4%; SAV: 23.9%; WSA: 18.8% | 1980-2016 | 1377-2117 | 110.00 | Stockade Beaver Creek - US |
| US_0004657 | 43.43 | -103.48 | Dfb: 49.5%; BSk: 49.5%; Dfa: 1.1% | GRA: 97.7%; URB: 0.8%; SAV: 0.8% | 1980-2016 | 1070-1720 | 136.00 | Fall River - US |
| US_0004658 | 43.58 | -103.48 | Dfb: 100% | GRA: 77.6%; ENF: 9.4%; SAV: 8.8% | 1980-2015 | 1301-1819 | 45.60 | Beaver Creek - US |
| US_0004659 | 43.47 | -103.31 | Dfb: 64.4%; BSk: 30.8%; Dfa: 4.9% | GRA: 90.3%; ENF: 4.5%; SAV: 3.2% | 1980-2016 | 973-1819 | 127.00 | Beaver Creek - US |
| US_0004661 | 43.72 | -103.37 | Dfb: 100% | GRA: 78%; WSA: 11.7%; SAV: 6% | 1980-2016 | 1244-2075 | 105.00 | French Creek - US |
| US_0004663 | 43.87 | -103.34 | Dfb: 89.5%; Dwb: 10.5% | WSA: 59.3%; GRA: 34%; ENF: 6.6% | 1980-2016 | 1177-2008 | 58.50 | Battle Creek - US |
| US_0004664 | 43.76 | -103.36 | Dfb: 100% | GRA: 54.3%; WSA: 39%; ENF: 6.7% | 1980-2016 | 1270-1731 | 26.80 | Grace Coolidge Creek - US |
| US_0004667 | 43.98 | -103.35 | Dfb: 88.9%; Dwb: 10.6%; Dfc: 0.4% | WSA: 75.5%; GRA: 20.9%; ENF: 2.5% | 1980-2016 | 1224-2162 | 163.00 | Spring Creek - US |




Continuation of Table A5

| Station | Lat | Lon | Climate | Biome | Period | Elev | Area | River |
|---|---|---|---|---|---|---|---|---|
| US_0004670 | 44.13 | -103.86 | Dfb: 70.6%; Dfc: 29.4% | WSA: 100% | 1980-2016 | 1937-2159 | 7.84 | Rhoads Fork - US |
| US_0004673 | 44.08 | -103.58 | Dfb: 89.1%; Dfc: 10.9% | WSA: 81.4%; GRA: 17%; SAV: 1.1% | 1980-2016 | 1466-2166 | 293.00 | Rapid Creek Above Pactola Reservoir - US |
| US_0004674 | 44.08 | -103.48 | Dfb: 89.6%; Dfc: 9.9%; Dwb: 0.4% | WSA: 82.4%; GRA: 15.5%; SAV: 1% | 1980-2016 | 1398-2166 | 321.00 | Rapid Creek - US |
| US_0004675 | 44.05 | -103.31 | Dfb: 79.9%; Dwb: 11.2%; Dfc: 8.6% | WSA: 82.3%; GRA: 15.4%; SAV: 0.9% | 1980-2016 | 1077-2166 | 374.00 | Rapid Creek - US |
| US_0004679 | 44.14 | -103.45 | Dfb: 100% | WSA: 82.5%; ENF: 9.8%; GRA: 7.7% | 1980-2016 | 1358-1955 | 94.20 | Boxelder Creek - US |
| US_0004680 | 44.13 | -103.30 | Dfb: 84.5%; Dwb: 13.3%; Dwa: 2.2% | WSA: 77%; GRA: 11.6%; ENF: 11.4% | 1980-2009 | 1063-1955 | 127.00 | Boxelder Creek - US |
| US_0004693 | 44.52 | -104.08 | Dfb: 94.7%; Dfa: 3.4%; Dfc: 1.9% | WSA: 61.1%; GRA: 35%; ENF: 2.2% | 1980-2016 | 1108-2138 | 274.00 | Sand Creek - US |
| US_0004696 | 44.30 | -103.87 | Dfb: 94.7%; Dfc: 5.3% | WSA: 93%; GRA: 6.6%; ENF: 0.4% | 1980-2016 | 1656-2157 | 64.80 | Spearfish Creek - US |
| US_0004698 | 44.35 | -103.94 | Dfb: 100% | WSA: 97.1%; GRA: 2.9% | 1980-2016 | 1626-2063 | 27.80 | Little Spearfish Creek - US |
| US_0004699 | 44.40 | -103.89 | Dfb: 97.5%; Dfc: 2.5% | WSA: 94.2%; GRA: 5.4%; ENF: 0.4% | 1980-2009 | 1407-2157 | 143.00 | Spearfish Creek - US |
| US_0004700 | 44.48 | -103.86 | Dfb: 95.2%; Dfa: 2.6%; Dfc: 2.2% | WSA: 91%; GRA: 7.9%; ENF: 0.7% | 1980-2016 | 1141-2157 | 165.00 | Spearfish Creek - US |
| US_0004705 | 44.44 | -103.63 | Dfb: 100% | WSA: 65.7%; GRA: 28.2%; URB: 2.8% | 1980-2016 | 1127-2080 | 56.50 | Whitewood Creek - US |
| US_0004711 | 44.34 | -103.64 | Dfb: 100% | WSA: 85%; GRA: 15% | 1980-2016 | 1524-1820 | 15.70 | Bear Butte Creek - US |
| US_0004887 | 39.16 | -105.31 | Dfc: 51.7%; BSk: 31.9%; Dfb: 8.9% | GRA: 86.6%; WSA: 7.5%; SAV: 5% | 1980-2016 | 2177-4227 | 1615.10 | South Platte River - US |
| US_0004889 | 39.21 | -105.27 | Dfc: 52.1%; BSk: 29.6%; Dfb: 11.2% | GRA: 84.1%; WSA: 9.5%; SAV: 5.4% | 1980-2007 | 2086-4227 | 1740.10 | South Platte River - US |
| US_0004890 | 39.17 | -105.12 | Dfb: 66.2%; Dfc: 33.8% | GRA: 44.8%; WSA: 36.2%; SAV: 15% | 1980-2016 | 2307-3089 | 106.00 | Trout Creek - US |
| US_0004905 | 39.65 | -105.17 | Dfb: 54.9%; Dfc: 38.7%; ET: 5.5% | WSA: 60.4%; GRA: 21.4%; ENF: 12% | 1980-2016 | 1740-4009 | 176.00 | Bear Creek - US |
| US_0005681 | 36.68 | -104.79 | Dfb: 58.4%; BSk: 24.6%; Dfc: 16.6% | GRA: 68.8%; SAV: 27.1%; WSA: 3.7% | 1980-2016 | 1978-3818 | 301.00 | Vermejo River - US |
| US_0005682 | 36.55 | -105.27 | Dfc: 51.8%; Dfb: 48.2% | GRA: 83.6%; WSA: 15.6%; ENF: 0.8% | 1980-2010 | 2512-3647 | 73.80 | Moreno Creek - US |
| US_0005683 | 36.49 | -105.27 | Dfb: 89.7%; Dfc: 10.3% | GRA: 59.2%; WSA: 40.3%; SAV: 0.4% | 1980-2010 | 2512-3360 | 56.00 | Cieneguilla Creek - US |
| US_0005684 | 36.52 | -105.28 | Dfc: 67.3%; Dfb: 32.7% | WSA: 76.2%; GRA: 23.8% | 1980-2010 | 2512-3578 | 10.50 | Sixmile Creek - US |
| US_0005685 | 36.53 | -105.23 | Dfb: 68.8%; Dfc: 31.2% | GRA: 68%; WSA: 31.4%; ENF: 0.3% | 1980-2016 | 2497-3647 | 167.00 | Cimarron River - US |
| US_0005686 | 36.52 | -104.98 | Dfb: 68.1%; Dfc: 28.9%; BSk: 3% | GRA: 62.3%; WSA: 35.9%; SAV: 1.3% | 1980-2016 | 2038-3663 | 294.00 | Cimarron River - US |
| US_0005687 | 36.57 | -104.95 | Dfb: 65.3%; Dfc: 21.4%; BSk: 13.4% | GRA: 78.8%; SAV: 12.4%; WSA: 8.8% | 1980-2016 | 2081-3671 | 171.00 | Ponil Creek - US |
| US_0005694 | 35.92 | -105.16 | Dfb: 90.7%; Dfc: 6.8%; BSk: 2.5% | GRA: 73.1%; WSA: 26.4%; ENF: 0.4% | 1980-2016 | 2090-3303 | 215.00 | Coyote Creek - US |
| US_0006466 | 38.22 | -106.09 | Dfc: 75.1%; BSk: 18.6%; ET: 6.2% | GRA: 63.8%; WSA: 28.9%; SAV: 7.2% | 1980-2007 | 2662-3781 | 45.40 | Kerber Creek - US |





Continuation of Table A5

| Station | Lat | Lon | Climate | Biome | Period | Elev | Area | River |
|---|---|---|---|---|---|---|---|---|
| US_0006486 | 36.90 | -105.25 | Dfc: 98.9%; ET: 1.1% | GRA: 48%; SAV: 26.7%; WSA: 22.7% | 1980-2016 | 2889-3864 | 25.10 | Costilla Creek - US |
| US_0006487 | 36.90 | -105.26 | Dfc: 78.5%; ET: 21.5% | GRA: 58.2%; SAV: 28.4%; WSA: 13.4% | 1980-2016 | 2905-3832 | 16.60 | Casias Creek - US |
| US_0006489 | 36.87 | -105.28 | Dfc: 91.6%; ET: 8.4% | GRA: 54.2%; SAV: 25.3%; WSA: 19.5% | 1980-2016 | 2868-3832 | 54.60 | Costilla Creek - US |
| US_0006490 | 36.97 | -105.51 | Dfc: 74.1%; Dfb: 17.9%; BSk: 5.1% | GRA: 57.5%; SAV: 22.3%; WSA: 18.5% | 1980-2016 | 2458-3865 | 195.00 | Costilla Creek - US |
| US_0006494 | 36.70 | -105.57 | Dfc: 72%; Dfb: 25.3%; BSk: 1.9% | WSA: 43.4%; GRA: 38.1%; ENF: 9.8% | 1980-2016 | 2338-3877 | 113.00 | Red River - US |
| US_0006496 | 36.68 | -105.65 | Dfc: 61.3%; Dfb: 22%; BSk: 16.2% | GRA: 45.2%; WSA: 39.9%; SAV: 7.3% | 1980-2016 | 2219-3877 | 185.00 | Red River - US |
| US_0006498 | 36.54 | -105.56 | Dfc: 84.8%; Dfb: 15.2% | WSA: 63.3%; GRA: 25.8%; ENF: 7% | 1980-2016 | 2477-3795 | 36.20 | River Hondo - US |
| US_0006501 | 36.44 | -105.50 | Dfb: 51.8%; Dfc: 46.6%; BSk: 1.6% | WSA: 79.8%; GRA: 10.8%; ENF: 8.9% | 1980-2016 | 2337-3693 | 66.60 | River Pueblo De Taos - US |
| US_0006502 | 36.51 | -105.53 | Dfc: 82.1%; Dfb: 14.9%; ET: 3% | WSA: 48.3%; GRA: 36.2%; SAV: 8.6% | 1980-2016 | 2577-3829 | 16.60 | River Lucero - US |
| US_0006504 | 36.30 | -105.58 | Dfb: 66.4%; Dfc: 26.2%; BSk: 7.4% | WSA: 80%; ENF: 11%; GRA: 8.6% | 1980-2016 | 2229-3568 | 83.00 | River Grande Del Rancho - US |
| US_0006508 | 36.17 | -105.60 | Dfc: 53.6%; Dfc: 46.4% | WSA: 72.1%; GRA: 14.4%; ENF: 13.5% | 1980-2016 | 2470-3732 | 101.00 | River Pueblo - US |
| US_0006509 | 36.21 | -105.91 | Dfb: 43.3%; BSk: 28.9%; Dfc: 27.7% | WSA: 55.7%; GRA: 28.7%; ENF: 7.6% | 1980-2016 | 1823-3828 | 305.00 | Embudo Creek - US |
| US_0006515 | 36.32 | -106.60 | Dfb: 46.7%; BSk: 37%; Dfc: 16.1% | GRA: 76.4%; SAV: 15.1%; WSA: 7.5% | 1980-2016 | 1928-3723 | 1500.00 | River Chama - US |
| US_0006517 | 36.35 | -106.04 | Dfb: 45.2%; BSk: 42.1%; Dfc: 12.6% | GRA: 67.2%; SAV: 16.6%; WSA: 16.1% | 1980-2016 | 1953-3241 | 419.00 | River Ojo Caliente - US |
| US_0006519 | 35.96 | -105.90 | Dfb: 42.7%; Dfc: 33.2%; BSk: 24.1% | GRA: 50.2%; WSA: 26.1%; ENF: 12.4% | 1980-2016 | 2031-3674 | 86.00 | Santa Cruz River - US |
| US_0006520 | 35.85 | -105.89 | Dfb: 62.4%; Dfc: 30.4%; BSk: 7.2% | GRA: 73.8%; WSA: 15.9%; ENF: 5.6% | 1980-2012 | 2141-3686 | 25.00 | River Nambe - US |
| US_0006521 | 35.85 | -105.91 | Dfb: 60.2%; Dfc: 27.8%; BSk: 12% | GRA: 69.6%; WSA: 14.8%; SAV: 10.4% | 1980-2016 | 2043-3686 | 34.10 | River Nambe - US |
| US_0006522 | 35.74 | -105.90 | Dfb: 73.3%; Dfc: 15.6%; BSk: 11.1% | GRA: 42.5%; ENF: 20%; WSA: 20% | 1980-2016 | 2203-3495 | 11.70 | Tesuque Creek - US |
| US_0006523 | 35.73 | -105.91 | Dfb: 86.7%; BSk: 13.3% | GRA: 60.9%; WSA: 39.1% | 1980-2009 | 2251-3101 | 7.61 | Little Tesuque Creek - US |
| US_0006526 | 35.69 | -105.82 | Dfb: 58.5%; Dfc: 41.5% | GRA: 70.2%; ENF: 12.8%; WSA: 8.5% | 1980-2016 | 2525-3578 | 13.50 | Santa Fe River - US |
| US_0006527 | 35.69 | -105.84 | Dfb: 69.9%; Dfc: 30.1% | GRA: 78.5%; ENF: 9.2%; WSA: 6.2% | 1980-2016 | 2426-3578 | 18.20 | Santa Fe River - US |
| US_0006528 | 35.55 | -106.23 | BSk: 82.9%; Dfb: 14.7%; Dfc: 2.5% | GRA: 83.6%; SAV: 7.7%; URB: 5.6% | 1980-2016 | 1697-3578 | 231.00 | Santa Fe River - US |
| US_0006534 | 35.73 | -106.76 | Dfb: 78.5%; BSk: 13.5%; Dfc: 8.1% | WSA: 55.1%; GRA: 43.9%; SAV: 0.7% | 1980-1996 | 1900-3208 | 235.00 | River Guadalupe - US |
| US_0006560 | 35.78 | -105.66 | Dfc: 79%; Dfb: 21% | WSA: 76.9%; GRA: 13.6%; ENF: 9.5% | 1980-2016 | 2515-3634 | 53.20 | River Mora - US |
| US_0006561 | 35.71 | -105.68 | Dfc: 69.9%; Dfb: 30.1% | WSA: 72%; GRA: 19.7%; ENF: 8.3% | 1980-2016 | 2344-3695 | 189.00 | Pecos River - US |



Continuation of Table A5

| Station | Lat | Lon | Climate | Biome | Period | Elev | Area | River |
|---------|-----|-----|---------|-------|--------|------|------|-------|
| US_0006563 | 35.65 | -105.32 | Dfb: 82.2%; Dfc: 17.8% | WSA: 75.3%; GRA: 23.9%; ENF: 0.8% | 1980-2016 | 2174-3494 | 84.00 | Gallinas Creek - US |
| US_0006575 | 33.34 | -105.73 | Dfb: 100% | GRA: 58.6%; WSA: 39.7%; ENF: 1.7% | 1980-2016 | 2254-3391 | 18.30 | River Ruidoso - US |
| US_0006576 | 33.33 | -105.63 | Cfb: 62.9%; Dfb: 31.8%; BSk: 5.3% | GRA: 52.6%; WSA: 35.5%; SAV: 5% | 1980-2016 | 1992-3391 | 120.00 | River Ruidoso - US |
| US_0006577 | 33.39 | -105.72 | Dfb: 100% | GRA: 92%; CSH: 8% | 1980-2016 | 2385-3192 | 8.14 | Eagle Creek - US |
| US_0006602 | 32.85 | -107.97 | BSk: 70.9%; Dsb: 29.1% | SAV: 43.5%; GRA: 23.3%; CSH: 19% | 1980-2016 | 1813-3049 | 184.00 | Mimbres River - US |
| US_0006603 | 33.14 | -105.90 | Cfb: 35.7%; Dfb: 31.1%; BSk: 24.5% | WSA: 45%; GRA: 36.9%; SAV: 12.5% | 1980-2016 | 1681-2709 | 120.00 | Tularosa Creek - US |
| US_0006610 | 39.85 | -105.75 | Dfc: 57.5%; ET: 42.5% | GRA: 69.7%; WSA: 21.2%; SAV: 9.1% | 1980-2016 | 2993-3839 | 10.50 | Fraser River - US |
| US_0006661 | 39.55 | -106.40 | Dfc: 76.7%; ET: 23.1%; Dfb: 0.1% | WSA: 43.3%; GRA: 33.9%; SAV: 12.1% | 1980-2016 | 2478-4016 | 186.00 | Eagle River - US |
| US_0006683 | 39.21 | -106.80 | Dfc: 68.9%; ET: 31.1% | WSA: 38.4%; SAV: 29%; GRA: 26.8% | 1980-2016 | 2682-3845 | 41.70 | Hunter Creek - US |
| US_0006709 | 38.66 | -106.85 | Dfc: 82.7%; ET: 13.3%; Dfb: 3.6% | GRA: 47.1%; WSA: 25.9%; SAV: 24.3% | 1980-2016 | 2465-3973 | 289.00 | East River - US |
| US_0006735 | 38.18 | -107.75 | Dfc: 42.5%; Dfb: 26.5%; BSk: 15.6% | SAV: 45.5%; GRA: 40%; WSA: 7.8% | 1980-2016 | 2119-3968 | 149.00 | Uncompahgre River - US |
| US_0006855 | 40.13 | -111.02 | Dfb: 67.9%; Dfc: 32.1% | SAV: 58.7%; GRA: 41.3% | 1980-1994 | 2371-2938 | 43.00 | Strawberry River - US |
| US_0006948 | 37.04 | -107.88 | Dfc: 39%; Dfb: 34.6%; BSk: 15.3% | WSA: 41%; GRA: 36.8%; SAV: 20.2% | 1980-2016 | 1844-4088 | 1090.00 | Animas River - US |
| US_0006955 | 36.74 | -108.25 | BSk: 66.4%; Dfb: 23.9%; Dfc: 4.5% | GRA: 85.6%; SAV: 7.7%; OSH: 4.6% | 1980-2015 | 1605-3804 | 583.00 | La Plata River - US |
| US_0006973 | 34.02 | -109.46 | Dsb: 53.1%; Dsc: 34.7%; Dfb: 12.2% | GRA: 45.6%; WSA: 42.2%; ENF: 12.2% | 1980-2016 | 2555-3328 | 29.10 | Little Colorado River - US |
| US_0006984 | 35.28 | -108.55 | BSk: 88.8%; Dfb: 11.2% | GRA: 97%; SAV: 3% | 1980-2016 | 2170-2624 | 71.40 | River Nutria - US |
| US_0006995 | 34.67 | -111.01 | Dsb: 93.3%; BSk: 6.7% | WSA: 54.1%; SAV: 33.4%; ENF: 10.7% | 1980-1993 | 1927-2404 | 317.00 | Clear Creek - US |
| US_0007035 | 36.71 | -114.70 | BWh: 100% | GRA: 100% | 1980-2016 | 528-549 | 40.00 | Muddy River - US |
| US_0007048 | 36.12 | -114.90 | BWk: 44.2%; BWh: 43.4%; BSk: 11.1% | OSH: 41.5%; BSV: 28.3%; URB: 22.2% | 1980-2016 | 414-3379 | 1586.00 | Lv Wash - US |
| US_0007051 | 34.54 | -113.45 | BSk: 86.4%; BWh: 10.1%; BSh: 3.3% | GRA: 40.6%; OSH: 35.6%; SAV: 22.3% | 1980-2016 | 609-2128 | 601.00 | Burro Creek - US |
| US_0007053 | 34.31 | -113.35 | BSk: 67.3%; BWh: 19%; BSh: 12.4% | OSH: 54.6%; GRA: 20.9%; SAV: 20.5% | 1980-2016 | 435-2113 | 1129.00 | Santa Maria River - US |
| US_0007056 | 33.06 | -108.54 | BSk: 80.3%; Dsb: 17.9%; Dfb: 1.6% | SAV: 46.8%; GRA: 37.3%; WSA: 6.7% | 1980-2016 | 1454-3174 | 1864.00 | Gila River - US |
| US_0007057 | 33.17 | -108.65 | Dsb: 57%; BSk: 43% | SAV: 70.1%; GRA: 25.6%; CSH: 2.1% | 1980-2016 | 1755-3131 | 69.00 | Mogollon Creek - US |
| US_0007068 | 33.06 | -109.44 | BSk: 88%; Csb: 8.9%; Dsb: 3% | SAV: 35.1%; GRA: 24.1%; OSH: 19.9% | 1980-2016 | 1171-2839 | 622.00 | Eagle Creek - US |
| US_0007076 | 33.00 | -110.77 | BSk: 66.9%; BWk: 14.9%; Dsb: 6.5% | OSH: 44.7%; GRA: 24.5%; SAV: 21% | 1980-2004 | 617-3227 | 382.00 | Gila River - US |
| US_0007084 | 32.84 | -110.63 | BSk: 86.8%; BSh: 13.1%; Csa: 0.1% | OSH: 69.3%; GRA: 20.2%; SAV: 10.3% | 1980-2016 | 732-2293 | 537.00 | Aravaipa Creek - US |



Continuation of Table A5

| Station | Lat | Lon | Climate | Biome | Period | Elev | Area | River |
|---------|-----|-----|---------|-------|--------|------|------|-------|
| US_0007152 | 34.32 | -112.06 | BSk: 88.9%; Csa: 7%; Csb: 4.1% | GRA: 53.9%; SAV: 20%; OSH: 13.4% | 1980-2016 | 1070-2277 | 585.00 | Agua Fria River - US |
| US_0007153 | 34.02 | -112.17 | BSk: 76.2%; BSh: 12.1%; Csa: 7.4% | GRA: 49.2%; OSH: 24.7%; SAV: 17.2% | 1980-2016 | 564-2281 | 1111.00 | Agua Fria River - US |
| US_0007488 | 42.79 | -118.87 | BSk: 75.5%; Dsb: 12.9%; Dsc: 11.6% | GRA: 100% | 1980-2016 | 1371-2875 | 200.00 | Donner Und Blitzen River - US |
| US_0008472 | 48.42 | -116.50 | Dsb: 63.3%; Dsc: 35.8%; Dfc: 0.9% | ENF: 46.8%; WSA: 34.7%; GRA: 14.1% | 1980-2016 | 647-2131 | 124.00 | Pack River - US |
| US_0008473 | 48.45 | -116.90 | Dsb: 58%; Dsc: 18.3%; Dfc: 12.8% | ENF: 78.4%; WSA: 14.8%; GRA: 4.7% | 1980-2006 | 727-2128 | 611.00 | Priest River - US |
| US_0008474 | 48.22 | -116.91 | Dsb: 70.5%; Dsc: 13.9%; Dfc: 8.4% | ENF: 77.3%; WSA: 16.2%; GRA: 3.3% | 1980-2016 | 643-2128 | 902.00 | Priest River - US |
| US_0008483 | 47.71 | -115.98 | Dsb: 98.8%; Dsc: 1.2% | ENF: 97.2%; WSA: 2.4%; MF: 0.4% | 1980-2016 | 830-1775 | 335.00 | Nf Coeur D Alene River - US |
| US_0008484 | 47.57 | -116.25 | Dsb: 97.1%; Dsc: 2.9% | ENF: 96.2%; WSA: 3.2%; GRA: 0.2% | 1980-2016 | 665-1975 | 895.00 | Nf Coeur D Alene River - US |
| US_0008528 | 47.74 | -120.37 | Dsb: 54.8%; Dsc: 45.2% | ENF: 45.9%; GRA: 33.9%; WSA: 14.4% | 1980-2016 | 426-1946 | 92.40 | Mad River - US |
| US_0008570 | 43.86 | -110.59 | Dfc: 99.6%; ET: 0.4% | GRA: 39.4%; WSA: 31.2%; SAV: 24.9% | 1980-2016 | 2060-3450 | 807.00 | Snake River - US |
| US_0008583 | 43.14 | -110.98 | Dfc: 83.7%; Dfb: 16.3% | GRA: 49.6%; WSA: 26.2%; SAV: 24.1% | 1980-2016 | 1837-3319 | 448.00 | Greys River - US |
| US_0008606 | 43.44 | -111.73 | Dfb: 84.5%; BSk: 14.8%; Dfc: 0.6% | GRA: 84.4%; SAV: 8.9%; WSA: 4.1% | 1980-2016 | 1619-2821 | 568.00 | Willow Creek - US |
| US_0008652 | 43.49 | -114.06 | Dsb: 24.8%; Dfc: 24.2%; Dfb: 20.2% | GRA: 86.2%; SAV: 12.4%; BSV: 1% | 1980-2016 | 1652-3415 | 248.00 | Little Wood River - US |
| US_0008672 | 43.50 | -115.31 | Dsb: 56%; Dsc: 44% | GRA: 61%; SAV: 35.9%; WSA: 2.1% | 1980-2016 | 1302-2988 | 635.00 | Sf Boise River - US |
| US_0008685 | 43.57 | -118.21 | BSk: 75%; Dsb: 22.6%; Dsc: 2.4% | GRA: 81.7%; WSA: 15%; SAV: 1.9% | 1980-2016 | 1032-2412 | 1100.00 | Malheur River - US |
| US_0008687 | 43.91 | -118.15 | BSk: 60%; Dsb: 31.6%; Dsc: 8.4% | GRA: 75.7%; WSA: 17.1%; SAV: 4.3% | 1980-2016 | 1000-2329 | 440.00 | North Fork Malheur River - US |
| US_0008696 | 44.91 | -116.00 | Dsc: 99.6%; Dsb: 0.4% | WSA: 42.3%; SAV: 34.5%; GRA: 22.2% | 1980-2016 | 1619-2557 | 48.90 | Lake Fork Payette River - US |
| US_0008698 | 44.52 | -116.05 | Dsb: 53.9%; Dsc: 46.1% | WSA: 48.1%; GRA: 32.9%; SAV: 16.6% | 1980-2015 | 1469-2557 | 616.00 | Nf Payette River - US |
| US_0008710 | 44.29 | -116.78 | Dsa: 54%; Dsb: 25.7%; BSk: 20.3% | GRA: 99.2%; BSV: 0.6%; CRO: 0.2% | 1980-2016 | 696-1547 | 288.00 | Crane Creek - US |
| US_0008720 | 44.95 | -116.87 | Dsb: 51.2%; Dsa: 21.9%; Dsc: 18.5% | GRA: 52.8%; SAV: 20%; WSA: 17.8% | 1980-1996 | 621-2717 | 230.00 | Pine Creek - US |
| US_0008723 | 44.22 | -114.93 | Dsc: 83.7%; Dfc: 16.3% | GRA: 56.3%; WSA: 24.7%; SAV: 17.3% | 1980-2016 | 1905-2993 | 147.00 | Valley Creek - US |
| US_0008726 | 44.30 | -114.48 | Dfc: 96.6%; Dfb: 3.4% | SAV: 57.7%; GRA: 19.2%; WSA: 15.4% | 1980-2016 | 1936-2626 | 6.29 | Bruno Creek - US |
| US_0008766 | 46.37 | -116.16 | Dsb: 100% | ENF: 74.8%; WSA: 11.9%; GRA: 10.1% | 1980-2016 | 467-1782 | 243.00 | Lolo Creek - US |



*Author contributions.* Conceptualization, coding and first draft writing:DS. All authors provided scientific input, reviewed and edited the final manuscript

*Competing interests.* The authors declare no competing interests

*Acknowledgements.* DS, ICP and RN acknowledge support from the European Research Council under the European Union's Horizon 2020 research and innovation programme (Grant Agreement No: 787203 REALM). DS also acknowledges support from the National Secretariat for Higher Education, Science, Technology, and Innovation of Ecuador (Grant Agreement No:CZ02-000388-2017). This research is a contribution to the Land Ecosystem Models based On New Theory, obseRvations and ExperimEnts (LEMONTREE) project funded through the generosity of Eric and Wendy Schmidt by recommendation of the Schmidt Futures program.



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
