# Peer review of "Simple process-led algorithms for simulating habitats (SPLASH v.2.0): robust calculations of water and energy fluxes"

_EGUsphere, 2023_

## Referee Comment (RC1)

The article presents an update to the SPLASH model, incorporating analytical solutions to reduce its computational demand. These analytical solutions consider topographical characteristics that are often overlooked in land-surface models, such as the effect of terrain slope on infiltration. Additionally, the article describes the model's parameterization using global datasets of observed soil textural properties and meteorological data. To describe different water and energy processes, the model employs a set of equations selected and calibrated using the observed datasets. Overall, the model demonstrates good performance in representing fluxes and environmental states, even outperforming the benchmarked VIC model for some environmental variables. Therefore, I consider that with minor correction this manuscript can be published in the GMD journal as it will be a good contribution to the modelling community.

Please find below some general and specific comments about the article.

I wonder if the authors could provide a general overview of the model structure at the beginning of the model description, including, if possible, a schematic representation. This would help the reader understand the connection between the different components described in subsequent sections. The author might also consider adding a diagram to illustrate the model parameterization. This would guide the reader through the description of the method sections, showing the dataset used for parameter calibration and the one used for validation. Additionally, it would be useful for the reader to find model parameters and variables summarized in one or two tables. The table contents should specify the type of variable (input/output/parameters), the corresponding units, and the potential source of information.

If one of the main goals of the manuscript is to enhance the current version of the SPLASH model, I wonder why it was not also compared with previous versions of the model. Additionally, while it is mentioned that the new updates improve simulation times, it would be very useful for users to understand how the speed of this version compares to the previous one.

Is it possible to provide different values, not steady-state, for initial conditions, as specified in subsection 2.3? Or does this section only apply to the analysis performed in the manuscript? If so, the initial conditions section should be moved to a more appropriate location, perhaps to the simulation protocol section. Additionally, to reduce the influence of the initial conditions, is a warm-up period considered in the analysis performed later in the evaluation section?

Since global simulations have been added to the manuscript, it is not clear how the model was parameterized at the 1 and 5 km resolution. Specifically, how were model parameters aggregated into the coarser resolution? It would make easier to the reader to clearly describe in separated subsections how the point scale and global simulations were performed. In any case, results at the coarser scale show a decrease in performance for different variables, which is expected, as heterogeneity may not be well represented by spatial aggregation at the coarser resolution.

The discussion section appears to conclude abruptly, leaving the reader without clear takeaways from the article. To enhance the reader's understanding and provide a comprehensive summary, it is recommended to include a dedicated conclusion section. This section will emphasize the main findings and key insights obtained throughout the article.

Overall, it is essential to ensure that all figures and their accompanying labels or text are appropriately sized. The readability of text in the majority of figures is compromised when the document is printed.

Consider using specific titles for each section instead of simply starting with a generic term like "Methods." Given the manuscript's length, employing distinct and relevant names for each section will enhance readability. It in turn will facilitate the reader's ability to locate information pertaining to each specific numerical experiment. This is especially important for distinguishing between experiments conducted on a point scale and those involving global scale simulations.

Density plots require a colorbar to explain difference on colour variations, also, a more detailed description is required for figures description.

There are quite a lot of typos, so I recommend to carefully look at the entire document before submitting it again. Line 780: "Ssince", Line 360: "soving", Line 432: "abovementioned". Also, subscript text of some variables are not properly formatted, e.g. Figure 14 description.

Figure reference is missing in line 463.

The term $t_0$ in the equation 44 is not described.

Change "determination coefficient" to "coefficient of determination" in line 466

Figure 4 is misleading the reader as it shows that the water table within the cell varies not only with depth but also along the x/y axis, which is not true as the equations do not describe such variation. Therefore, I suggest that the author update the figure or provide a more specific description on the figure description.

Figure 15 has a very poor resolution, labels are unreadable. It would be good if the background can be deleted as it does not provide any additional or useful information.

The use of the first person to describe the authors' assumptions should, at the very least, reflect the contribution of all authors. Thus, I recommend the authors to use the "we" instead of "I" when the first person is used (e.g. Line: 694).

Line 480: Some parameters have already been described (Line 364), so to avoid repetition delete the ones that have been already described. Or are they different from the previous ones?

Line 483: information related to the HPC has already been mentioned in previous section (line 435), also the use of this specific HPC could be added to the acknowledgement section instead of the main section, as it does not add any additional information that contribute to the understanding of the article.

GMD do not recommend the use of footnotes as they usually disrupt the flow of the document (Line 98, 364, 370, 428, 496).

Line 500: what does the number 17, in brackets, means?

Line 574, what does the number 29 mean at the end the sentence, is it the figure number? If so please add the reference code.

Figure 36: Axis label of figure 36b, FTS, is not defined.

Figures 25, 30, 31, 32, 38, and 41: Add a description of climatic zones initials or refer the reader to the section where the climatic zones are defined.

---

## Author Comment (AC1)

The article presents an update to the SPLASH model, incorporating analytical solutions to reduce its computational demand. These analytical solutions consider topographical characteristics that are often overlooked in land-surface models, such as the effect of terrain slope on infiltration.
Additionally, the article describes the model's parameterization using global datasets of observed soil textural properties and meteorological data. To describe different water and energy processes, the model employs a set of equations selected and calibrated using the observed datasets. Overall, the model demonstrates good performance in representing fluxes and environmental states, even outperforming the benchmarked VIC model for some environmental variables. Therefore, I consider that with minor correction this manuscript can be published in the GMD journal as it will be a good contribution to the modelling community.

Please find below some general and specific comments about the article.

**RC1.1.** I wonder if the authors could provide a general overview of the model structure at the beginning of the model description, including, if possible, a schematic representation. This would help the reader understand the connection between the different components described in subsequent sections. The author might also consider adding a diagram to illustrate the model parameterization. This would guide the reader through the description of the method sections, showing the dataset used for parameter calibration and the one used for validation. Additionally, it would be useful for the reader to find model parameters and variables summarized in one or two tables. The table contents should specify the type of variable (input/output/parameters), the corresponding units, and the potential source of information.

A general overview of the model structure was already presented in Davis et al., (2017, GMD) when SPLASH V1.0 was introduced, together with tables describing variables' names and units. In this new version the general structure has not changed, nonetheless a general overview is provided again in the introduction.

**RC1.2.** If one of the main goals of the manuscript is to enhance the current version of the SPLASH model, I wonder why it was not also compared with previous versions of the model. Additionally, while it is mentioned that the new updates improve simulation times, it would be very useful for users to understand how the speed of this version compares to the previous one.

At the time of writing the manuscript our results were not compared with V1.0 because how slow and problematic that code was when running with data from the EC towers. Instead, we used the results from the original paper to compare with. Since then, and updated version of the code for V1.0 has been developed for a different application, making it possible to run with data from the flux towers.

So, following your suggestion we have included comparisons with results from SPLASH v1.0.
While doing the comparison, we noticed that Fig. 29, which evaluates condensation, was mistakenly showing data only from the same stations used to evaluate soil moisture. So, we updated the figure to include the data from all the flux towers. We also excluded crops to evaluate evapotranspiration, because irrigation was not reported in the original data and we only used precipitation as input for every site. The evaluations, replacing Figs. 22, 24 and 29 now look as follows:

[Figure]

Fig.22.

[Figure]

Fig. 24.

[Figure]

Fig. 29.

Regarding the simulation time, we were referring to the R version of splash V1.0, which is easier to run for average users. Version 2.0 is coded in C++ but wrapped in R, since C++ is hundreds of times faster than R, this was not a fair comparison. However, version 1.0 also has a C++ version, but very limited (the public version only runs 1 year). Comparing both versions in C++, they both take almost identical

time to run the spin-up + one year (~8.78 ms).

**RC1.3.** Is it possible to provide different values, not steady-state, for initial conditions, as specified in subsection 2.3? Or does this section only apply to the analysis performed in the manuscript? If so, the initial conditions section should be moved to a more appropriate location, perhaps to the simulation protocol section. Additionally, to reduce the influence of the initial conditions, is a warm- up period considered in the analysis performed later in the evaluation section?

With the current structure of the code it is technically not possible, the spin-up routine was considered as one of the assumptions of the model and it is hardcoded in the package. All the simulations presented in the manuscript were initialized after steady-state was reached. However, we understand that some applications may not need this feature, so in a future release we plan to incorporate an option to disable it and use user-specified initial conditions instead.

**RC1.4.** Since global simulations have been added to the manuscript, it is not clear how the model was parameterized at the 1 and 5 km resolution. Specifically, how were model parameters aggregated into the coarser resolution? It would make easier to the reader to clearly describe in separated subsections how the point scale and global simulations were performed. In any case, results at the coarser scale show a decrease in performance for different variables, which is expected, as heterogeneity may not be well represented by spatial aggregation at the coarser resolution.

The parameters of the model are global, they were obtained from global point-type observational datasets. We assumed they could remain the same for any resolution, which should hold better if higher spatial resolutions are used. The global simulations were done by looping the algorithm pixel by pixel. As you suggest, we are including these points in Section 4 (Methods: Simulation protocol and Performance evaluation) to add clarity.

Different forcing datasets in their native resolution were used. Only soil properties were resampled from the original from soilgrids (250m) to 1 and 5 km to match the resolution of the forcing data.

**RC1.5.** The discussion section appears to conclude abruptly, leaving the reader without clear takeaways from the article. To enhance the reader's understanding and provide a comprehensive summary, it is recommended to include a dedicated conclusion section. This section will emphasize the main findings and key insights obtained throughout the article.

Yes, we agree, so we included this text at the end of the discussion:

Despite the limitations discussed, the updated SPLASH model provides fast and parsimonious means to generate robust estimates of water and energy budgets across regions, regardless of their topographic complexity and spatial scale. The calibration-free approach enhances the model's portability and the ecological interpretability of the results. Moreover, since the structure of the model contains fewer moving parts than any other LSMs, it facilitates formulating ecologically-driven working hypotheses on why results do not match the observations in case of big discrepancies. With targeted refinements, SPLASH could become an even more robust tool for ecohydrological modeling and for exploring hydrological impacts of global change.

**Reviewer 2**

This manuscript presents an ambitious global modeling framework, describing terrestrial water and energy processes using many analytical solutions. It is a valuable modeling exercise. Reviewer #1 provided nice, detailed comments. Here my comments are relatively high level.

**RC2.1.** "Calibration-free" in the title is misleading or at least questionable. In fact, the authors used the words "calibrate" and "calibration" several times in their own method description, e.g., L390, L403, L425, L439. Perhaps, "parameter-parsimonious" is more appropriate.

Yes, we agree.

Although the model does not need local calibration, we settled in:

Simple process-led algorithms for simulating habitats (SPLASH v.2.0): robust calculations of water and energy fluxes

**RC2.2.** Using analytical solutions to describe water/energy processes in global LSMs is attempting, but may not be feasible for all the processes that are necessary in LSMs. Most analytical solutions are based on strong assumptions, and these assumptions may be valid at certain locations or time scales, but not at others. For example, the Green-Ampt equation was originally derived as a point-scale theory, hence a key assumption underpinning it is the (horizontal) homogeneity of soil properties and other relevant variables. Here "point scale" is roughly at the order of 1-10 square meters. When applied globally, each spatial unit (say a lat/lon grid) is much larger than 1 square kilometer and thus contains strong spatial heterogeneity. It is quite often that, within the same lat/lon grid, at the same time, there are patches already saturated, and others are unsaturated. Whether the Green-Ampt equation can be applied at such a spatial scale is highly questionable. The runoff scheme used in the VIC model, nonetheless, is not empirical but at least partially physically based. It explicitly embraces the spatial heterogeneity of soil moisture deficit levels in a spatial unit, hence theoretically more suitable to be used in LSMs for regional or global applications. The authors should at least point out these theoretical limitations (hence potential limitations in SPLASH applications) in their introduction and discussion.

Indeed, SPLASH use strong assumptions, which fail in some sites and work well in other. Nonetheless, SPLASH is capable to account for spatial heterogeneity in global simulations if it is feed with higher spatial resolution data. Whether the greater complexity and extra inputs (which also have uncertainties) added to VIC to represent the spatial heterogeneity is more suitable to be used in LSMs, as opposite to fewer inputs but higher spatial resolution, is certainly an idea we want to explore in the discussion, now that a comparison of runoff simulations with 1km of spatial resolution of SPLASH vs VIC is added to the manuscript.

**RC2.3.** More clarifications need to be provided on the comparison between SPLASH and VIC-3L at the global scale. Are they applied at the same spatiotemporal resolutions? Has VIC-3L been properly spun up and calibrated? Why not use more hydrologic data to compare these two models, e.g., observed streamflow data?

VIC was not run at the global scale in this manuscript; it was run at site locations with the same observed data from the stations as SPLASH. For these simulations VIC-3L was not calibrated nor spun-up, but the initial soil moisture was provided by Schaperow2020.
Following your suggestion, we compared the performance of SPLASH and VIC using hydrologic data from 133 watersheds in the US. We used DAYMET4 at 1km resolution as inputs. The VIC simulation (using the same forcing) comes from (Kao et al., 2022) and it was calibrated with historical data.

[Figure]

**RC2.4.** Many observed data are already available globally in a spatially distributed fashion (e.g., SWE). Showing the comparisons in the form of spatial maps might be more informative than climatic zones. For global LSMs, one main objective is to capture the spatial variability in the variables of interest. Overall, it is essential to ensure that all figures and their accompanying labels or text are appropriately sized. The readability of text in the majority of figures is compromised when the document is printed.

Although some data are available as spatial datasets some of these datasets use modeling, either to interpolate or to calculate quantities from remote sensing. To get a sense of the validity of our assumptions we aimed for evaluations, strictly with data observed in situ. We agree, some of the labels are not properly sized. This is fixed now.

**RC2.5.** Consider using specific titles for each section instead of simply starting with a generic term like "Methods." Given the manuscript's length, employing distinct and relevant names for each section will enhance readability. It in turn will facilitate the reader's ability to locate information pertaining to each specific numerical experiment. This is especially important for distinguishing between experiments conducted on a point scale and those involving global scale simulations.

Yes, we combined this reply with **RC1.4.**

**RC2.6.** Density plots require a colorbar to explain difference on colour variations, also, a more detailed description is required for figures description.

We added the following test in the captions: …. Color scale shows point density, red means higher point density.

**RC2.7.** There are quite a lot of typos, so I recommend to carefully look at the entire document before submitting it again. Line 780: "Ssince", Line 360: "soving", Line 432: "abovementioned". Also, subscript text of some variables are not properly formatted, e.g. Figure 14 description.

All the typos mentioned are fixed now.

**RC2.8.** Figure reference is missing in line 463.
The term t0 in the equation 44 is not described.
Change "determination coefficient" to "coefficient of determination" in line 466
Fixed

**RC2.9.** Figure 4 is misleading the reader as it shows that the water table within the cell varies not only with depth but also along the x/y axis, which is not true as the equations do not describe such variation. Therefore, I suggest that the author update the figure or provide a more specific description on the figure description.

Yes, the scale of the figure is misleading, the figure has been u[updated to show only the vertical profile of the soil moisture.

**RC2.10.** Figure 15 has a very poor resolution, labels are unreadable. It would be good if the background can be deleted as it does not provide any additional or useful information.
Figure 15 has been deleted, since it did not provide any meaningful information.

**RC2.11.** The use of the first person to describe the authors' assumptions should, at the very least, reflect the contribution of all authors. Thus, I recommend the authors to use the "we" instead of "I"

when the first person is used (e.g. Line: 694).

Line 480: Some parameters have already been described (Line 364), so to avoid repetition delete the ones that have been already described. Or are they different from the previous ones?

Line 483: information related to the HPC has already been mentioned in previous section (line 435), also the use of this specific HPC could be added to the acknowledgement section instead of the main section, as it does not add any additional information that contribute to the understanding of the article.

GMD do not recommend the use of footnotes as they usually disrupt the flow of the document (Line 98, 364, 370, 428, 496).

Line 500: what does the number 17, in brackets, means?

Line 574, what does the number 29 mean at the end the sentence, is it the figure number? If so please add the reference code.

Figure 36: Axis label of figure 36b, FTS, is not defined.

Figures 25, 30, 31, 32, 38, and 41: Add a description of climatic zones initials or refer the reader to the section where the climatic zones are defined.

Fixed exactly as suggested.